# Offline Reinforcement Learning with Differential Privacy

**Dan Qiao**
Department of Computer Science
UC Santa Barbara
Santa Barbara, CA 93106
danqiao@ucsb.edu

**Yu-Xiang Wang**
Department of Computer Science
UC Santa Barbara
Santa Barbara, CA 93106
yuxiangw@cs.ucsb.edu

## Abstract

The offline reinforcement learning (RL) problem is often motivated by the need to learn data-driven decision policies in financial, legal and healthcare applications. However, the learned policy could retain sensitive information of individuals in the training data (e.g., treatment and outcome of patients), thus susceptible to various privacy risks. We design offline RL algorithms with differential privacy guarantees which provably prevent such risks. These algorithms also enjoy strong instance-dependent learning bounds under both tabular and linear Markov Decision Process (MDP) settings. Our theory and simulation suggest that the privacy guarantee comes at (almost) no drop in utility comparing to the non-private counterpart for a medium-size dataset.

## 1 Introduction

Offline Reinforcement Learning (or batch RL) aims to learn a near-optimal policy in an unknown environment[1] through a static dataset gathered from some behavior policy $\mu$. Since offline RL does not require access to the environment, it can be applied to problems where interaction with environment is infeasible, *e.g.*, when collecting new data is costly (trade or finance [Zhang et al., 2020]), risky (autonomous driving [Sallab et al., 2017]) or illegal / unethical (healthcare [Raghu et al., 2017]). In such practical applications, the data used by an RL agent usually contains sensitive information. Take medical history for instance, for each patient, at each time step, the patient reports her health condition (age, disease, etc.), then the doctor decides the treatment (which medicine to use, the dosage of medicine, etc.), finally there is treatment outcome (whether the patient feels good, etc.) and the patient transitions to another health condition. Here, (health condition, treatment, treatment outcome) corresponds to (state, action, reward) and the dataset can be considered as $n$ (number of patients) trajectories sampled from a MDP with horizon $H$ (number of treatment steps). However, learning agents are known to implicitly memorize details of individual training data points verbatim [Carlini et al., 2019], even if they are irrelevant for learning [Brown et al., 2021], which makes offline RL models vulnerable to various privacy attacks.

Differential privacy (DP) [Dwork et al., 2006] is a well-established definition of privacy with many desirable properties. A differentially private offline RL algorithm will return a decision policy that is indistinguishable from a policy trained in an alternative universe any individual user is replaced, thereby preventing the aforementioned privacy risks. There is a surge of recent interest in developing RL algorithms with DP guarantees, but they focus mostly on the online setting [Vietri et al., 2020, Garcelon et al., 2021, Liao et al., 2021, Chowdhury and Zhou, 2021, Luyo et al., 2021].

---

[1]The environment is usually characterized by a Markov Decision Process (MDP) in this paper.

37th Conference on Neural Information Processing Systems (NeurIPS 2023).

Offline RL is arguably more practically relevant than online RL in the applications with sensitive data. For example, in the healthcare domain, online RL requires actively running new exploratory policies (clinical trials) with every new patient, which often involves complex ethical / legal clearances, whereas offline RL uses only historical patient records that are often accessible for research purposes. Clear communication of the adopted privacy enhancing techniques (e.g., DP) to patients was reported to further improve data access [Kim et al., 2017].

**Our contributions.** In this paper, we present the first provably efficient algorithms for offline RL with differential privacy. Our contributions are twofold.

- We design two new pessimism-based algorithms DP-APVI (Algorithm 1) and DP-VAPVI (Algorithm 2), one for the tabular setting (finite states and actions), the other for the case with linear function approximation (under linear MDP assumption). Both algorithms enjoy DP guarantees (pure DP or zCDP) and instance-dependent learning bounds where the cost of privacy appears as lower order terms.
- We perform numerical simulations to evaluate and compare the performance of our algorithm DP-VAPVI (Algorithm 2) with its non-private counterpart VAPVI [Yin et al., 2022] as well as a popular baseline PEVI [Jin et al., 2021]. The results complement the theoretical findings by demonstrating the practicality of DP-VAPVI under strong privacy parameters.

**Related work.** To our knowledge, differential privacy in offline RL tasks has not been studied before, except for much simpler cases where the agent only evaluates a single policy [Balle et al., 2016, Xie et al., 2019]. Balle et al. [2016] privatized first-visit Monte Carlo-Ridge Regression estimator by an output perturbation mechanism and Xie et al. [2019] used DP-SGD. Neither paper considered offline learning (or policy optimization), which is our focus.

There is a larger body of work on private RL in the online setting, where the goal is to minimize regret while satisfying either joint differential privacy [Vietri et al., 2020, Chowdhury and Zhou, 2021, Ngo et al., 2022, Luyo et al., 2021] or local differential privacy [Garcelon et al., 2021, Liao et al., 2021, Luyo et al., 2021, Chowdhury and Zhou, 2021]. The offline setting introduces new challenges in DP as we cannot *algorithmically enforce* good "exploration", but have to work with a static dataset and privately estimate the uncertainty in addition to the value functions. A private online RL algorithm can sometimes be adapted for private offline RL too, but those from existing work yield suboptimal and non-adaptive bounds. We give a more detailed technical comparison in Appendix B.

Among non-private offline RL works, we build directly upon non-private offline RL methods known as Adaptive Pessimistic Value Iteration (APVI, for tabular MDPs) [Yin and Wang, 2021b] and Variance-Aware Pessimistic Value Iteration (VAPVI, for linear MDPs) [Yin et al., 2022], as they give the strongest theoretical guarantees to date. We refer readers to Appendix B for a more extensive review of the offline RL literature. Introducing DP to APVI and VAPVI while retaining the same sample complexity (modulo lower order terms) require nontrivial modifications to the algorithms.

**A remark on technical novelty.** Our algorithms involve substantial technical innovation over previous works on online DP-RL with joint DP guarantee[2]. Different from previous works, our DP-APVI (Algorithm 1) operates on Bernstein type pessimism, which requires our algorithm to deal with conditional variance using private statistics. Besides, our DP-VAPVI (Algorithm 2) replaces the LSVI technique with variance-aware LSVI (also known as weighted ridge regression, first appears in [Zhou et al., 2021]). Our DP-VAPVI releases conditional variance privately, and further applies weighted ridge regression privately. Both approaches ensure tighter instance-dependent bounds on the suboptimality of the learned policy.

## 2 Problem Setup

**Markov Decision Process.** A finite-horizon *Markov Decision Process* (MDP) is denoted by a tuple $M = (\mathcal{S}, \mathcal{A}, P, r, H, d_1)$ [Sutton and Barto, 2018], where $\mathcal{S}$ is state space and $\mathcal{A}$ is action space. A non-stationary transition kernel $P_h : \mathcal{S} \times \mathcal{A} \times \mathcal{S} \mapsto [0,1]$ maps each state action $(s_h, a_h)$ to a probability distribution $P_h(\cdot|s_h, a_h)$ and $P_h$ can be different across time. Besides, $r_h : \mathcal{S} \times A \mapsto \mathbb{R}$ is the expected immediate reward satisfying $0 \le r_h \le 1$, $d_1$ is the initial state distribution and $H$ is the horizon.

---

[2]Here we only compare our techniques (for offline RL) with the works for online RL under joint DP guarantee, as both settings allow access to the raw data.

A policy $\pi = (\pi_1, \cdots, \pi_H)$ assigns each state $s_h \in \mathcal{S}$ a probability distribution over actions according to the map $s_h \mapsto \pi_h(\cdot|s_h), \forall h \in [H]$. A random trajectory $s_1, a_1, r_1, \cdots, s_H, a_H, r_H, s_{H+1}$ is generated according to $s_1 \sim d_1, a_h \sim \pi_h(\cdot|s_h), r_h \sim r_h(s_h, a_h), s_{h+1} \sim P_h(\cdot|s_h, a_h), \forall h \in [H]$.

For tabular MDP, we have $\mathcal{S} \times \mathcal{A}$ is the discrete state-action space and $S := |\mathcal{S}|, A := |\mathcal{A}|$ are finite. In this work, we assume that $r$ is known[3]. In addition, we denote the per-step marginal state-action occupancy $d_h^\pi(s, a)$ as: $d_h^\pi(s, a) := \mathbb{P}[s_h = s|s_1 \sim d_1, \pi] \cdot \pi_h(a|s)$, which is the marginal state-action probability at time $h$.

**Value function, Bellman (optimality) equations.** The value function $V_h^\pi(\cdot)$ and Q-value function $Q_h^\pi(\cdot, \cdot)$ for any policy $\pi$ is defined as: $V_h^\pi(s) = \mathbb{E}_\pi[\sum_{t=h}^H r_t|s_h = s], \quad Q_h^\pi(s, a) = \mathbb{E}_\pi[\sum_{t=h}^H r_t|s_h = s, a_h = s, a], \forall h, s, a \in [H] \times \mathcal{S} \times \mathcal{A}$. The performance is defined as $v^\pi := \mathbb{E}_{d_1}[V_1^\pi] = \mathbb{E}_{\pi, d_1}\left[\sum_{t=1}^H r_t\right]$. The Bellman (optimality) equations follow $\forall h \in [H]$: $Q_h^\pi = r_h + P_h V_{h+1}^\pi, \quad V_h^\pi = \mathbb{E}_{a \sim \pi_h}[Q_h^\pi], \quad Q_h^\star = r_h + P_h V_{h+1}^\star, \quad V_h^\star = \max_a Q_h^\star(\cdot, a)$.

**Linear MDP [Jin et al., 2020b].** An episodic MDP $(\mathcal{S}, \mathcal{A}, H, P, r)$ is called a linear MDP with known feature map $\phi : \mathcal{S} \times \mathcal{A} \to \mathbb{R}^d$ if there exist $H$ unknown signed measures $\nu_h \in \mathbb{R}^d$ over $\mathcal{S}$ and $H$ unknown reward vectors $\theta_h \in \mathbb{R}^d$ such that

$$P_h(s' \mid s, a) = \langle \phi(s, a), \nu_h(s') \rangle, \quad r_h(s, a) = \langle \phi(s, a), \theta_h \rangle, \quad \forall (h, s, a, s') \in [H] \times \mathcal{S} \times \mathcal{A} \times \mathcal{S}.$$

Without loss of generality, we assume $\|\phi(s, a)\|_2 \le 1$ and $\max(\|\nu_h(\mathcal{S})\|_2, \|\theta_h\|_2) \le \sqrt{d}$ for all $h, s, a \in [H] \times \mathcal{S} \times \mathcal{A}$. An important property of linear MDP is that the value functions are linear in the feature map, which is summarized in Lemma E.14.

**Offline setting and the goal.** The offline RL requires the agent to find a policy $\pi$ in order to maximize the performance $v^\pi$, given only the episodic data $\mathcal{D} = \{(s_h^\tau, a_h^\tau, r_h^\tau, s_{h+1}^\tau)\}_{\tau \in [n]}^{h \in [H]}$[4] rolled out from some fixed and possibly unknown behavior policy $\mu$, which means we cannot change $\mu$ and in particular we do not assume the functional knowledge of $\mu$. In conclusion, based on the batch data $\mathcal{D}$ and a targeted accuracy $\epsilon > 0$, the agent seeks to find a policy $\pi_{\text{alg}}$ such that $v^\star - v^{\pi_{\text{alg}}} \le \epsilon$.

## 2.1 Assumptions in offline RL

In order to show that our privacy-preserving algorithms can generate near optimal policy, certain coverage assumptions are needed. In this section, we will list the assumptions we use in this paper.

**Assumptions for tabular setting.**

**Assumption 2.1** ([Liu et al., 2019]). *There exists one optimal policy $\pi^\star$, such that $\pi^\star$ is fully covered by $\mu$, i.e. $\forall s_h, a_h \in \mathcal{S} \times \mathcal{A}, d_h^{\pi^\star}(s_h, a_h) > 0$ only if $d_h^\mu(s_h, a_h) > 0$. Furthermore, we denote the trackable set as $\mathcal{C}_h := \{(s_h, a_h) : d_h^\mu(s_h, a_h) > 0\}$.*

Assumption 2.1 is the weakest assumption needed for accurately learning the optimal value $v^\star$ by requiring $\mu$ to trace the state-action space of one optimal policy ($\mu$ can be agnostic at other locations). Similar to [Yin and Wang, 2021b], we will use Assumption 2.1 for the tabular part of this paper, which enables comparison between our sample complexity to the conclusion in [Yin and Wang, 2021b], whose algorithm serves as a non-private baseline.

**Assumptions for linear setting.** First, we define the expectation of covariance matrix under the behavior policy $\mu$ for all time step $h \in [H]$ as below:

$$\Sigma_h^p := \mathbb{E}_\mu\left[\phi(s_h, a_h)\phi(s_h, a_h)^\top\right]. \tag{1}$$

As have been shown in [Wang et al., 2021a, Yin et al., 2022], learning a near-optimal policy from offline data requires coverage assumptions. Here in linear setting, such coverage is characterized by the minimum eigenvalue of $\Sigma_h^p$. Similar to [Yin et al., 2022], we apply the following assumption for the sake of comparison.

**Assumption 2.2** (Feature Coverage, Assumption 2 in [Wang et al., 2021a]). *The data distributions $\mu$ satisfy the minimum eigenvalue condition: $\forall h \in [H], \kappa_h := \lambda_{\min}(\Sigma_h^p) > 0$. Furthermore, we denote $\kappa = \min_h \kappa_h$.*

---

[3]This is due to the fact that the uncertainty of reward function is dominated by that of transition kernel in RL.

[4]For clarity we use $n$ for tabular MDP and $K$ for linear MDP when referring to the sample complexity.

## 2.2 Differential Privacy in offline RL

In this work, we aim to design privacy-preserving algorithms for offline RL. We apply differential privacy as the formal notion of privacy. Below we revisit the definition of differential privacy.

**Definition 2.3** (Differential Privacy [Dwork et al., 2006])**.** *A randomized mechanism $M$ satisfies $(\epsilon, \delta)$-differential privacy ($(\epsilon, \delta)$-DP) if for all neighboring datasets $U, U'$ that differ by one data point and for all possible event $E$ in the output range, it holds that*

$$\mathbb{P}[M(U) \in E] \le e^\epsilon \cdot \mathbb{P}[M(U') \in E] + \delta.$$

*When $\delta = 0$, we say pure DP, while for $\delta > 0$, we say approximate DP.*

In the problem of offline RL, the dataset consists of several trajectories, therefore one data point in Definition 2.3 refers to one single trajectory. Hence the definition of Differential Privacy means that the difference in the distribution of the output policy resulting from replacing one trajectory in the dataset will be small. In other words, an adversary can not infer much information about any single trajectory in the dataset from the output policy of the algorithm.

**Remark 2.4.** *For a concrete motivating example, please refer to the first paragraph of Introduction. We remark that our definition of DP is consistent with Joint DP and Local DP defined under the online RL setting where JDP/LDP also cast each user as one trajectory and provide user-wise privacy protection. For detailed definitions and more discussions about JDP/LDP, please refer to Qiao and Wang [2023a].*

During the whole paper, we will use zCDP (defined below) as a surrogate for DP, since it enables cleaner analysis for privacy composition and Gaussian mechanism. The properties of zCDP (e.g., composition, conversion formula to DP) are deferred to Appendix E.3.

**Definition 2.5** (zCDP [Dwork and Rothblum, 2016, Bun and Steinke, 2016])**.** *A randomized mechanism $M$ satisfies $\rho$-Zero-Concentrated Differential Privacy ($\rho$-zCDP), if for all neighboring datasets $U, U'$ and all $\alpha \in (1, \infty)$,*

$$D_\alpha(M(U)\|M(U')) \le \rho\alpha,$$

*where $D_\alpha$ is the Renyi-divergence [Van Erven and Harremos, 2014].*

Finally, we go over the definition and privacy guarantee of Gaussian mechanism.

**Definition 2.6** (Gaussian Mechanism [Dwork et al., 2014])**.** *Define the $\ell_2$ sensitivity of a function $f : \mathbb{N}^\mathcal{X} \mapsto \mathbb{R}^d$ as*

$$\Delta_2(f) = \sup_{neighboring\ U, U'} \|f(U) - f(U')\|_2.$$

*The Gaussian mechanism $\mathcal{M}$ with noise level $\sigma$ is then given by*

$$\mathcal{M}(U) = f(U) + \mathcal{N}(0, \sigma^2 I_d).$$

**Lemma 2.7** (Privacy guarantee of Gaussian mechanism [Dwork et al., 2014, Bun and Steinke, 2016])**.** *Let $f : \mathbb{N}^\mathcal{X} \mapsto \mathbb{R}^d$ be an arbitrary $d$-dimensional function with $\ell_2$ sensitivity $\Delta_2$. Then for any $\rho > 0$, Gaussian Mechanism with parameter $\sigma^2 = \frac{\Delta_2^2}{2\rho}$ satisfies $\rho$-zCDP. In addition, for all $0 < \delta, \epsilon < 1$, Gaussian Mechanism with parameter $\sigma = \frac{\Delta_2}{\epsilon}\sqrt{2\log\frac{1.25}{\delta}}$ satisfies $(\epsilon, \delta)$-DP.*

We emphasize that the privacy guarantee covers any input data. It does *not* require any distributional assumptions on the data. The RL-specific assumptions (e.g., linear MDP and coverage assumptions) are only used for establishing provable utility guarantees.

## 3 Results under tabular MDP: DP-APVI (Algorithm 1)

For reinforcement learning, the tabular MDP setting is the most well-studied setting and our first result applies to this regime. We begin with the construction of private counts.

**Private Model-based Components.** Given data $\mathcal{D} = \{(s_h^\tau, a_h^\tau, r_h^\tau, s_{h+1}^\tau)\}_{\tau \in [n]}^{h \in [H]}$, we denote $n_{s_h, a_h} := \sum_{\tau=1}^n \mathbb{1}[s_h^\tau, a_h^\tau = s_h, a_h]$ be the total counts that visit $(s_h, a_h)$ pair at time $h$ and $n_{s_h, a_h, s_{h+1}} :=$

$\sum_{\tau=1}^{n} \mathbb{1}\left[s_h^\tau, a_h^\tau, s_{h+1}^\tau = s_h, a_h, s_{h+1}\right]$ be the total counts that visit $(s_h, a_h, s_{h+1})$ pair at time $h$, then given the budget $\rho$ for zCDP, we add *independent* Gaussian noises to all the counts:

$$n'_{s_h,a_h} = \left\{n_{s_h,a_h} + \mathcal{N}(0, \sigma^2)\right\}^+, \ n'_{s_h,a_h,s_{h+1}} = \left\{n_{s_h,a_h,s_{h+1}} + \mathcal{N}(0, \sigma^2)\right\}^+, \ \sigma^2 = \frac{2H}{\rho}. \quad (2)$$

However, after adding noise, the noisy counts $n'$ may not satisfy $n'_{s_h,a_h} = \sum_{s_{h+1}\in\mathcal{S}} n'_{s_h,a_h,s_{h+1}}$. To address this problem, we choose the private counts of visiting numbers as the solution to the following optimization problem (here $E_\rho = 4\sqrt{\frac{H \log \frac{4HS^2A}{\delta}}{\rho}}$ is chosen as a high probability uniform bound of the noises we add):

$$\{\widetilde{n}_{s_h,a_h,s'}\}_{s'\in\mathcal{S}} = \text{argmin}_{\{x_{s'}\}_{s'\in\mathcal{S}}} \max_{s'\in\mathcal{S}} \left|x_{s'} - n'_{s_h,a_h,s'}\right|$$

$$\text{such that } \left|\sum_{s'\in\mathcal{S}} x_{s'} - n'_{s_h,a_h}\right| \le \frac{E_\rho}{2} \text{ and } x_{s'} \ge 0, \forall\, s' \in \mathcal{S}. \quad (3)$$

$$\widetilde{n}_{s_h,a_h} = \sum_{s'\in\mathcal{S}} \widetilde{n}_{s_h,a_h,s'}.$$

**Remark 3.1** (Some explanations). *The optimization problem above serves as a post-processing step which will not affect the DP guarantee of our algorithm. Briefly speaking, (3) finds a set of noisy counts such that $\widetilde{n}_{s_h,a_h} = \sum_{s'\in\mathcal{S}} \widetilde{n}_{s_h,a_h,s'}$ and the estimation error for each $\widetilde{n}_{s_h,a_h}$ and $\widetilde{n}_{s_h,a_h,s'}$ is roughly $E_\rho$.[5] In contrast, if we directly take the crude approach that $\widetilde{n}_{s_h,a_h,s_{h+1}} = n'_{s_h,a_h,s_{h+1}}$ and $\widetilde{n}_{s_h,a_h} = \sum_{s_{h+1}\in\mathcal{S}} \widetilde{n}_{s_h,a_h,s_{h+1}}$, we can only derive $|\widetilde{n}_{s_h,a_h} - n_{s_h,a_h}| \le \widetilde{O}(\sqrt{S}E_\rho)$ through concentration on summation of $S$ i.i.d. Gaussian noises. In conclusion, solving the optimization problem (3) enables tight analysis for the lower order term (the additional cost of privacy).*

**Remark 3.2** (Computational efficiency). *The optimization problem (3) can be reformulated as:*

$$\min \ t, \ s.t. \ |x_{s'} - n'_{s_h,a_h,s'}| \le t \text{ and } x_{s'} \ge 0 \ \forall\, s' \in \mathcal{S}, \ \left|\sum_{s'\in\mathcal{S}} x_{s'} - n'_{s_h,a_h}\right| \le \frac{E_\rho}{2}. \quad (4)$$

*Note that (4) is a* Linear Programming *problem with $S+1$ variables and $2S+2$ linear constraints (one constraint on absolute value is equivalent to two linear constraints), which can be solved efficiently by the simplex method [Ficken, 2015] or other provably efficient algorithms [Nemhauser and Wolsey, 1988]. Therefore, our Algorithm 1 is computationally friendly.*

The private estimation of the transition kernel is defined as:

$$\widetilde{P}_h(s'|s_h, a_h) = \frac{\widetilde{n}_{s_h,a_h,s'}}{\widetilde{n}_{s_h,a_h}}, \quad (5)$$

if $\widetilde{n}_{s_h,a_h} > E_\rho$ and $\widetilde{P}_h(s'|s_h, a_h) = \frac{1}{S}$ otherwise.

**Remark 3.3.** *Different from the transition kernel estimate in previous works [Vietri et al., 2020, Chowdhury and Zhou, 2021] that may not be a distribution, we have to ensure that ours is a probability distribution, because our Bernstein type pessimism (line 5 in Algorithm 1) needs to take variance over this transition kernel estimate. The intuition behind the construction of our private transition kernel is that, for those state-action pairs with $\widetilde{n}_{s_h,a_h} \le E_\rho$, we can not distinguish whether the non-zero private count comes from noise or actual visitation. Therefore we only take the empirical estimate of the state-action pairs with sufficiently large $\widetilde{n}_{s_h,a_h}$.*

---

[5]This conclusion is summarized in Lemma C.3.

---

**Algorithm 1** Differentially Private Adaptive Pessimistic Value Iteration (DP-APVI)

---

1: **Input:** Offline dataset $\mathcal{D} = \{(s_h^\tau, a_h^\tau, r_h^\tau, s_{h+1}^\tau)\}_{\tau,h=1}^{n,H}$. Reward function $r$. Constants $C_1 = \sqrt{2}, C_2 = 16, C > 1$, failure probability $\delta$, budget for zCDP $\rho$.

2: **Initialization:** Calculate $\widetilde{n}_{s_h,a_h}, \widetilde{n}_{s_h,a_h,s_{h+1}}$ as (3), $\widetilde{P}_h(s_{h+1}|s_h,a_h)$ as (5). $\widetilde{V}_{H+1}(\cdot) \leftarrow 0$. $E_\rho \leftarrow 4\sqrt{\frac{H \log \frac{4HS^2A}{\delta}}{\rho}}$. $\iota \leftarrow \log(HSA/\delta)$.

3: **for** $h = H, H-1, \ldots, 1$ **do**

4:     $\widetilde{Q}_h(\cdot,\cdot) \leftarrow r_h(\cdot,\cdot) + (\widetilde{P}_h \cdot \widetilde{V}_{h+1})(\cdot,\cdot)$

5:     $\forall s_h, a_h$, let $\Gamma_h(s_h,a_h) \leftarrow C_1 \sqrt{\frac{\mathrm{Var}_{\widetilde{P}_{s_h,a_h}}(\widetilde{V}_{h+1})\cdot\iota}{\widetilde{n}_{s_h,a_h} - E_\rho}} + \frac{C_2 SHE_\rho \cdot \iota}{\widetilde{n}_{s_h,a_h}}$ if $\widetilde{n}_{s_h,a_h} > E_\rho$, otherwise $CH$.

6:     $\widehat{Q}_h^p(\cdot,\cdot) \leftarrow \widetilde{Q}_h(\cdot,\cdot) - \Gamma_h(\cdot,\cdot)$.

7:     $\overline{Q}_h(\cdot,\cdot) \leftarrow \min\{\widehat{Q}_h^p(\cdot,\cdot), H-h+1\}^+$.

8:     $\forall s_h$, let $\widehat{\pi}_h(\cdot|s_h) \leftarrow \mathrm{argmax}_{\pi_h}\langle \overline{Q}_h(s_h,\cdot), \pi_h(\cdot|s_h)\rangle$ and $\widetilde{V}_h(s_h) \leftarrow \langle \overline{Q}_h(s_h,\cdot), \widehat{\pi}_h(\cdot|s_h)\rangle$.

9: **end for**

10: **Output:** $\{\widehat{\pi}_h\}$.

---

**Algorithmic design.** Our algorithmic design originates from the idea of pessimism, which holds conservative view towards the locations with high uncertainty and prefers the locations we have more confidence about. Based on the Bernstein type pessimism in APVI [Yin and Wang, 2021b], we design a similar pessimistic algorithm with private counts to ensure differential privacy. If we replace $\widetilde{n}$ and $\widetilde{P}$ with $n$ and $\widehat{P}$[6], then our DP-APVI (Algorithm 1) will degenerate to APVI. Compared to the pessimism defined in APVI, our pessimistic penalty has an additional term $\widetilde{O}\left(\frac{SHE_\rho}{\widetilde{n}_{s_h,a_h}}\right)$, which accounts for the additional pessimism due to our application of private statistics.

We state our main theorem about DP-APVI below, the proof sketch is deferred to Appendix C.1 and detailed proof is deferred to Appendix C due to space limit.

**Theorem 3.4.** *DP-APVI (Algorithm 1) satisfies $\rho$-zCDP. Furthermore, under Assumption 2.1, denote $\bar{d}_m := \min_{h\in[H]}\{d_h^\mu(s_h,a_h) : d_h^\mu(s_h,a_h) > 0\}$. For any $0 < \delta < 1$, there exists constant $c_1 > 0$, such that when $n > c_1 \cdot \max\{H^2, E_\rho\}/\bar{d}_m \cdot \iota$ ($\iota = \log(HSA/\delta)$), with probability $1-\delta$, the output policy $\widehat{\pi}$ of DP-APVI satisfies*

$$0 \le v^\star - v^{\widehat{\pi}} \le 4\sqrt{2}\sum_{h=1}^H \sum_{(s_h,a_h)\in\mathcal{C}_h} d_h^{\pi^\star}(s_h,a_h)\sqrt{\frac{\mathrm{Var}_{P_h(\cdot|s_h,a_h)}(V_{h+1}^\star(\cdot))\cdot\iota}{nd_h^\mu(s_h,a_h)}} + \widetilde{O}\left(\frac{H^3 + SH^2E_\rho}{n\cdot\bar{d}_m}\right),$$
(6)

*where $\widetilde{O}$ hides constants and Polylog terms, $E_\rho = 4\sqrt{\frac{H\log\frac{4HS^2A}{\delta}}{\rho}}$.*

**Comparison to non-private counterpart APVI [Yin and Wang, 2021b].** According to Theorem 4.1 in [Yin and Wang, 2021b], the sub-optimality bound of APVI is for large enough $n$, with high probability, the output $\widehat{\pi}$ satisfies:

$$0 \le v^\star - v^{\widehat{\pi}} \le \widetilde{O}\left(\sum_{h=1}^H \sum_{(s_h,a_h)\in\mathcal{C}_h} d_h^{\pi^\star}(s_h,a_h)\sqrt{\frac{\mathrm{Var}_{P_h(\cdot|s_h,a_h)}(V_{h+1}^\star(\cdot))}{nd_h^\mu(s_h,a_h)}}\right) + \widetilde{O}\left(\frac{H^3}{n\cdot\bar{d}_m}\right). \quad (7)$$

Compared to our Theorem 3.4, the additional sub-optimality bound due to differential privacy is $\widetilde{O}\left(\frac{SH^2E_\rho}{n\cdot\bar{d}_m}\right) = \widetilde{O}\left(\frac{SH^{\frac{5}{2}}}{n\cdot\bar{d}_m\sqrt{\rho}}\right) = \widetilde{O}\left(\frac{SH^{\frac{5}{2}}}{n\cdot\bar{d}_m\epsilon}\right)$.[7] In the most popular regime where the privacy budget $\rho$ or $\epsilon$ is a constant, the additional term due to differential privacy appears as a lower order term, hence becomes negligible as the sample complexity $n$ becomes large.

**Comparison to Hoeffding type pessimism.** We can simply revise our algorithm by using Hoeffding type pessimism, which replaces the pessimism in line 5 with $C_1 H \cdot \sqrt{\frac{\iota}{\widetilde{n}_{s_h,a_h} - E_\rho}} + \frac{C_2 SHE_\rho \cdot \iota}{\widetilde{n}_{s_h,a_h}}$. Then

---

[6] The non-private empirical estimate, defined as (15) in Appendix C.

[7] Here we apply the second part of Lemma 2.7 to achieve $(\epsilon,\delta)$-DP, the notation $\widetilde{O}$ also absorbs $\log\frac{1}{\delta}$ (only here $\delta$ denotes the privacy budget instead of failure probability).

with a similar proof schedule, we can arrive at a sub-optimality bound that with high probability,

$$0 \leq v^\star - v^{\widehat{\pi}} \leq \widetilde{O}\left(H \cdot \sum_{h=1}^{H} \sum_{(s_h, a_h) \in \mathcal{C}_h} d_h^{\pi^\star}(s_h, a_h) \sqrt{\frac{1}{n d_h^\mu(s_h, a_h)}}\right) + \widetilde{O}\left(\frac{SH^2 E_\rho}{n \cdot \bar{d}_m}\right). \quad (8)$$

Compared to our Theorem 3.4, our bound is tighter because we express the dominate term by the system quantities instead of explicit dependence on $H$ (and $\mathrm{Var}_{P_h(\cdot|s_h, a_h)}(V_{h+1}^\star(\cdot)) \leq H^2$). In addition, we highlight that according to Theorem G.1 in [Yin and Wang, 2021b], our main term nearly matches the non-private minimax lower bound. For more detailed discussions about our main term and how it subsumes other optimal learning bounds, we refer readers to [Yin and Wang, 2021b].

**Apply Laplace Mechanism to achieve pure DP.** To achieve Pure DP instead of $\rho$-zCDP, we can simply replace Gaussian Mechanism with Laplace Mechanism (defined as Definition E.19). Given privacy budget for Pure DP $\epsilon$, since the $\ell_1$ sensitivity of $\{n_{s_h, a_h}\} \cup \{n_{s_h, a_h, s_{h+1}}\}$ is $\Delta_1 = 4H$, we can add *independent* Laplace noises $\mathrm{Lap}(\frac{4H}{\epsilon})$ to each count to achieve $\epsilon$-DP due to Lemma E.20. Then by using $E_\epsilon = \widetilde{O}\left(\frac{H}{\epsilon}\right)$ instead of $E_\rho$ and keeping everything else ((3), (5) and Algorithm 1) the same, we can reach a similar result to Theorem 3.4 with the same proof schedule. The only difference is that here the additional learning bound is $\widetilde{O}\left(\frac{SH^3}{n \cdot \bar{d}_m \epsilon}\right)$, which still appears as a lower order term.

# 4 Results under linear MDP: DP-VAPVI(Algorithm 2)

In large MDPs, to address the computational issues, the technique of function approximation is widely applied, and linear MDP is a concrete model to study linear function approximations. Our second result applies to the linear MDP setting. Generally speaking, function approximation reduces the dimensionality of private releases comparing to the tabular MDPs. We begin with private counts.

**Private Model-based Components.** Given the two datasets $\mathcal{D}$ and $\mathcal{D}'$ (both from $\mu$) as in Algorithm 2, we can apply variance-aware pessimistic value iteration to learn a near optimal policy as in VAPVI [Yin et al., 2022]. To ensure differential privacy, we add *independent* Gaussian noises to the $5H$ statistics as in DP-VAPVI (Algorithm 2) below. Since there are $5H$ statistics, by the adaptive composition of zCDP (Lemma E.17), it suffices to keep each count $\rho_0$-zCDP, where $\rho_0 = \frac{\rho}{5H}$. In DP-VAPVI, we use $\phi_1, \phi_2, \phi_3, K_1, K_2$[8] to denote the noises we add. For all $\phi_i$, we directly apply Gaussian Mechanism. For $K_i$, in addition to the noise matrix $\frac{1}{\sqrt{2}}(Z + Z^\top)$, we also add $\frac{E}{2} I_d$ to ensure that all $K_i$ are positive definite with high probability (The detailed definition of $E, L$ can be found in Appendix A).

Below we will show the algorithmic design of DP-VAPVI (Algorithm 2). For the offline dataset, we divide it into two independent parts with equal length: $\mathcal{D} = \{(s_h^\tau, a_h^\tau, r_h^\tau, s_{h+1}^\tau)\}_{\tau \in [K]}^{h \in [H]}$ and $\mathcal{D}' = \{(\bar{s}_h^\tau, \bar{a}_h^\tau, \bar{r}_h^\tau, \bar{s}_{h+1}^\tau)\}_{\tau \in [K]}^{h \in [H]}$. One for estimating variance and the other for calculating $Q$-values.

**Estimating conditional variance.** The first part (line 4 to line 8) aims to estimate the conditional variance of $\widetilde{V}_{h+1}$ via the definition of variance: $[\mathrm{Var}_h \widetilde{V}_{h+1}](s, a) = [P_h(\widetilde{V}_{h+1})^2](s, a) - ([P_h \widetilde{V}_{h+1}](s, a))^2$. For the first term, by the definition of linear MDP, it holds that $\left[P_h \widetilde{V}_{h+1}^2\right](s, a) = \phi(s, a)^\top \int_{\mathcal{S}} \widetilde{V}_{h+1}^2(s') \, \mathrm{d}\nu_h(s') = \langle \phi, \int_{\mathcal{S}} \widetilde{V}_{h+1}^2(s') \, \mathrm{d}\nu_h(s') \rangle$. We can estimate $\beta_h = \int_{\mathcal{S}} \widetilde{V}_{h+1}^2(s') \, \mathrm{d}\nu_h(s')$ by applying ridge regression. Below is the output of ridge regression with raw statistics without noise:

$$\operatorname*{argmin}_{\beta \in \mathbb{R}^d} \sum_{k=1}^{K} \left[\left\langle \phi(\bar{s}_h^k, \bar{a}_h^k), \beta \right\rangle - \widetilde{V}_{h+1}^2\left(\bar{s}_{h+1}^k\right)\right]^2 + \lambda \|\beta\|_2^2 = \bar{\Sigma}_h^{-1} \sum_{k=1}^{K} \phi(\bar{s}_h^k, \bar{a}_h^k) \widetilde{V}_{h+1}^2\left(\bar{s}_{h+1}^k\right),$$

---

[8]We need to add noise to each of the $5H$ counts, therefore for $\phi_1$, we actually sample $H$ i.i.d samples $\phi_{1,h}$, $h = 1, \cdots, H$ from the distribution of $\phi_1$. Then we add $\phi_{1,h}$ to $\sum_{\tau=1}^{K} \phi(\bar{s}_h^\tau, \bar{a}_h^\tau) \cdot \widetilde{V}_{h+1}(\bar{s}_{h+1}^\tau)^2$, $\forall h \in [H]$. For simplicity, we use $\phi_1$ to represent all the $\phi_{1,h}$. The procedure applied to the other $4H$ statistics are similar.

---
**Algorithm 2** Differentially Private Variance-Aware Pessimistic Value Iteration (DP-VAPVI)
---
1: **Input:** Dataset $\mathcal{D} = \{(s_h^\tau, a_h^\tau, r_h^\tau, s_{h+1}^\tau)\}_{\tau,h=1}^{K,H}$ $\mathcal{D}' = \{(\bar{s}_h^\tau, \bar{a}_h^\tau, \bar{r}_h^\tau, \bar{s}_{h+1}^\tau)\}_{\tau,h=1}^{K,H}$. Budget for zCDP $\rho$. Failure probability $\delta$. Universal constant $C$.

2: **Initialization:** Set $\rho_0 \leftarrow \frac{\rho}{5H}$, $\widetilde{V}_{H+1}(\cdot) \leftarrow 0$. Sample $\phi_1 \sim \mathcal{N}\left(0, \frac{2H^4}{\rho_0}I_d\right)$, $\phi_2, \phi_3 \sim \mathcal{N}\left(0, \frac{2H^2}{\rho_0}I_d\right)$, $K_1, K_2 \leftarrow \frac{E}{2}I_d + \frac{1}{\sqrt{2}}(Z + Z^\top)$, where $Z_{i,j} \sim \mathcal{N}\left(0, \frac{1}{4\rho_0}\right)$ (i.i.d.), $E = \widetilde{O}\left(\sqrt{\frac{Hd}{\rho}}\right)$. Set $D \leftarrow \widetilde{O}\left(\frac{H^2L}{\kappa} + \frac{H^4E\sqrt{d}}{\kappa^{3/2}} + H^3\sqrt{d}\right)$.

3: **for** $h = H, H-1, \ldots, 1$ **do**

4:      Set $\widetilde{\Sigma}_h \leftarrow \sum_{\tau=1}^K \phi(\bar{s}_h^\tau, \bar{a}_h^\tau)\phi(\bar{s}_h^\tau, \bar{a}_h^\tau)^\top + \lambda I + K_1$

5:      Set $\widetilde{\beta}_h \leftarrow \widetilde{\Sigma}_h^{-1}[\sum_{\tau=1}^K \phi(\bar{s}_h^\tau, \bar{a}_h^\tau) \cdot \widetilde{V}_{h+1}(\bar{s}_{h+1}^\tau)^2 + \phi_1]$

6:      Set $\widetilde{\theta}_h \leftarrow \widetilde{\Sigma}_h^{-1}[\sum_{\tau=1}^K \phi(\bar{s}_h^\tau, \bar{a}_h^\tau) \cdot \widetilde{V}_{h+1}(\bar{s}_{h+1}^\tau) + \phi_2]$

7:      Set $\left[\widetilde{\mathrm{Var}}_h\widetilde{V}_{h+1}\right](\cdot,\cdot) \leftarrow \left\langle\phi(\cdot,\cdot), \widetilde{\beta}_h\right\rangle_{[0,(H-h+1)^2]} - \left[\left\langle\phi(\cdot,\cdot), \widetilde{\theta}_h\right\rangle_{[0,H-h+1]}\right]^2$

8:      Set $\widetilde{\sigma}_h(\cdot,\cdot)^2 \leftarrow \max\{1, \widetilde{\mathrm{Var}}_h\widetilde{V}_{h+1}(\cdot,\cdot)\}$

9:      Set $\widetilde{\Lambda}_h \leftarrow \sum_{\tau=1}^K \phi\left(s_h^\tau, a_h^\tau\right)\phi\left(s_h^\tau, a_h^\tau\right)^\top / \widetilde{\sigma}_h^2(s_h^\tau, a_h^\tau) + \lambda I + K_2$

10:      Set $\widetilde{w}_h \leftarrow \widetilde{\Lambda}_h^{-1}\left(\sum_{\tau=1}^K \phi\left(s_h^\tau, a_h^\tau\right)\cdot\left(r_h^\tau + \widetilde{V}_{h+1}\left(s_{h+1}^\tau\right)\right)/\widetilde{\sigma}_h^2(s_h^\tau, a_h^\tau) + \phi_3\right)$

11:      Set $\Gamma_h(\cdot,\cdot) \leftarrow C\sqrt{d} \cdot \left(\phi(\cdot,\cdot)^\top \widetilde{\Lambda}_h^{-1}\phi(\cdot,\cdot)\right)^{1/2} + \frac{D}{K}$

12:      Set $\bar{Q}_h(\cdot,\cdot) \leftarrow \phi(\cdot,\cdot)^\top\widetilde{w}_h - \Gamma_h(\cdot,\cdot)$

13:      Set $\widehat{Q}_h(\cdot,\cdot) \leftarrow \min\left\{\bar{Q}_h(\cdot,\cdot), H-h+1\right\}^+$

14:      Set $\widehat{\pi}_h(\cdot \mid \cdot) \leftarrow \mathrm{argmax}_{\pi_h}\left\langle\widehat{Q}_h(\cdot,\cdot), \pi_h(\cdot \mid \cdot)\right\rangle_{\mathcal{A}}$, $\widetilde{V}_h(\cdot) \leftarrow \max_{\pi_h}\left\langle\widehat{Q}_h(\cdot,\cdot), \pi_h(\cdot \mid \cdot)\right\rangle_{\mathcal{A}}$

15: **end for**

16: **Output:** $\{\widehat{\pi}_h\}_{h=1}^H$.

---

where definition of $\bar{\Sigma}_h$ can be found in Appendix A. Instead of using the raw statistics, we replace them with private ones with Gaussian noises as in line 5. The second term is estimated similarly in line 6. The final estimator is defined as in line 8: $\widetilde{\sigma}_h(\cdot,\cdot)^2 = \max\{1, \widetilde{\mathrm{Var}}_h\widetilde{V}_{h+1}(\cdot,\cdot)\}$.[9]

**Variance-weighted LSVI.** Instead of directly applying LSVI [Jin et al., 2021], we can solve the variance-weighted LSVI (line 10). The result of variance-weighted LSVI with non-private statistics is shown below:

$$\mathrm{argmin}_{w\in\mathbb{R}^d} \lambda\|w\|_2^2 + \sum_{k=1}^K \frac{\left[\langle\phi(s_h^k, a_h^k), w\rangle - r_h^k - \widetilde{V}_{h+1}(s_{h+1}^k)\right]^2}{\widetilde{\sigma}_h^2(s_h^k, a_h^k)} = \widehat{\Lambda}_h^{-1}\sum_{k=1}^K \frac{\phi\left(s_h^k, a_h^k\right)\cdot\left[r_h^k + \widetilde{V}_{h+1}\left(s_{h+1}^k\right)\right]}{\widetilde{\sigma}_h^2(s_h^k, a_h^k)},$$

where definition of $\widehat{\Lambda}_h$ can be found in Appendix A. For the sake of differential privacy, we use private statistics instead and derive the $\widetilde{w}_h$ as in line 10.

**Our private pessimism.** Notice that if we remove all the Gaussian noises we add, our DP-VAPVI (Algorithm 2) will degenerate to VAPVI [Yin et al., 2022]. We design a similar pessimistic penalty using private statistics (line 11), with additional $\frac{D}{K}$ accounting for the extra pessimism due to DP.

**Main theorem.** We state our main theorem about DP-VAPVI below, the proof sketch is deferred to Appendix D.1 and detailed proof is deferred to Appendix D due to space limit. Note that quantities $\mathcal{M}_i, L, E$ can be found in Appendix A and briefly, $L = \widetilde{O}(\sqrt{H^3d/\rho})$, $E = \widetilde{O}(\sqrt{Hd/\rho})$. For the sample complexity lower bound, within the practical regime where the privacy budget is not very small, $\max\{\mathcal{M}_i\}$ is dominated by $\max\{\widetilde{O}(H^{12}d^3/\kappa^5), \widetilde{O}(H^{14}d/\kappa^5)\}$, which also appears in the sample complexity lower bound of VAPVI [Yin et al., 2022]. The $\sigma_V^2(s,a)$ in Theorem 4.1 is defined as $\max\{1, \mathrm{Var}_{P_h}(V)(s,a)\}$ for any $V$.

**Theorem 4.1.** *DP-VAPVI (Algorithm 2) satisfies $\rho$-zCDP. Furthermore, let $K$ be the number of episodes. Under the condition that $K > \max\{\mathcal{M}_1, \mathcal{M}_2, \mathcal{M}_3, \mathcal{M}_4\}$ and $\sqrt{d} > \xi$, where $\xi :=$ $\sup_{V\in[0,H], s'\sim P_h(s,a), h\in[H]}\left|\frac{r_h+V(s')-(\mathcal{T}_hV)(s,a)}{\sigma_V(s,a)}\right|$, for any $0 < \lambda < \kappa$, with probability $1-\delta$, for*

---

[9]The $\max\{1,\cdot\}$ operator here is for technical reason only: we want a lower bound for each variance estimate.

*all policy $\pi$ simultaneously, the output $\widehat{\pi}$ of DP-VAPVI satisfies*

$$v^{\pi} - v^{\widehat{\pi}} \leq \widetilde{O}\left(\sqrt{d} \cdot \sum_{h=1}^{H} \mathbb{E}_{\pi}\left[\sqrt{\phi(\cdot,\cdot)^{\top}\Lambda_h^{-1}\phi(\cdot,\cdot)}\right]\right) + \frac{DH}{K}, \qquad (9)$$

*where* $\Lambda_h = \sum_{k=1}^{K} \frac{\phi(s_h^k,a_h^k)\cdot\phi(s_h^k,a_h^k)^{\top}}{\sigma_{\bar{V}_{h+1}(s_h^k,a_h^k)}^2} + \lambda I_d$, $D = \widetilde{O}\left(\frac{H^2 L}{\kappa} + \frac{H^4 E\sqrt{d}}{\kappa^{3/2}} + H^3\sqrt{d}\right)$ *and* $\widetilde{O}$ *hides constants and Polylog terms.*

*In particular, define* $\Lambda_h^{\star} = \sum_{k=1}^{K} \frac{\phi(s_h^k,a_h^k)\cdot\phi(s_h^k,a_h^k)^{\top}}{\sigma_{V_{h+1}^{\star}(s_h^k,a_h^k)}^2} + \lambda I_d$, *we have with probability* $1 - \delta$,

$$v^{\star} - v^{\widehat{\pi}} \leq \widetilde{O}\left(\sqrt{d} \cdot \sum_{h=1}^{H} \mathbb{E}_{\pi^{\star}}\left[\sqrt{\phi(\cdot,\cdot)^{\top}\Lambda_h^{\star-1}\phi(\cdot,\cdot)}\right]\right) + \frac{DH}{K}. \qquad (10)$$

**Comparison to non-private counterpart VAPVI [Yin et al., 2022].** Plugging in the definition of $L, E$ (Appendix A), under the meaningful case that the privacy budget is not very large, $DH$ is dominated by $\widetilde{O}\left(\frac{H^{\frac{11}{2}} d/\kappa^{\frac{3}{2}}}{\sqrt{\rho}}\right)$. According to Theorem 3.2 in [Yin et al., 2022], the sub-optimality bound of VAPVI is for sufficiently large $K$, with high probability, the output $\widehat{\pi}$ satisfies:

$$v^{\star} - v^{\widehat{\pi}} \leq \widetilde{O}\left(\sqrt{d} \cdot \sum_{h=1}^{H} \mathbb{E}_{\pi^{\star}}\left[\sqrt{\phi(\cdot,\cdot)^{\top}\Lambda_h^{\star-1}\phi(\cdot,\cdot)}\right]\right) + \frac{2H^4\sqrt{d}}{K}. \qquad (11)$$

Compared to our Theorem 4.1, the additional sub-optimality bound due to differential privacy is $\widetilde{O}\left(\frac{H^{\frac{11}{2}} d/\kappa^{\frac{3}{2}}}{\sqrt{\rho}\cdot K}\right) = \widetilde{O}\left(\frac{H^{\frac{11}{2}} d/\kappa^{\frac{3}{2}}}{\epsilon\cdot K}\right)$.[10] In the most popular regime where the privacy budget $\rho$ or $\epsilon$ is a constant, the additional term due to differential privacy also appears as a lower order term.

**Instance-dependent sub-optimality bound.** Similar to DP-APVI (Algorithm 1), our DP-VAPVI (Algorithm 2) also enjoys instance-dependent sub-optimality bound. First, the main term in (10) improves PEVI [Jin et al., 2021] over $O(\sqrt{d})$ on feature dependence. Also, our main term admits no explicit dependence on $H$, thus improves the sub-optimality bound of PEVI on horizon dependence. For more detailed discussions about our main term, we refer readers to [Yin et al., 2022].

**Private and safe policy improvement.** In addition to the sub-optimality bound (10), we have the so called oracle inequality (9). Therefore, the performance $v^{\widehat{\pi}}$ can be lower bounded by $\sup_{\pi}\left[v^{\pi} - \widetilde{O}\left(\sqrt{d} \cdot \sum_{h=1}^{H} \mathbb{E}_{\pi}\left[\sqrt{\phi(\cdot,\cdot)^{\top}\Lambda_h^{-1}\phi(\cdot,\cdot)}\right]\right) - \frac{DH}{K}\right]$. When choosing $\pi$ to be the optimal policy in the neighborhood of the behavior policy $\mu$, our DP-VAPVI (Algorithm 2) sheds light on safe policy improvement with differential privacy guarantee.

# 5 Tightness of our results

We believe our bounds for offline RL with DP is tight. To the best of our knowledge, APVI and VAPVI provide the tightest bound under tabular MDP and linear MDP, respectively. The suboptimality bounds of our algorithms match these two in the main term, with some lower order additional terms. The leading terms are known to match multiple information-theoretical lower bounds for offline RL simultaneously (this was illustrated in Yin and Wang [2021b], Yin et al. [2022]), for this reason our bound cannot be improved in general. For the lower order terms, the dependence on sample complexity $n$ and privacy budget $\epsilon$: $\widetilde{O}(\frac{1}{n\epsilon})$ is optimal since policy learning is a special case of ERM problems and such dependence is optimal in DP-ERM. In addition, we believe the dependence on other parameters $(H, S, A, d)$ in the lower order term is tight due to our special tricks as (3) and Lemma D.6.

---

[10] Here we apply the second part of Lemma 2.7 to achieve $(\epsilon, \delta)$-DP, the notation $\widetilde{O}$ also absorbs $\log\frac{1}{\delta}$ (only here $\delta$ denotes the privacy budget instead of failure probability).

# 6 Simulations

In this section, we carry out simulations to evaluate the performance of our DP-VAPVI (Algorithm 2), and compare it with its non-private counterpart VAPVI [Yin et al., 2022] and another pessimism-based algorithm PEVI [Jin et al., 2021] which does not have privacy guarantee.

**Experimental setting.** We evaluate DP-VAPVI (Algorithm 2) on a synthetic linear MDP example that originates from the linear MDP in [Min et al., 2021, Yin et al., 2022] but with some modifications.[11] For details of the linear MDP setting, please refer to Appendix F. The two MDP instances we use both have horizon $H = 20$. We compare different algorithms in figure 1(a), while in figure 1(b), we compare our DP-VAPVI with different privacy budgets. When doing empirical evaluation, we do not split the data for DP-VAPVI or VAPVI and for DP-VAPVI, we run the simulation for 5 times and take the average performance.

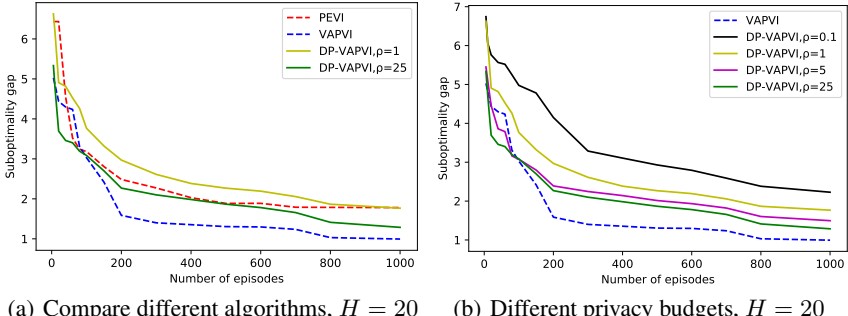

(a) Compare different algorithms, $H = 20$   (b) Different privacy budgets, $H = 20$

Figure 1: Comparison between performance of PEVI, VAPVI and DP-VAPVI (with different privacy budgets) under the linear MDP example described above. In each figure, y-axis represents sub-optimality gap $v^\star - v^{\widehat{\pi}}$ while x-axis denotes the number of episodes $K$. The horizons are fixed to be $H = 20$. The number of episodes takes value from 5 to 1000.

**Results and discussions.** From Figure 1, we can observe that DP-VAPVI (Algorithm 2) performs slightly worse than its non-private version VAPVI [Yin et al., 2022]. This is due to the fact that we add Gaussian noise to each count. However, as the size of dataset goes larger, the performance of DP-VAPVI will converge to that of VAPVI, which supports our theoretical conclusion that the cost of privacy only appears as lower order terms. For DP-VAPVI with larger privacy budget, the scale of noise will be smaller, thus the performance will be closer to VAPVI, as shown in figure 1(b). Furthermore, in most cases, DP-VAPVI still outperforms PEVI, which does not have privacy guarantee. This arises from our privitization of variance-aware LSVI instead of LSVI.

# 7 Conclusion and future works

In this work, we take the first steps towards the well-motivated task of designing private offline RL algorithms. We propose algorithms for both tabular MDPs and linear MDPs, and show that they enjoy instance-dependent sub-optimality bounds while guaranteeing differential privacy (either zCDP or pure DP). Our results highlight that the cost of privacy only appears as lower order terms, thus become negligible as the number of samples goes large.

Future extensions are numerous. We believe the technique in our algorithms (privitization of Bernstein-type pessimism and variance-aware LSVI) and the corresponding analysis can be used in online settings too to obtain tighter regret bounds for private algorithms. For the offline RL problems, we plan to consider more general function approximations and differentially private (deep) offline RL which will bridge the gap between theory and practice in offline RL applications. Many techniques we developed could be adapted to these more general settings.

---

[11]We keep the state space $\mathcal{S} = \{1, 2\}$, action space $\mathcal{A} = \{1, \cdots, 100\}$ and feature map of state-action pairs while we choose stochastic transition (instead of the original deterministic transition) and more complex reward.

## Acknowledgments

The research is partially supported by NSF Awards #2007117 and #2048091. The authors would like to thank Ming Yin for helpful discussions.

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

# A Notation List

## A.1 Notations for tabular MDP

| | |
|---|---|
| $E_\rho$ | $4\sqrt{\frac{H\log\frac{4HS^2A}{\delta}}{\rho}}$ |
| $n$ | The original counts of visitation |
| $n'$ | The noisy counts, as defined in (2) |
| $\widetilde{n}$ | Final choice of private counts, as defined in (3) |
| $\widetilde{P}$ | Private estimate of transition kernel, as defined in (5) |
| $\widehat{P}$ | Non-private estimate of transition kernel, as defined in (15) |
| $\iota$ | $\log\frac{HSA}{\delta}$ |
| $\rho$ | Budget for zCDP |
| $\delta$ | Failure probability |

## A.2 Notations for linear MDP

| | |
|---|---|
| $L$ | $2H\sqrt{\frac{5Hd\log(\frac{10Hd}{\delta})}{\rho}}$ |
| $E$ | $\sqrt{\frac{10Hd}{\rho}}\left(2+\left(\frac{\log(5c_1H/\delta)}{c_2d}\right)^{\frac{2}{3}}\right)$ |
| $D$ | $\widetilde{O}\left(\frac{H^2L}{\kappa}+\frac{H^4E\sqrt{d}}{\kappa^{3/2}}+H^3\sqrt{d}\right)$ |
| $\widehat{\Lambda}_h$ | $\sum_{k=1}^{K}\phi(s_h^k,a_h^k)\phi(s_h^k,a_h^k)^\top/\widetilde{\sigma}_h^2(s_h^k,a_h^k)+\lambda I_d$ |
| $\widetilde{\Lambda}_h$ | $\sum_{k=1}^{K}\phi(s_h^k,a_h^k)\phi(s_h^k,a_h^k)^\top/\widetilde{\sigma}_h^2(s_h^k,a_h^k)+\lambda I_d+K_2$ |
| $\widetilde{\Lambda}_h^p$ | $\mathbb{E}_{\mu,h}[\widetilde{\sigma}_h^{-2}(s,a)\phi(s,a)\phi(s,a)^\top]$ |
| $\Lambda_h$ | $\sum_{\tau=1}^{K}\phi(s_h^\tau,a_h^\tau)\phi(s_h^\tau,a_h^\tau)^\top/\sigma_{\widetilde{V}_{h+1}}^2(s_h^\tau,a_h^\tau)+\lambda I$ |
| $\Lambda_h^p$ | $\mathbb{E}_{\mu,h}[\sigma_{\widetilde{V}_{h+1}}^{-2}(s,a)\phi(s,a)\phi(s,a)^\top]$ |
| $\Lambda_h^\star$ | $\sum_{\tau=1}^{K}\phi(s_h^\tau,a_h^\tau)\phi(s_h^\tau,a_h^\tau)^\top/\sigma_{V_{h+1}^\star}^2(s_h^\tau,a_h^\tau)+\lambda I$ |
| $\bar{\Sigma}_h$ | $\sum_{\tau=1}^{K}\phi(\bar{s}_h^\tau,\bar{a}_h^\tau)\phi(\bar{s}_h^\tau,\bar{a}_h^\tau)^\top+\lambda I_d$ |
| $\widetilde{\Sigma}_h$ | $\sum_{\tau=1}^{K}\phi(\bar{s}_h^\tau,\bar{a}_h^\tau)\phi(\bar{s}_h^\tau,\bar{a}_h^\tau)^\top+\lambda I_d+K_1$ |
| $\Sigma_h^p$ | $\mathbb{E}_{\mu,h}\left[\phi(s,a)\phi(s,a)^\top\right]$ |
| $\kappa$ | $\min_h\lambda_{\min}(\Sigma_h^p)$ |
| $\sigma_V^2(s,a)$ | $\max\{1,\text{Var}_{P_h}(V)(s,a)\}$ for any $V$ |
| $\sigma_h^{\star 2}(s,a)$ | $\max\left\{1,\text{Var}_{P_h}V_{h+1}^\star(s,a)\right\}$ |
| $\widetilde{\sigma}_h^2(s,a)$ | $\max\{1,\widetilde{\text{Var}}_h\widetilde{V}_{h+1}(s,a)\}$ |
| $\mathcal{M}_1$ | $\max\{2\lambda,128\log(2dH/\delta),\frac{128H^4\log(2dH/\delta)}{\kappa^2},\frac{\sqrt{2}L}{\sqrt{d\kappa}}\}$ |
| $\mathcal{M}_2$ | $\max\{\widetilde{O}(H^{12}d^3/\kappa^5),\widetilde{O}(H^{14}d/\kappa^5)\}$ |
| $\mathcal{M}_3$ | $\max\left\{\frac{512H^4\log\left(\frac{2dH}{\delta}\right)}{\kappa^2},\frac{4\lambda H^2}{\kappa}\right\}$ |
| $\mathcal{M}_4$ | $\max\{\frac{H^2L^2}{d\kappa},\frac{H^6E^2}{\kappa^2},H^4\kappa\}$ |
| $\rho$ | Budget for zCDP |
| $\delta$ | Failure probability (*not* the $\delta$ of $(\epsilon,\delta)$-DP) |
| $\xi$ | $\sup_{V\in[0,H],\,s'\sim P_h(s,a),\,h\in[H]}\left\|\frac{r_h+V(s')-(\mathcal{T}_hV)(s,a)}{\sigma_V(s,a)}\right\|$ |

# B  Extended related work

**Online reinforcement learning under JDP or LDP.** For online RL, some recent works analyze this setting under *Joint Differential Privacy (JDP)*, which requires the RL agent to minimize regret while handling user's raw data privately. Under tabular MDP, Vietri et al. [2020] design PUCB by revising UBEV [Dann et al., 2017]. Private-UCB-VI [Chowdhury and Zhou, 2021] results from UCBVI (with bonus-1) [Azar et al., 2017]. However, both works privatize Hoeffding type bonus, which lead to sub-optimal regret bound. Under linear MDP, Private LSVI-UCB [Ngo et al., 2022] and Privacy-Preserving LSVI-UCB [Luyo et al., 2021] are private versions of LSVI-UCB [Jin et al., 2020b], while LinOpt-VI-Reg [Zhou, 2022] and Privacy-Preserving UCRL-VTR [Luyo et al., 2021] generalize UCRL-VTR [Ayoub et al., 2020]. However, these works are usually based on the LSVI technique [Jin et al., 2020b] (unweighted ridge regression), which does not ensure optimal regret bound.

In addition to JDP, another common privacy guarantee for online RL is *Local Differential Privacy (LDP)*, LDP is a stronger definition of DP since it requires that the user's data is protected before the RL agent has access to it. Under LDP, Garcelon et al. [2021] reach a regret lower bound and design LDP-OBI which has matching regret upper bound. The result is generalized by Liao et al. [2021] to linear mixture setting. Later, Luyo et al. [2021] provide an unified framework for analyzing JDP and LDP under linear setting.

**Some other differentially private learning algorithms.** There are some other works about differentially private online learning [Guha Thakurta and Smith, 2013, Agarwal and Singh, 2017, Hu et al., 2021] and various settings of bandit [Shariff and Sheffet, 2018, Gajane et al., 2018, Basu et al., 2019, Zheng et al., 2020, Chen et al., 2020, Tossou and Dimitrakakis, 2017]. For the reinforcement learning setting, Wang and Hegde [2019] propose privacy-preserving Q-learning to protect the reward information. Ono and Takahashi [2020] study the problem of distributed reinforcement learning under LDP. Lebensold et al. [2019] present an actor critic algorithm with differentially private critic. Cundy and Ermon [2020] tackle DP-RL under the policy gradient framework. Chowdhury et al. [2021] consider the adaptive control of differentially private linear quadratic (LQ) systems. Zhao et al. [2022] designed differentially private linear sketch algorirthms.

**Offline reinforcement learning under tabular MDP.** Under tabular MDP, there are several works achieving optimal sub-optimality/sample complexity bounds under different coverage assumptions. For the problem of off-policy evaluation (OPE), Yin and Wang [2020] uses Tabular-MIS estimator to achieve asymptotic efficiency. In addition, the idea of uniform OPE is used to achieve the optimal sample complexity $O(H^3/d_m\epsilon^2)$ [Yin et al., 2021] for non-stationary MDP and the optimal sample complexity $O(H^2/d_m\epsilon^2)$ [Yin and Wang, 2021a] for stationary MDP, where $d_m$ is the lower bound for state-action occupancy. Such uniform convergence idea also supports some works regarding online exploration [Jin et al., 2020a, Qiao et al., 2022, Xu et al., 2023]. For offline RL with single concentrability assumption, Xie et al. [2021b] arrive at the optimal sample complexity $O(H^3SC^\star/\epsilon^2)$. Recently, Yin and Wang [2021b] propose APVI which can lead to instance-dependent sub-optimality bound, which subsumes previous optimal results under several assumptions. Madhow et al. [2023] consider offline policy evaluation for adaptively collected datasets.

**Offline reinforcement learning under linear MDP.** Recently, many works focus on offline RL under linear representation. Jin et al. [2021] present PEVI which applies the idea of pessimistic value iteration (the idea originates from [Jin et al., 2020b]), and PEVI is provably efficient for offline RL under linear MDP. Yin et al. [2022] improve the sub-optimality bound in [Jin et al., 2021] by replacing LSVI by variance-weighted LSVI. Xie et al. [2021a] consider Bellman consistent pessimism for general function approximation, and their result improves the sample complexity in [Jin et al., 2021] by order $O(d)$ (shown in Theorem 3.2). However, there is no improvement on horizon dependence. Zanette et al. [2021] propose a new offline actor-critic algorithm that naturally incorporates the pessimism principle. Besides, Wang et al. [2021a], Zanette [2021] study the statistical hardness of offline RL with linear representations by presenting exponential lower bounds. When relaxing the offline setting to low adaptive RL, Gao et al. [2021], Wang et al. [2021b], Qiao and Wang [2023b], Qiao et al. [2023] designed algorithms with low adaptivity.

## C Proof of Theorem 3.4

### C.1 Proof sketch

Since the whole proof for privacy guarantee is not very complex, we present it in Section C.2 below and only sketch the proof for suboptimality bound.

First of all, we bound the scale of noises we add to show that the $\widetilde{n}$ derived from (3) are close to real visitation numbers. Therefore, denoting the non-private empirical transition kernel by $\widehat{P}$ (detailed definition in (15)), we can show that $\|\widetilde{P} - \widehat{P}\|_1$ and $|\sqrt{\mathrm{Var}_{\widetilde{P}}(V)} - \sqrt{\mathrm{Var}_{\widehat{P}}(V)}|$ are small.

Next, resulting from the conditional independence of $\widetilde{V}_{h+1}$ and $\widetilde{P}_h$, we apply Empirical Bernstein's inequality to get $|(\widetilde{P}_h - P_h)\widetilde{V}_{h+1}| \lesssim \sqrt{\mathrm{Var}_{\widetilde{P}}(\widetilde{V}_{h+1})/\widetilde{n}_{s_h,a_h}} + SHE_\rho/\widetilde{n}_{s_h,a_h}$. Together with our definition of private pessimism and the key lemma: extended value difference (Lemma E.7 and E.8), we can bound the suboptimality of our output policy $\widehat{\pi}$ by:

$$v^\star - v^{\widehat{\pi}} \lesssim \sum_{h=1}^{H} \sum_{(s_h,a_h)\in\mathcal{C}_h} d_h^{\pi^\star}(s_h,a_h)\sqrt{\frac{\mathrm{Var}_{\widetilde{P}_h(\cdot|s_h,a_h)}(\widetilde{V}_{h+1}(\cdot))}{\widetilde{n}_{s_h,a_h}}} + SHE_\rho/\widetilde{n}_{s_h,a_h}. \tag{12}$$

Finally, we further bound the above suboptimality via replacing private statistics by non-private ones. Specifically, we replace $\widetilde{n}$ by $n$, $\widetilde{P}$ by $P$ and $\widetilde{V}$ by $V^\star$. Due to (12), we have $\|\widetilde{V} - V^\star\|_\infty \lesssim \sqrt{\frac{1}{n\bar{d}_m}}$. Together with the upper bounds of $\|\widetilde{P} - \widehat{P}\|_1$ and $|\sqrt{\mathrm{Var}_{\widetilde{P}}(V)} - \sqrt{\mathrm{Var}_{\widehat{P}}(V)}|$, we have

$$\sqrt{\frac{\mathrm{Var}_{\widetilde{P}_h(\cdot|s_h,a_h)}(\widetilde{V}_{h+1}(\cdot))}{\widetilde{n}_{s_h,a_h}}} \lesssim \sqrt{\frac{\mathrm{Var}_{\widetilde{P}_h(\cdot|s_h,a_h)}(V_{h+1}^\star(\cdot))}{\widetilde{n}_{s_h,a_h}}} + \frac{1}{n\bar{d}_m}$$
$$\lesssim \sqrt{\frac{\mathrm{Var}_{\widehat{P}_h(\cdot|s_h,a_h)}(V_{h+1}^\star(\cdot))}{\widetilde{n}_{s_h,a_h}}} + \frac{1}{n\bar{d}_m} \lesssim \sqrt{\frac{\mathrm{Var}_{P_h(\cdot|s_h,a_h)}(V_{h+1}^\star(\cdot))}{\widetilde{n}_{s_h,a_h}}} + \frac{1}{n\bar{d}_m} \tag{13}$$
$$\lesssim \sqrt{\frac{\mathrm{Var}_{P_h(\cdot|s_h,a_h)}(V_{h+1}^\star(\cdot))}{n d_h^\mu(s_h,a_h)}} + \frac{1}{n\bar{d}_m}.$$

The final bound using non-private statistics results from (12) and (13).

### C.2 Proof of the privacy guarantee

The privacy guarantee of DP-APVI (Algorithm 1) is summarized by Lemma C.1 below.

**Lemma C.1** (Privacy analysis of DP-APVI (Algorithm 1)). *DP-APVI (Algorithm 1) satisfies $\rho$-zCDP.*

*Proof of Lemma C.1.* The $\ell_2$ sensitivity of $\{n_{s_h,a_h}\}$ is $\sqrt{2H}$. According to Lemma 2.7, the Gaussian Mechanism used on $\{n_{s_h,a_h}\}$ with $\sigma^2 = \frac{2H}{\rho}$ satisfies $\frac{\rho}{2}$-zCDP. Similarly, the Gaussian Mechanism used on $\{n_{s_h,a_h,s_{h+1}}\}$ with $\sigma^2 = \frac{2H}{\rho}$ also satisfies $\frac{\rho}{2}$-zCDP. Combining these two results, due to the composition of zCDP (Lemma E.16), the construction of $\{n'\}$ satisfies $\rho$-zCDP. Finally, DP-APVI satisfies $\rho$-zCDP because the output $\widehat{\pi}$ is post processing of $\{n'\}$. $\qquad\square$

### C.3 Proof of the sub-optimality bound

#### C.3.1 Utility analysis

First of all, the following Lemma C.2 gives a high probability bound for $|n' - n|$.

**Lemma C.2.** *Let $E_\rho = 2\sqrt{2}\sigma\sqrt{\log\frac{4HS^2A}{\delta}} = 4\sqrt{\frac{H\log\frac{4HS^2A}{\delta}}{\rho}}$, then with probability $1-\delta$, for all $s_h, a_h, s_{h+1}$, it holds that*

$$|n'_{s_h,a_h} - n_{s_h,a_h}| \le \frac{E_\rho}{2}, \ |n'_{s_h,a_h,s_{h+1}} - n_{s_h,a_h,s_{h+1}}| \le \frac{E_\rho}{2}. \tag{14}$$

*Proof of Lemma C.2.* The inequalities directly result from the concentration inequality of Gaussian distribution and a union bound. ☐

According to the utility analysis above, we have the following Lemma C.3 giving a high probability bound for $|\widetilde{n} - n|$.

**Lemma C.3.** *Under the high probability event in Lemma C.2, for all $s_h, a_h, s_{h+1}$, it holds that*

$$|\widetilde{n}_{s_h,a_h} - n_{s_h,a_h}| \leq E_\rho, \ |\widetilde{n}_{s_h,a_h,s_{h+1}} - n_{s_h,a_h,s_{h+1}}| \leq E_\rho.$$

*Proof of Lemma C.3.* When the event in Lemma C.2 holds, the original counts $\{n_{s_h,a_h,s'}\}_{s'\in\mathcal{S}}$ is a feasible solution to the optimization problem, which means that

$$\max_{s'} |\widetilde{n}_{s_h,a_h,s'} - n'_{s_h,a_h,s'}| \leq \max_{s'} |n_{s_h,a_h,s'} - n'_{s_h,a_h,s'}| \leq \frac{E_\rho}{2}.$$

Due to the second part of (14), it holds that for any $s_h, a_h, s_{h+1}$,

$$|\widetilde{n}_{s_h,a_h,s_{h+1}} - n_{s_h,a_h,s_{h+1}}| \leq |\widetilde{n}_{s_h,a_h,s_{h+1}} - n'_{s_h,a_h,s_{h+1}}| + |n'_{s_h,a_h,s_{h+1}} - n_{s_h,a_h,s_{h+1}}| \leq E_\rho.$$

For the second part, because of the constraints in the optimization problem, it holds that

$$|\widetilde{n}_{s_h,a_h} - n'_{s_h,a_h}| \leq \frac{E_\rho}{2}.$$

Due to the first part of (14), it holds that for any $s_h, a_h$,

$$|\widetilde{n}_{s_h,a_h} - n_{s_h,a_h}| \leq |\widetilde{n}_{s_h,a_h} - n'_{s_h,a_h}| + |n'_{s_h,a_h} - n_{s_h,a_h}| \leq E_\rho.$$

☐

Let the non-private empirical estimate be:

$$\widehat{P}_h(s'|s_h, a_h) = \frac{n_{s_h,a_h,s'}}{n_{s_h,a_h}}, \tag{15}$$

if $n_{s_h,a_h} > 0$ and $\widehat{P}_h(s'|s_h, a_h) = \frac{1}{S}$ otherwise. We will show that the private transition kernel $\widetilde{P}$ is close to $\widehat{P}$ by the Lemma C.4 and Lemma C.5 below.

**Lemma C.4.** *Under the high probability event of Lemma C.3, for $s_h, a_h$, if $\widetilde{n}_{s_h,a_h} \geq 3E_\rho$, it holds that*

$$\left\| \widetilde{P}_h(\cdot|s_h, a_h) - \widehat{P}_h(\cdot|s_h, a_h) \right\|_1 \leq \frac{5SE_\rho}{\widetilde{n}_{s_h,a_h}}. \tag{16}$$

*Proof of Lemma C.4.* If $\widetilde{n}_{s_h,a_h} \geq 3E_\rho$ and the conclusion in Lemma C.3 hold, we have

$$\left\| \widetilde{P}_h(\cdot|s_h, a_h) - \widehat{P}_h(\cdot|s_h, a_h) \right\|_1 \leq \sum_{s'\in\mathcal{S}} \left| \widetilde{P}_h(s'|s_h, a_h) - \widehat{P}_h(s'|s_h, a_h) \right|$$

$$\leq \sum_{s'\in\mathcal{S}} \left( \frac{\widetilde{n}_{s_h,a_h,s'} + E_\rho}{\widetilde{n}_{s_h,a_h} - E_\rho} - \frac{\widetilde{n}_{s_h,a_h,s'}}{\widetilde{n}_{s_h,a_h}} \right)$$

$$\leq \sum_{s'\in\mathcal{S}} \left[ \left( \frac{1}{\widetilde{n}_{s_h,a_h}} + \frac{2E_\rho}{\widetilde{n}_{s_h,a_h}^2} \right) (\widetilde{n}_{s_h,a_h,s'} + E_\rho) - \frac{\widetilde{n}_{s_h,a_h,s'}}{\widetilde{n}_{s_h,a_h}} \right] \tag{17}$$

$$\leq \frac{SE_\rho}{\widetilde{n}_{s_h,a_h}} + \frac{2E_\rho}{\widetilde{n}_{s_h,a_h}} + \frac{2SE_\rho^2}{\widetilde{n}_{s_h,a_h}^2}$$

$$\leq \frac{5SE_\rho}{\widetilde{n}_{s_h,a_h}}.$$

The second inequality is because $\frac{\widetilde{n}_{s_h,a_h,s'} - E_\rho}{\widetilde{n}_{s_h,a_h} + E_\rho} \leq \frac{n_{s_h,a_h,s'}}{n_{s_h,a_h}} \leq \frac{\widetilde{n}_{s_h,a_h,s'} + E_\rho}{\widetilde{n}_{s_h,a_h} - E_\rho}$ and $\frac{\widetilde{n}_{s_h,a_h,s'} + E_\rho}{\widetilde{n}_{s_h,a_h} - E_\rho} - \frac{\widetilde{n}_{s_h,a_h,s'}}{\widetilde{n}_{s_h,a_h}} \geq \frac{\widetilde{n}_{s_h,a_h,s'}}{\widetilde{n}_{s_h,a_h}} - \frac{\widetilde{n}_{s_h,a_h,s'} - E_\rho}{\widetilde{n}_{s_h,a_h} + E_\rho}$. The third inequality is because of Lemma E.6. The last inequality is because $\widetilde{n}_{s_h,a_h} \geq 3E_\rho$. ☐

**Lemma C.5.** *Let $V \in \mathbb{R}^S$ be any function with $\|V\|_\infty \leq H$, under the high probability event of Lemma C.3, for $s_h, a_h$, if $\widetilde{n}_{s_h,a_h} \geq 3E_\rho$, it holds that*

$$\left| \sqrt{\mathrm{Var}_{\widehat{P}_h(\cdot|s_h,a_h)}(V)} - \sqrt{\mathrm{Var}_{\widetilde{P}_h(\cdot|s_h,a_h)}(V)} \right| \leq 4H\sqrt{\frac{SE_\rho}{\widetilde{n}_{s_h,a_h}}}. \tag{18}$$

*Proof of Lemma C.5.* For $s_h, a_h$ such that $\widetilde{n}_{s_h,a_h} \geq 3E_\rho$, we use $\widetilde{P}(\cdot)$ and $\widehat{P}(\cdot)$ instead of $\widetilde{P}_h(\cdot|s_h,a_h)$ and $\widehat{P}_h(\cdot|s_h,a_h)$ for simplicity. Because of Lemma C.4, we have

$$\left\| \widetilde{P}(\cdot) - \widehat{P}(\cdot) \right\|_1 \leq \frac{5SE_\rho}{\widetilde{n}_{s_h,a_h}}.$$

Therefore, it holds that

$$\left| \sqrt{\mathrm{Var}_{\widehat{P}(\cdot)}(V)} - \sqrt{\mathrm{Var}_{\widetilde{P}(\cdot)}(V)} \right| \leq \sqrt{\left| \mathrm{Var}_{\widehat{P}(\cdot)}(V) - \mathrm{Var}_{\widetilde{P}(\cdot)}(V) \right|}$$

$$\leq \sqrt{\sum_{s' \in \mathcal{S}} \left| \widehat{P}(s') - \widetilde{P}(s') \right| V(s')^2 + \left| \sum_{s' \in \mathcal{S}} \left[ \widehat{P}(s') + \widetilde{P}(s') \right] V(s') \right| \cdot \sum_{s' \in \mathcal{S}} \left| \widehat{P}(s') - \widetilde{P}(s') \right| V(s')}$$

$$\leq \sqrt{H^2 \left\| \widetilde{P}(\cdot) - \widehat{P}(\cdot) \right\|_1 + 2H^2 \left\| \widetilde{P}(\cdot) - \widehat{P}(\cdot) \right\|_1}$$

$$\leq 4H\sqrt{\frac{SE_\rho}{\widetilde{n}_{s_h,a_h}}}.$$

$$\tag{19}$$

The second inequality is due to the definition of variance. $\qquad \square$

### C.3.2 Validity of our pessimistic penalty

Now we are ready to present the key lemma (Lemma C.6) below to justify our use of $\Gamma$ as the pessimistic penalty.

**Lemma C.6.** *Under the high probability event of Lemma C.3, with probability $1 - \delta$, for any $s_h, a_h$, if $\widetilde{n}_{s_h,a_h} \geq 3E_\rho$ (which implies $n_{s_h,a_h} > 0$), it holds that*

$$\left| (\widetilde{P}_h - P_h) \cdot \widetilde{V}_{h+1}(s_h, a_h) \right| \leq \sqrt{\frac{2\mathrm{Var}_{\widetilde{P}_h(\cdot|s_h,a_h)}(\widetilde{V}_{h+1}(\cdot)) \cdot \iota}{\widetilde{n}_{s_h,a_h} - E_\rho}} + \frac{16SHE_\rho \cdot \iota}{\widetilde{n}_{s_h,a_h}}, \tag{20}$$

*where $\widetilde{V}$ is the private version of estimated V function, which appears in Algorithm 1 and $\iota = \log(HSA/\delta)$.*

*Proof of Lemma C.6.*

$$\left| (\widetilde{P}_h - P_h) \cdot \widetilde{V}_{h+1}(s_h, a_h) \right| \leq \left| (\widetilde{P}_h - \widehat{P}_h) \cdot \widetilde{V}_{h+1}(s_h, a_h) \right| + \left| (\widehat{P}_h - P_h) \cdot \widetilde{V}_{h+1}(s_h, a_h) \right|$$

$$\leq H \left\| \widetilde{P}_h(\cdot|s_h,a_h) - \widehat{P}_h(\cdot|s_h,a_h) \right\|_1 + \left| (\widehat{P}_h - P_h) \cdot \widetilde{V}_{h+1}(s_h, a_h) \right| \tag{21}$$

$$\leq \frac{5SHE_\rho}{\widetilde{n}_{s_h,a_h}} + \left| (\widehat{P}_h - P_h) \cdot \widetilde{V}_{h+1}(s_h, a_h) \right|,$$

where the third inequality is due to Lemma C.4.

Next, recall $\widehat{\pi}_{h+1}$ in Algorithm 1 is computed backwardly therefore only depends on sample tuple from time $h + 1$ to $H$. As a result, $\widetilde{V}_{h+1} = \langle \overline{Q}_{h+1}, \widehat{\pi}_{h+1} \rangle$ also only depends on the sample tuple from time $h + 1$ to $H$ and some Gaussian noise that is independent to the offline dataset. On the other side, by the definition, $\widehat{P}_h$ only depends on the sample tuples from time $h$ to $h + 1$. Therefore $\widetilde{V}_{h+1}$ and $\widehat{P}_h$ are *Conditionally* independent (This trick is also used in [Yin et al., 2021] and [Yin and Wang,

2021b]), by Empirical Bernstein's inequality (Lemma E.4) and a union bound, with probability $1 - \delta$, for all $s_h, a_h$ such that $\widetilde{n}_{s_h,a_h} \geq 3E_\rho$,

$$\left| (\widehat{P}_h - P_h) \cdot \widetilde{V}_{h+1}(s_h, a_h) \right| \leq \sqrt{\frac{2\mathrm{Var}_{\widehat{P}_h(\cdot|s_h,a_h)}(\widetilde{V}_{h+1}(\cdot)) \cdot \iota}{n_{s_h,a_h}}} + \frac{7H \cdot \iota}{3n_{s_h,a_h}}. \tag{22}$$

Therefore, we have

$$\left| (\widetilde{P}_h - P_h) \cdot \widetilde{V}_{h+1}(s_h, a_h) \right| \leq \sqrt{\frac{2\mathrm{Var}_{\widehat{P}_h(\cdot|s_h,a_h)}(\widetilde{V}_{h+1}(\cdot)) \cdot \iota}{n_{s_h,a_h}}} + \frac{7H \cdot \iota}{3n_{s_h,a_h}} + \frac{5SHE_\rho}{\widetilde{n}_{s_h,a_h}}$$

$$\leq \sqrt{\frac{2\mathrm{Var}_{\widehat{P}_h(\cdot|s_h,a_h)}(\widetilde{V}_{h+1}(\cdot)) \cdot \iota}{n_{s_h,a_h}}} + \frac{9SHE_\rho \cdot \iota}{\widetilde{n}_{s_h,a_h}}$$

$$\leq \frac{9SHE_\rho \cdot \iota}{\widetilde{n}_{s_h,a_h}} + \sqrt{\frac{2\mathrm{Var}_{\widetilde{P}_h(\cdot|s_h,a_h)}(\widetilde{V}_{h+1}(\cdot)) \cdot \iota}{n_{s_h,a_h}}} + 4\sqrt{2}H\sqrt{\frac{SE_\rho \cdot \iota}{\widetilde{n}_{s_h,a_h} \cdot n_{s_h,a_h}}} \tag{23}$$

$$\leq \sqrt{\frac{2\mathrm{Var}_{\widetilde{P}_h(\cdot|s_h,a_h)}(\widetilde{V}_{h+1}(\cdot)) \cdot \iota}{n_{s_h,a_h}}} + \frac{16SHE_\rho \cdot \iota}{\widetilde{n}_{s_h,a_h}}$$

$$\leq \sqrt{\frac{2\mathrm{Var}_{\widetilde{P}_h(\cdot|s_h,a_h)}(\widetilde{V}_{h+1}(\cdot)) \cdot \iota}{\widetilde{n}_{s_h,a_h} - E_\rho}} + \frac{16SHE_\rho \cdot \iota}{\widetilde{n}_{s_h,a_h}}.$$

The second and forth inequality is because when $\widetilde{n}_{s_h,a_h} \geq 3E_\rho$, $n_{s_h,a_h} \geq \frac{2\widetilde{n}_{s_h,a_h}}{3}$. Specifically, these two inequalities are also because usually we only care about the case when $SE_\rho \geq 1$, which is equivalent to $\rho$ being not very large. The third inequality is due to Lemma C.5. The last inequality is due to Lemma C.3. $\qquad\square$

Note that the previous Lemmas rely on the condition that $\widetilde{n}$ is not very small ($\widetilde{n}_{s_h,a_h} \geq 3E_\rho$). Below we state the Multiplicative Chernoff bound (Lemma C.7 and Remark C.8) to show that under our condition in Theorem 3.4, for $(s_h, a_h) \in \mathcal{C}_h$, $\widetilde{n}_{s_h,a_h}$ will be larger than $3E_\rho$ with high probability.

**Lemma C.7** (Lemma B.1 in [Yin and Wang, 2021b]). *For any $0 < \delta < 1$, there exists an absolute constant $c_1$ such that when total episode $n > c_1 \cdot 1/\bar{d}_m \cdot \log(HSA/\delta)$, then with probability $1 - \delta$, $\forall h \in [H]$*

$$n_{s_h,a_h} \geq n \cdot d_h^\mu(s_h, a_h)/2, \quad \forall (s_h, a_h) \in \mathcal{C}_h.$$

*Furthermore, we denote*

$$\mathcal{E} := \{n_{s_h,a_h} \geq n \cdot d_h^\mu(s_h, a_h)/2, \ \forall (s_h, a_h) \in \mathcal{C}_h, \ h \in [H].\} \tag{24}$$

*then equivalently $P(\mathcal{E}) > 1 - \delta$.*

*In addition, we denote*

$$\mathcal{E}' := \{n_{s_h,a_h} \leq \frac{3}{2}n \cdot d_h^\mu(s_h, a_h), \ \forall (s_h, a_h) \in \mathcal{C}_h, \ h \in [H].\} \tag{25}$$

*then similarly $P(\mathcal{E}') > 1 - \delta$.*

**Remark C.8.** *According to Lemma C.7, for any failure probability $\delta$, there exists some constant $c_1 > 0$ such that when $n \geq \frac{c_1 E_\rho \cdot \iota}{\bar{d}_m}$, with probability $1 - \delta$, for all $(s_h, a_h) \in \mathcal{C}_h$, $n_{s_h,a_h} \geq 4E_\rho$. Therefore, under the condition of Theorem 3.4 and the high probability events in Lemma C.3 and Lemma C.7, it holds that for all $(s_h, a_h) \in \mathcal{C}_h$, $\widetilde{n}_{s_h,a_h} \geq 3E_\rho$ while for all $(s_h, a_h) \notin \mathcal{C}_h$, $\widetilde{n}_{s_h,a_h} \leq E_\rho$.*

**Lemma C.9.** *Define $(\mathcal{T}_h V)(\cdot, \cdot) := r_h(\cdot, \cdot) + (P_h V)(\cdot, \cdot)$ for any $V \in \mathbb{R}^S$. Note $\widehat{\pi}, \overline{Q}_h, \widetilde{V}_h$ are defined in Algorithm 1 and denote $\xi_h(s, a) = (\mathcal{T}_h \widetilde{V}_{h+1})(s, a) - \overline{Q}_h(s, a)$. Then it holds that*

$$V_1^{\pi^\star}(s) - V_1^{\widehat{\pi}}(s) \leq \sum_{h=1}^{H} \mathbb{E}_{\pi^\star}\left[\xi_h(s_h, a_h) \mid s_1 = s\right] - \sum_{h=1}^{H} \mathbb{E}_{\widehat{\pi}}\left[\xi_h(s_h, a_h) \mid s_1 = s\right]. \tag{26}$$

*Furthermore, (26) holds for all $V_h^{\pi^\star}(s) - V_h^{\widehat{\pi}}(s)$.*

*Proof of Lemma C.9.* Lemma C.9 is a direct corollary of Lemma E.8 with $\pi = \pi^\star$, $\widehat{Q}_h = \overline{Q}_h$, $\widehat{V}_h = \widetilde{V}_h$ and $\widehat{\pi} = \widehat{\pi}$ in Algorithm 1, we can obtain this result since by the definition of $\widehat{\pi}$ in Algorithm 1, $\langle \overline{Q}_h(s_h, \cdot), \pi_h(\cdot|s_h) - \widehat{\pi}_h(\cdot|s_h) \rangle \leq 0$. The proof for $V_h^{\pi^\star}(s) - V_h^{\widehat{\pi}}(s)$ is identical. $\qquad\square$

Next we prove the asymmetric bound for $\xi_h$, which is the key to the proof.

**Lemma C.10** (Private version of Lemma D.6 in [Yin and Wang, 2021b]). *Denote* $\xi_h(s,a) = (\mathcal{T}_h \widetilde{V}_{h+1})(s,a) - \overline{Q}_h(s,a)$, *where* $\widetilde{V}_{h+1}$ *and* $\overline{Q}_h$ *are the quantities in Algorithm 1 and* $\mathcal{T}_h(V) := r_h + P_h \cdot V$ *for any* $V \in \mathbb{R}^S$. *Then under the high probability events in Lemma C.3 and Lemma C.6, for any* $h, s_h, a_h$ *such that* $\widetilde{n}_{s_h,a_h} > 3E_\rho$, *we have*

$$
\begin{aligned}
0 \leq \xi_h(s_h, a_h) &= (\mathcal{T}_h \widetilde{V}_{h+1})(s_h, a_h) - \overline{Q}_h(s_h, a_h) \\
&\leq 2\sqrt{\frac{2\mathrm{Var}_{\widetilde{P}_h(\cdot|s_h,a_h)}(\widetilde{V}_{h+1}(\cdot)) \cdot \iota}{\widetilde{n}_{s_h,a_h} - E_\rho}} + \frac{32SHE_\rho \cdot \iota}{\widetilde{n}_{s_h,a_h}},
\end{aligned}
$$

*where* $\iota = \log(HSA/\delta)$.

*Proof of Lemma C.10.* **The first inequality:** We first prove $\xi_h(s_h, a_h) \geq 0$ for all $(s_h, a_h)$, such that $\widetilde{n}_{s_h,a_h} \geq 3E_\rho$.

Indeed, if $\widehat{Q}_h^p(s_h, a_h) < 0$, then $\overline{Q}_h(s_h, a_h) = 0$. In this case, $\xi_h(s_h, a_h) = (\mathcal{T}_h \widetilde{V}_{h+1})(s_h, a_h) \geq 0$ (note $\widetilde{V}_h \geq 0$ by the definition). If $\widehat{Q}_h^p(s_h, a_h) \geq 0$, then by definition $\overline{Q}_h(s_h, a_h) = \min\{\widehat{Q}_h^p(s_h, a_h), H - h + 1\}^+ \leq \widehat{Q}_h^p(s_h, a_h)$ and this implies

$$
\begin{aligned}
\xi_h(s_h, a_h) &\geq (\mathcal{T}_h \widetilde{V}_{h+1})(s_h, a_h) - \widehat{Q}_h^p(s_h, a_h) \\
&= (P_h - \widetilde{P}_h) \cdot \widetilde{V}_{h+1}(s_h, a_h) + \Gamma_h(s_h, a_h) \\
&\geq -\sqrt{\frac{2\mathrm{Var}_{\widetilde{P}_h(\cdot|s_h,a_h)}(\widetilde{V}_{h+1}(\cdot)) \cdot \iota}{\widetilde{n}_{s_h,a_h} - E_\rho}} - \frac{16SHE_\rho \cdot \iota}{\widetilde{n}_{s_h,a_h}} + \Gamma_h(s_h, a_h) = 0,
\end{aligned}
$$

where the second inequality uses Lemma C.6, and the last equation uses Line 5 of Algorithm 1.

**The second inequality:** Then we prove $\xi_h(s_h, a_h) \leq 2\sqrt{\frac{2\mathrm{Var}_{\widetilde{P}_h(\cdot|s_h,a_h)}(\widetilde{V}_{h+1}(\cdot)) \cdot \iota}{\widetilde{n}_{s_h,a_h} - E_\rho}} + \frac{32SHE_\rho \cdot \iota}{\widetilde{n}_{s_h,a_h}}$ for all $(s_h, a_h)$ such that $\widetilde{n}_{s_h,a_h} \geq 3E_\rho$.

First, since by construction $\widetilde{V}_h \leq H - h + 1$ for all $h \in [H]$, this implies

$$
\widehat{Q}_h^p = \widetilde{Q}_h - \Gamma_h \leq \widetilde{Q}_h = r_h + (\widetilde{P}_h \cdot \widetilde{V}_{h+1}) \leq 1 + (H - h) = H - h + 1
$$

which is because $r_h \leq 1$ and $\widetilde{P}_h$ is a probability distribution. Therefore, we have the equivalent definition

$$
\overline{Q}_h := \min\{\widehat{Q}_h^p, H - h + 1\}^+ = \max\{\widehat{Q}_h^p, 0\} \geq \widehat{Q}_h^p.
$$

Then it holds that

$$
\begin{aligned}
\xi_h(s_h, a_h) &= (\mathcal{T}_h \widetilde{V}_{h+1})(s_h, a_h) - \overline{Q}_h(s_h, a_h) \leq (\mathcal{T}_h \widetilde{V}_{h+1})(s_h, a_h) - \widehat{Q}_h^p(s_h, a_h) \\
&= (\mathcal{T}_h \widetilde{V}_{h+1})(s_h, a_h) - \widetilde{Q}_h(s_h, a_h) + \Gamma_h(s_h, a_h) \\
&= (P_h - \widetilde{P}_h) \cdot \widetilde{V}_{h+1}(s_h, a_h) + \Gamma_h(s_h, a_h) \\
&\leq \sqrt{\frac{2\mathrm{Var}_{\widetilde{P}_h(\cdot|s_h,a_h)}(\widetilde{V}_{h+1}(\cdot)) \cdot \iota}{\widetilde{n}_{s_h,a_h} - E_\rho}} + \frac{16SHE_\rho \cdot \iota}{\widetilde{n}_{s_h,a_h}} + \Gamma_h(s_h, a_h) \\
&= 2\sqrt{\frac{2\mathrm{Var}_{\widetilde{P}_h(\cdot|s_h,a_h)}(\widetilde{V}_{h+1}(\cdot)) \cdot \iota}{\widetilde{n}_{s_h,a_h} - E_\rho}} + \frac{32SHE_\rho \cdot \iota}{\widetilde{n}_{s_h,a_h}}.
\end{aligned}
$$

The proof is complete by combining the two parts. $\qquad\square$

### C.3.3 Reduction to augmented absorbing MDP

Before we prove the theorem, we need to construct an augmented absorbing MDP to bridge $\widetilde{V}$ and $V^\star$. According to Line 5 in Algorithm 1, the locations with $\widetilde{n}_{s_h,a_h} \leq E_\rho$ is heavily penalized with penalty of order $\widetilde{O}(H)$. Therefore we can prove that under the high probability event in Remark C.8, $d_h^{\widehat{\pi}}(s_h, a_h) > 0$ only if $d_h^\mu(s_h, a_h) > 0$ by induction, where $\widehat{\pi}$ is the output of Algorithm 1. The conclusion holds for $h = 1$. Assume it holds for some $h > 1$ that $d_h^{\widehat{\pi}}(s_h, a_h) > 0$ only if $d_h^\mu(s_h, a_h) > 0$, then for any $s_{h+1} \in \mathcal{S}$ such that $d_{h+1}^{\widehat{\pi}}(s_{h+1}) > 0$, it holds that $d_{h+1}^\mu(s_{h+1}) > 0$, which leads to the conclusion that $d_{h+1}^{\widehat{\pi}}(s_{h+1}, a_{h+1}) > 0$ only if $d_{h+1}^\mu(s_{h+1}, a_{h+1}) > 0$. To summarize, we have

$$d_h^{\pi_0}(s_h, a_h) > 0 \text{ only if } d_h^\mu(s_h, a_h) > 0, \ \pi_0 \in \{\pi^\star, \widehat{\pi}\}. \tag{27}$$

Let us define $M^\dagger$ by adding one absorbing state $s_h^\dagger$ for all $h \in \{2, \ldots, H\}$, therefore the augmented state space $\mathcal{S}^\dagger = \mathcal{S} \cup \{s_h^\dagger\}$ and the transition and reward is defined as follows: (recall $\mathcal{C}_h := \{(s_h, a_h) : d_h^\mu(s_h, a_h) > 0\}$)

$$P_h^\dagger(\cdot \mid s_h, a_h) = \begin{cases} P_h(\cdot \mid s_h, a_h) & s_h, a_h \in \mathcal{C}_h, \\ \delta_{s_{h+1}^\dagger} & s_h = s_h^\dagger \text{ or } s_h, a_h \notin \mathcal{C}_h, \end{cases} \quad r_h^\dagger(s_h, a_h) = \begin{cases} r_h(s_h, a_h) & s_h, a_h \in \mathcal{C}_h \\ 0 & s_h = s_h^\dagger \text{ or } s_h, a_h \notin \mathcal{C}_h \end{cases}$$

and we further define for any $\pi$,

$$V_h^{\dagger\pi}(s) = \mathbb{E}_\pi^\dagger \left[ \sum_{t=h}^H r_t^\dagger \,\middle|\, s_h = s \right], v^{\dagger\pi} = \mathbb{E}_\pi^\dagger \left[ \sum_{t=1}^H r_t^\dagger \right] \ \forall h \in [H], \tag{28}$$

where $\mathbb{E}^\dagger$ means taking expectation under the absorbing MDP $M^\dagger$.

Note that because $\pi^\star$ and $\widehat{\pi}$ are fully covered by $\mu$ (27), it holds that

$$v^{\dagger\pi^\star} = v^{\pi^\star}, \ v^{\dagger\widehat{\pi}} = v^{\widehat{\pi}}. \tag{29}$$

Define $(\mathcal{T}_h^\dagger V)(\cdot, \cdot) := r_h^\dagger(\cdot, \cdot) + (P_h^\dagger V)(\cdot, \cdot)$ for any $V \in \mathbb{R}^{S+1}$. Note $\widehat{\pi}$, $\overline{Q}_h$, $\widetilde{V}_h$ are defined in Algorithm 1 (we extend the definition by letting $\widetilde{V}_h(s_h^\dagger) = 0$ and $\overline{Q}_h(s_h^\dagger, \cdot) = 0$) and denote $\xi_h^\dagger(s, a) = (\mathcal{T}_h^\dagger \widetilde{V}_{h+1})(s, a) - \overline{Q}_h(s, a)$. Using identical proof to Lemma C.9, we have

$$V_1^{\dagger\pi^\star}(s) - V_1^{\dagger\widehat{\pi}}(s) \leq \sum_{h=1}^H \mathbb{E}_{\pi^\star}^\dagger \left[ \xi_h^\dagger(s_h, a_h) \mid s_1 = s \right] - \sum_{h=1}^H \mathbb{E}_{\widehat{\pi}}^\dagger \left[ \xi_h^\dagger(s_h, a_h) \mid s_1 = s \right], \tag{30}$$

where $V_1^{\dagger\pi}$ is defined in (28). Furthermore, (30) holds for all $V_h^{\dagger\pi^\star}(s) - V_h^{\dagger\widehat{\pi}}(s)$.

### C.3.4 Finalize our result with non-private statistics

For those $(s_h, a_h) \in \mathcal{C}_h$, $\xi_h^\dagger(s_h, a_h) = r_h(s_h, a_h) + P_h \widetilde{V}_{h+1}(s_h, a_h) - \overline{Q}_h(s_h, a_h) = \xi_h(s_h, a_h)$. For those $(s_h, a_h) \notin \mathcal{C}_h$ or $s_h = s_h^\dagger$, we have $\xi_h^\dagger(s_h, a_h) = 0$.

Therefore, by (30) and Lemma C.10, under the high probability events in Lemma C.3, Lemma C.6 and Lemma C.7, we have for all $t \in [H]$, $s \in \mathcal{S}$ ($\mathcal{S}$ does not include the absorbing state $s_t^\dagger$),

$$V_t^{\dagger \pi^\star}(s) - V_t^{\dagger \widehat{\pi}}(s) \le \sum_{h=t}^{H} \mathbb{E}_{\pi^\star}^\dagger \left[ \xi_h^\dagger(s_h, a_h) \mid s_t = s \right] - \sum_{h=t}^{H} \mathbb{E}_{\widehat{\pi}}^\dagger \left[ \xi_h^\dagger(s_h, a_h) \mid s_t = s \right]$$

$$\le \sum_{h=t}^{H} \mathbb{E}_{\pi^\star}^\dagger \left[ \xi_h^\dagger(s_h, a_h) \mid s_t = s \right] - 0$$

$$\le \sum_{h=t}^{H} \mathbb{E}_{\pi^\star}^\dagger \left[ 2\sqrt{\frac{2\mathrm{Var}_{\widetilde{P}_h(\cdot|s_h,a_h)}(\widetilde{V}_{h+1}(\cdot)) \cdot \iota}{\widetilde{n}_{s_h,a_h} - E_\rho}} + \frac{32SHE_\rho \cdot \iota}{\widetilde{n}_{s_h,a_h}} \mid s_t = s \right] \cdot \mathbb{1}\left((s_h, a_h) \in \mathcal{C}_h\right)$$

$$\le \sum_{h=t}^{H} \mathbb{E}_{\pi^\star}^\dagger \left[ 2\sqrt{\frac{2\mathrm{Var}_{\widetilde{P}_h(\cdot|s_h,a_h)}(\widetilde{V}_{h+1}(\cdot)) \cdot \iota}{n_{s_h,a_h} - 2E_\rho}} + \frac{32SHE_\rho \cdot \iota}{n_{s_h,a_h} - E_\rho} \mid s_t = s \right] \cdot \mathbb{1}\left((s_h, a_h) \in \mathcal{C}_h\right) \qquad (31)$$

$$\le \sum_{h=t}^{H} \mathbb{E}_{\pi^\star}^\dagger \left[ 4\sqrt{\frac{\mathrm{Var}_{\widetilde{P}_h(\cdot|s_h,a_h)}(\widetilde{V}_{h+1}(\cdot)) \cdot \iota}{n_{s_h,a_h}}} + \frac{128SHE_\rho \cdot \iota}{3n_{s_h,a_h}} \mid s_t = s \right] \cdot \mathbb{1}\left((s_h, a_h) \in \mathcal{C}_h\right)$$

$$\le \sum_{h=t}^{H} \mathbb{E}_{\pi^\star}^\dagger \left[ 4\sqrt{\frac{2\mathrm{Var}_{\widetilde{P}_h(\cdot|s_h,a_h)}(\widetilde{V}_{h+1}(\cdot)) \cdot \iota}{nd_h^\mu(s_h, a_h)}} + \frac{256SHE_\rho \cdot \iota}{3nd_h^\mu(s_h, a_h)} \mid s_t = s \right] \cdot \mathbb{1}\left((s_h, a_h) \in \mathcal{C}_h\right)$$

The second and third inequality are because of Lemma C.10, Remark C.8 and the the fact that either $\xi^\dagger = 0$ or $\xi^\dagger = \xi$ while $(s_h, a_h) \in \mathcal{C}_h$. The forth inequality is due to Lemma C.3. The fifth inequality is because of Remark C.8. The last inequality is by Lemma C.7.

Below we present a crude bound of $\left| V_t^{\dagger \pi^\star}(s) - \widetilde{V}_t(s) \right|$, which can be further used to bound the main term in the main result.

**Lemma C.11** (Self-bounding, private version of Lemma D.7 in [Yin and Wang, 2021b]). *Under the high probability events in Lemma C.3, Lemma C.6 and Lemma C.7, it holds that for all $t \in [H]$ and $s \in \mathcal{S}$,*

$$\left| V_t^{\dagger \pi^\star}(s) - \widetilde{V}_t(s) \right| \le \frac{4\sqrt{2\iota}H^2}{\sqrt{n \cdot \bar{d}_m}} + \frac{256SH^2 E_\rho \cdot \iota}{3n \cdot \bar{d}_m}.$$

*where $\bar{d}_m$ is defined in Theorem 3.4.*

*Proof of Lemma C.11.* According to (31), since $\mathrm{Var}_{\widetilde{P}_h(\cdot|s_h,a_h)}(\widetilde{V}_{h+1}(\cdot)) \le H^2$, we have for all $t \in [H]$,

$$\left| V_t^{\dagger \pi^\star}(s) - V_t^{\dagger \widehat{\pi}}(s) \right| \le \frac{4\sqrt{2\iota}H^2}{\sqrt{n \cdot \bar{d}_m}} + \frac{256SH^2 E_\rho \cdot \iota}{3n \cdot \bar{d}_m} \qquad (32)$$

Next, apply Lemma E.7 by setting $\pi = \widehat{\pi}$, $\pi' = \pi^\star$, $\widehat{Q} = \overline{Q}$, $\widehat{V} = \widetilde{V}$ under $M^\dagger$, then we have

$$V_t^{\dagger \pi^\star}(s) - \widetilde{V}_t(s) = \sum_{h=t}^{H} \mathbb{E}_{\pi^\star}^\dagger \left[ \xi_h^\dagger(s_h, a_h) \mid s_t = s \right] + \sum_{h=t}^{H} \mathbb{E}_{\pi^\star}^\dagger \left[ \langle \overline{Q}_h(s_h, \cdot), \pi_h^\star(\cdot|s_h) - \widehat{\pi}_h(\cdot|s_h) \rangle \mid s_t = s \right]$$

$$\le \sum_{h=t}^{H} \mathbb{E}_{\pi^\star}^\dagger \left[ \xi_h^\dagger(s_h, a_h) \mid s_t = s \right]$$

$$\le \frac{4\sqrt{2\iota}H^2}{\sqrt{n \cdot \bar{d}_m}} + \frac{256SH^2 E_\rho \cdot \iota}{3n \cdot \bar{d}_m}.$$

$$(33)$$

Also, apply Lemma E.7 by setting $\pi = \pi' = \widehat{\pi}$, $\widehat{Q} = \overline{Q}$, $\widehat{V} = \widetilde{V}$ under $M^\dagger$, then we have

$$\widetilde{V}_t(s) - V_t^{\dagger\widehat{\pi}}(s) = -\sum_{h=t}^{H} \mathbb{E}_{\widehat{\pi}}^{\dagger}\left[\xi_h^{\dagger}(s_h, a_h) \mid s_t = s\right] \leq 0. \tag{34}$$

The proof is complete by combing (32), (33) and (34). $\qquad\square$

Now we are ready to bound $\sqrt{\mathrm{Var}_{\widetilde{P}_h(\cdot|s_h,a_h)}(\widetilde{V}_{h+1}(\cdot))}$ by $\sqrt{\mathrm{Var}_{P_h(\cdot|s_h,a_h)}(V_{h+1}^{\dagger\star}(\cdot))}$. Under the high probability events in Lemma C.3, Lemma C.6 and Lemma C.7, with probability $1 - \delta$, it holds that for all $(s_h, a_h) \in \mathcal{C}_h$,

$$
\begin{aligned}
&\sqrt{\mathrm{Var}_{\widetilde{P}_h(\cdot|s_h,a_h)}(\widetilde{V}_{h+1}(\cdot))} \leq \sqrt{\mathrm{Var}_{\widetilde{P}_h(\cdot|s_h,a_h)}(V_{h+1}^{\dagger\star}(\cdot))} + \left\|\widetilde{V}_{h+1} - V_{h+1}^{\dagger\pi^\star}\right\|_{\infty, s\in\mathcal{S}} \\
&\leq \sqrt{\mathrm{Var}_{\widetilde{P}_h(\cdot|s_h,a_h)}(V_{h+1}^{\dagger\star}(\cdot))} + \frac{4\sqrt{2\iota}H^2}{\sqrt{n\cdot\bar{d}_m}} + \frac{256SH^2E_\rho\cdot\iota}{3n\cdot\bar{d}_m} \\
&\leq \sqrt{\mathrm{Var}_{\widehat{P}_h(\cdot|s_h,a_h)}(V_{h+1}^{\dagger\star}(\cdot))} + \frac{4\sqrt{2\iota}H^2}{\sqrt{n\cdot\bar{d}_m}} + \frac{256SH^2E_\rho\cdot\iota}{3n\cdot\bar{d}_m} + 4H\sqrt{\frac{SE_\rho}{\widetilde{n}_{s_h,a_h}}} \\
&\leq \sqrt{\mathrm{Var}_{\widehat{P}_h(\cdot|s_h,a_h)}(V_{h+1}^{\dagger\star}(\cdot))} + \frac{4\sqrt{2\iota}H^2}{\sqrt{n\cdot\bar{d}_m}} + \frac{256SH^2E_\rho\cdot\iota}{3n\cdot\bar{d}_m} + 8H\sqrt{\frac{SE_\rho}{n\cdot\bar{d}_m}} \\
&\leq \sqrt{\mathrm{Var}_{P_h(\cdot|s_h,a_h)}(V_{h+1}^{\dagger\star}(\cdot))} + \frac{4\sqrt{2\iota}H^2}{\sqrt{n\cdot\bar{d}_m}} + \frac{256SH^2E_\rho\cdot\iota}{3n\cdot\bar{d}_m} + 8H\sqrt{\frac{SE_\rho}{n\cdot\bar{d}_m}} + 3H\sqrt{\frac{\iota}{n\cdot\bar{d}_m}} \\
&\leq \sqrt{\mathrm{Var}_{P_h(\cdot|s_h,a_h)}(V_{h+1}^{\dagger\star}(\cdot))} + \frac{9\sqrt{\iota}H^2}{\sqrt{n\cdot\bar{d}_m}} + \frac{256SH^2E_\rho\cdot\iota}{3n\cdot\bar{d}_m} + 8H\sqrt{\frac{SE_\rho}{n\cdot\bar{d}_m}}.
\end{aligned}
\tag{35}
$$

The second inequality is because of Lemma C.11. The third inequality is due to Lemma C.5. The forth inequality comes from Lemma C.3 and Remark C.8. The fifth inequality holds with probability $1 - \delta$ because of Lemma E.5 and a union bound.

Finally, by plugging (35) into (31) and averaging over $s_1$, we finally have with probability $1 - 4\delta$,

$$
\begin{aligned}
v^{\pi^\star} - v^{\widehat{\pi}} = v^{\dagger\pi^\star} - v^{\dagger\widehat{\pi}} &\leq \sum_{h=1}^{H} \mathbb{E}_{\pi^\star}^{\dagger}\left[4\sqrt{\frac{2\mathrm{Var}_{\widetilde{P}_h(\cdot|s_h,a_h)}(\widetilde{V}_{h+1}(\cdot))\cdot\iota}{nd_h^\mu(s_h,a_h)}} + \frac{256SHE_\rho\cdot\iota}{3nd_h^\mu(s_h,a_h)}\right] \\
&\leq 4\sqrt{2}\sum_{h=1}^{H} \mathbb{E}_{\pi^\star}^{\dagger}\left[\sqrt{\frac{\mathrm{Var}_{P_h(\cdot|s_h,a_h)}(V_{h+1}^{\dagger\star}(\cdot))\cdot\iota}{nd_h^\mu(s_h,a_h)}}\right] + \widetilde{O}\left(\frac{H^3 + SH^2E_\rho}{n\cdot\bar{d}_m}\right) \\
&= 4\sqrt{2}\sum_{h=1}^{H}\sum_{(s_h,a_h)\in\mathcal{C}_h} d_h^{\pi^\star}(s_h,a_h)\sqrt{\frac{\mathrm{Var}_{P_h(\cdot|s_h,a_h)}(V_{h+1}^{\dagger\star}(\cdot))\cdot\iota}{nd_h^\mu(s_h,a_h)}} + \widetilde{O}\left(\frac{H^3 + SH^2E_\rho}{n\cdot\bar{d}_m}\right) \\
&= 4\sqrt{2}\sum_{h=1}^{H}\sum_{(s_h,a_h)\in\mathcal{C}_h} d_h^{\pi^\star}(s_h,a_h)\sqrt{\frac{\mathrm{Var}_{P_h(\cdot|s_h,a_h)}(V_{h+1}^{\star}(\cdot))\cdot\iota}{nd_h^\mu(s_h,a_h)}} + \widetilde{O}\left(\frac{H^3 + SH^2E_\rho}{n\cdot\bar{d}_m}\right),
\end{aligned}
\tag{36}
$$

where $\widetilde{O}$ absorbs constants and Polylog terms. The first equation is due to (29). The first inequality is because of (31). The second inequality comes from (35) and our assumption that $n\cdot\bar{d}_m \geq c_1 H^2$. The second equation uses the fact that $d_h^{\pi^\star}(s_h, a_h) = d_h^{\dagger\pi^\star}(s_h, a_h)$, for all $(s_h, a_h)$. The last equation is because for any $(s_h, a_h, s_{h+1})$ such that $d_h^{\pi^\star}(s_h, a_h) > 0$ and $P_h(s_{h+1}|s_h, a_h) > 0$, $V_{h+1}^{\dagger\star}(s_{h+1}) = V_{h+1}^{\star}(s_{h+1})$.

## C.4 Put everything together

Combining Lemma C.1 and (36), the proof of Theorem 3.4 is complete.

# D  Proof of Theorem 4.1

## D.1  Proof sketch

Since the whole proof for privacy guarantee is not very complex, we present it in Section D.2 below and only sketch the proof for suboptimality bound.

First of all, by extended value difference (Lemma E.7 and E.8), we can convert bounding the suboptimality gap of $v^\star - v^{\widehat{\pi}}$ to bounding $\sum_{h=1}^{H} \mathbb{E}_\pi \left[ \Gamma_h(s_h, a_h) \right]$, given that $|(\mathcal{T}_h \widetilde{V}_{h+1} - \widetilde{\mathcal{T}}_h \widetilde{V}_{h+1})(s,a)| \leq \Gamma_h(s,a)$ for all $s, a, h$. To bound $(\mathcal{T}_h \widetilde{V}_{h+1} - \widetilde{\mathcal{T}}_h \widetilde{V}_{h+1})(s,a)$, according to our analysis about the upper bound of the noises we add, we can decompose $(\mathcal{T}_h \widetilde{V}_{h+1} - \widetilde{\mathcal{T}}_h \widetilde{V}_{h+1})(s,a)$ to lower order terms $(\widetilde{O}(\frac{1}{K}))$ and the following key quantity:

$$\phi(s,a)^\top \widehat{\Lambda}_h^{-1} \left[ \sum_{\tau=1}^{K} \phi\left(s_h^\tau, a_h^\tau\right) \cdot \left( r_h^\tau + \widetilde{V}_{h+1}\left(s_{h+1}^\tau\right) - \left(\mathcal{T}_h \widetilde{V}_{h+1}\right)\left(s_h^\tau, a_h^\tau\right)\right) / \widetilde{\sigma}_h^2\left(s_h^\tau, a_h^\tau\right) \right]. \quad (37)$$

For the term above, we prove an upper bound of $\left\| \sigma_{\widetilde{V}_{h+1}}^2 - \widetilde{\sigma}_h^2 \right\|_\infty$, so we can convert $\widetilde{\sigma}_h^2$ to $\sigma_{\widetilde{V}_{h+1}}^2$. Next, since $\mathrm{Var}\left[ r_h^\tau + \widetilde{V}_{h+1}\left(s_{h+1}^\tau\right) - \left(\mathcal{T}_h \widetilde{V}_{h+1}\right)\left(s_h^\tau, a_h^\tau\right) \mid s_h^\tau, a_h^\tau \right] \approx \sigma_{\widetilde{V}_{h+1}}^2$, we can apply Bernstein's inequality for self-normalized martingale (Lemma E.10) as in Yin et al. [2022] for deriving tighter bound.

Finally, we replace the private statistics by non-private ones. More specifically, we convert $\sigma_{\widetilde{V}_{h+1}}^2$ to $\sigma_h^{\star 2}$ ($\Lambda_h^{-1}$ to $\Lambda_h^{\star -1}$) by combining the crude upper bound of $\left\| \widetilde{V} - V^\star \right\|_\infty$ and matrix concentrations.

## D.2  Proof of the privacy guarantee

The privacy guarantee of DP-VAPVI (Algorithm 2) is summarized by Lemma D.1 below.

**Lemma D.1** (Privacy analysis of DP-VAPVI (Algorithm 2)). *DP-VAPVI (Algorithm 2) satisfies $\rho$-zCDP.*

*Proof of Lemma D.1.* For $\sum_{\tau=1}^{K} \phi(\bar{s}_h^\tau, \bar{a}_h^\tau) \cdot \widetilde{V}_{h+1}(\bar{s}_{h+1}^\tau)^2$, the $\ell_2$ sensitivity is $2H^2$. For $\sum_{\tau=1}^{K} \phi(\bar{s}_h^\tau, \bar{a}_h^\tau) \cdot \widetilde{V}_{h+1}(\bar{s}_{h+1}^\tau)$ and $\sum_{\tau=1}^{K} \phi\left(s_h^\tau, a_h^\tau\right) \cdot \left( r_h^\tau + \widetilde{V}_{h+1}\left(s_{h+1}^\tau\right)\right) / \widetilde{\sigma}_h^2(s_h^\tau, a_h^\tau)$, the $\ell_2$ sensitivity is $2H$. Therefore according to Lemma 2.7, the use of Gaussian Mechanism (the additional noises $\phi_1, \phi_2, \phi_3$) ensures $\rho_0$-zCDP for each counter. For $\sum_{\tau=1}^{K} \phi(\bar{s}_h^\tau, \bar{a}_h^\tau) \phi(\bar{s}_h^\tau, \bar{a}_h^\tau)^\top + \lambda I$ and $\sum_{\tau=1}^{K} \phi\left(s_h^\tau, a_h^\tau\right) \phi\left(s_h^\tau, a_h^\tau\right)^\top / \widetilde{\sigma}_h^2(s_h^\tau, a_h^\tau) + \lambda I$, according to Appendix D in [Redberg and Wang, 2021], the per-instance $\ell_2$ sensitivity is

$$\|\Delta_x\|_2 = \frac{1}{\sqrt{2}} \sup_{\phi : \|\phi\|_2 \leq 1} \left\| \phi \phi^\top \right\|_F = \frac{1}{\sqrt{2}} \sup_{\phi : \|\phi\|_2 \leq 1} \sqrt{\sum_{i,j} \phi_i^2 \phi_j^2} = \frac{1}{\sqrt{2}}.$$

Therefore the use of Gaussian Mechanism (the additional noises $K_1, K_2$) also ensures $\rho_0$-zCDP for each counter.[12] Combining these results, according to Lemma E.17, the whole algorithm satisfies $5H\rho_0 = \rho$-zCDP. □

## D.3  Proof of the sub-optimality bound

### D.3.1  Utility analysis and some preparation

We begin with the following high probability bound of the noises we add.

---

[12]For more detailed explanation, we refer the readers to Appendix D of [Redberg and Wang, 2021].

**Lemma D.2** (Utility analysis). *Let $L = 2H\sqrt{\frac{d}{\rho_0}\log(\frac{10Hd}{\delta})} = 2H\sqrt{\frac{5Hd\log(\frac{10Hd}{\delta})}{\rho}}$ and*

$E = \sqrt{\frac{2d}{\rho_0}}\left(2 + \left(\frac{\log(5c_1 H/\delta)}{c_2 d}\right)^{\frac{2}{3}}\right) = \sqrt{\frac{10Hd}{\rho}}\left(2 + \left(\frac{\log(5c_1 H/\delta)}{c_2 d}\right)^{\frac{2}{3}}\right)$ *for some universal constants*
$c_1, c_2$. *Then with probability $1 - \delta$, the following inequalities hold simultaneously:*

$$\text{For all } h \in [H], \ \|\phi_1\|_2 \leq HL, \|\phi_2\|_2 \leq L, \ \|\phi_3\|_2 \leq L.$$
$$\text{For all } h \in [H], \ K_1, K_2 \text{ are symmetric and positive definite and } \|K_i\|_2 \leq E, \ i \in \{1, 2\}. \tag{38}$$

*Proof of Lemma D.2.* The second line of (38) results from Lemma 19 in [Redberg and Wang, 2021] and Weyl's Inequality. The first line of (38) directly results from the concentration inequality for Guassian distribution and a union bound. $\square$

Define the Bellman update error $\zeta_h(s,a) := (\mathcal{T}_h \widetilde{V}_{h+1})(s,a) - \widehat{Q}_h(s,a)$ and recall
$\widehat{\pi}_h(s) = \arg\max_{\pi_h}\langle \widehat{Q}_h(s,\cdot), \pi_h(\cdot \mid s)\rangle_{\mathcal{A}}$, then because of Lemma E.8,

$$V_1^\pi(s) - V_1^{\widehat{\pi}}(s) \leq \sum_{h=1}^{H} \mathbb{E}_\pi\left[\zeta_h(s_h, a_h) \mid s_1 = s\right] - \sum_{h=1}^{H} \mathbb{E}_{\widehat{\pi}}\left[\zeta_h(s_h, a_h) \mid s_1 = s\right]. \tag{39}$$

Define $\widetilde{\mathcal{T}}_h \widetilde{V}_{h+1}(\cdot, \cdot) = \phi(\cdot, \cdot)^\top \widetilde{w}_h$. Then similar to Lemma C.10, we have the following lemma showing that in order to bound the sub-optimality, it is sufficient to bound the pessimistic penalty.

**Lemma D.3** (Lemma C.1 in [Yin et al., 2022]). *Suppose with probability $1 - \delta$, it holds for all $s, a, h \in \mathcal{S} \times \mathcal{A} \times [H]$ that $|(\mathcal{T}_h \widetilde{V}_{h+1} - \widetilde{\mathcal{T}}_h \widetilde{V}_{h+1})(s,a)| \leq \Gamma_h(s,a)$, then it implies $\forall s, a, h \in \mathcal{S} \times \mathcal{A} \times [H]$, $0 \leq \zeta_h(s,a) \leq 2\Gamma_h(s,a)$. Furthermore, with probability $1 - \delta$, it holds for any policy $\pi$ simultaneously,*

$$V_1^\pi(s) - V_1^{\widehat{\pi}}(s) \leq \sum_{h=1}^{H} 2 \cdot \mathbb{E}_\pi\left[\Gamma_h(s_h, a_h) \mid s_1 = s\right].$$

*Proof of Lemma D.3.* We first show given $|(\mathcal{T}_h \widetilde{V}_{h+1} - \widetilde{\mathcal{T}}_h \widetilde{V}_{h+1})(s,a)| \leq \Gamma_h(s,a)$, then $0 \leq \zeta_h(s,a) \leq 2\Gamma_h(s,a), \forall s, a, h \in \mathcal{S} \times \mathcal{A} \times [H]$.

**Step 1:** The first step is to show $0 \leq \zeta_h(s,a), \forall s, a, h \in \mathcal{S} \times \mathcal{A} \times [H]$.

Indeed, if $\bar{Q}_h(s,a) \leq 0$, then by definition $\widehat{Q}_h(s,a) = 0$ and therefore $\zeta_h(s,a) := (\mathcal{T}_h \widetilde{V}_{h+1})(s,a) - \widehat{Q}_h(s,a) = (\mathcal{T}_h \widetilde{V}_{h+1})(s,a) \geq 0$. If $\bar{Q}_h(s,a) > 0$, then $\widehat{Q}_h(s,a) \leq \bar{Q}_h(s,a)$ and

$$\begin{aligned}\zeta_h(s,a) :&= (\mathcal{T}_h \widetilde{V}_{h+1})(s,a) - \widehat{Q}_h(s,a) \geq (\mathcal{T}_h \widetilde{V}_{h+1})(s,a) - \bar{Q}_h(s,a)\\ &= (\mathcal{T}_h \widetilde{V}_{h+1})(s,a) - (\widetilde{\mathcal{T}}_h \widetilde{V}_{h+1})(s,a) + \Gamma_h(s,a) \geq 0.\end{aligned}$$

**Step 2:** The second step is to show $\zeta_h(s,a) \leq 2\Gamma_h(s,a), \forall s, a, h \in \mathcal{S} \times \mathcal{A} \times [H]$.

Under the assumption that $|(\mathcal{T}_h \widetilde{V}_{h+1} - \widetilde{\mathcal{T}}_h \widetilde{V}_{h+1})(s,a)| \leq \Gamma_h(s,a)$, we have

$$\bar{Q}_h(s,a) = (\widetilde{\mathcal{T}}_h \widetilde{V}_{h+1})(s,a) - \Gamma_h(s,a) \leq (\mathcal{T}_h \widetilde{V}_{h+1})(s,a) \leq H - h + 1,$$

which implies that $\widehat{Q}_h(s,a) = \max(\bar{Q}_h(s,a), 0)$. Therefore, it holds that

$$\begin{aligned}\zeta_h(s,a) :&= (\mathcal{T}_h \widetilde{V}_{h+1})(s,a) - \widehat{Q}_h(s,a) \leq (\mathcal{T}_h \widetilde{V}_{h+1})(s,a) - \bar{Q}_h(s,a)\\ &= (\mathcal{T}_h \widetilde{V}_{h+1})(s,a) - (\widetilde{\mathcal{T}}_h \widetilde{V}_{h+1})(s,a) + \Gamma_h(s,a) \leq 2 \cdot \Gamma_h(s,a).\end{aligned}$$

For the last statement, denote $\mathfrak{F} := \{0 \leq \zeta_h(s,a) \leq 2\Gamma_h(s,a), \ \forall s, a, h \in \mathcal{S} \times \mathcal{A} \times [H]\}$. Note conditional on $\mathfrak{F}$, then by (39), $V_1^\pi(s) - V_1^{\widehat{\pi}}(s) \leq \sum_{h=1}^{H} 2 \cdot \mathbb{E}_\pi[\Gamma_h(s_h, a_h) \mid s_1 = s]$ holds for any

policy $\pi$ almost surely. Therefore,

$$\mathbb{P}\left[\forall\pi,\ V_1^\pi(s)-V_1^{\widehat{\pi}}(s)\leq\sum_{h=1}^H 2\cdot\mathbb{E}_\pi[\Gamma_h(s_h,a_h)\mid s_1=s].\right]$$

$$=\mathbb{P}\left[\forall\pi,\ V_1^\pi(s)-V_1^{\widehat{\pi}}(s)\leq\sum_{h=1}^H 2\cdot\mathbb{E}_\pi[\Gamma_h(s_h,a_h)\mid s_1=s]\middle|\mathfrak{F}\right]\cdot\mathbb{P}[\mathfrak{F}]$$

$$+\mathbb{P}\left[\forall\pi,\ V_1^\pi(s)-V_1^{\widehat{\pi}}(s)\leq\sum_{h=1}^H 2\cdot\mathbb{E}_\pi[\Gamma_h(s_h,a_h)\mid s_1=s]\middle|\mathfrak{F}^c\right]\cdot\mathbb{P}[\mathfrak{F}^c]$$

$$\geq\mathbb{P}\left[\forall\pi,\ V_1^\pi(s)-V_1^{\widehat{\pi}}(s)\leq\sum_{h=1}^H 2\cdot\mathbb{E}_\pi[\Gamma_h(s_h,a_h)\mid s_1=s]\middle|\mathfrak{F}\right]\cdot\mathbb{P}[\mathfrak{F}]=1\cdot\mathbb{P}[\mathfrak{F}]\geq 1-\delta,$$

which finishes the proof. $\qquad\square$

### D.3.2 Bound the pessimistic penalty

By Lemma D.3, it remains to bound $|(\mathcal{T}_h\widetilde{V}_{h+1})(s,a)-(\widetilde{\mathcal{T}}_h\widetilde{V}_{h+1})(s,a)|$. Suppose $w_h$ is the coefficient corresponding to the $\mathcal{T}_h\widetilde{V}_{h+1}$ (such $w_h$ exists by Lemma E.14), *i.e.* $\mathcal{T}_h\widetilde{V}_{h+1}=\phi^\top w_h$, and recall $(\widetilde{\mathcal{T}}_h\widetilde{V}_{h+1})(s,a)=\phi(s,a)^\top\widetilde{w}_h$, then:

$$\left(\mathcal{T}_h\widetilde{V}_{h+1}\right)(s,a)-\left(\widetilde{\mathcal{T}}_h\widetilde{V}_{h+1}\right)(s,a)=\phi(s,a)^\top(w_h-\widetilde{w}_h)$$

$$=\phi(s,a)^\top w_h-\phi(s,a)^\top\widetilde{\Lambda}_h^{-1}\left(\sum_{\tau=1}^K\phi(s_h^\tau,a_h^\tau)\cdot\left(r_h^\tau+\widetilde{V}_{h+1}\left(s_{h+1}^\tau\right)\right)/\widetilde{\sigma}_h^2(s_h^\tau,a_h^\tau)+\phi_3\right)$$

$$=\underbrace{\phi(s,a)^\top w_h-\phi(s,a)^\top\widehat{\Lambda}_h^{-1}\left(\sum_{\tau=1}^K\phi(s_h^\tau,a_h^\tau)\cdot\left(r_h^\tau+\widetilde{V}_{h+1}\left(s_{h+1}^\tau\right)\right)/\widetilde{\sigma}_h^2(s_h^\tau,a_h^\tau)\right)}_{\text{(i)}}$$

$$\underbrace{-\phi(s,a)^\top\widehat{\Lambda}_h^{-1}\phi_3}_{\text{(ii)}}+\underbrace{\phi(s,a)^\top(\widehat{\Lambda}_h^{-1}-\widetilde{\Lambda}_h^{-1})\left(\sum_{\tau=1}^K\phi(s_h^\tau,a_h^\tau)\cdot\left(r_h^\tau+\widetilde{V}_{h+1}\left(s_{h+1}^\tau\right)\right)/\widetilde{\sigma}_h^2(s_h^\tau,a_h^\tau)+\phi_3\right)}_{\text{(iii)}},$$

$$\tag{40}$$

where $\widehat{\Lambda}_h=\widetilde{\Lambda}_h-K_2=\sum_{\tau=1}^K\phi(s_h^\tau,a_h^\tau)\phi(s_h^\tau,a_h^\tau)^\top/\widetilde{\sigma}_h^2(s_h^\tau,a_h^\tau)+\lambda I$.

Term (ii) can be handled by the following Lemma D.4

**Lemma D.4.** *Recall $\kappa$ in Assumption 2.2. Under the high probability event in Lemma D.2, suppose* $K\geq\max\left\{\frac{512H^4\cdot\log\left(\frac{2Hd}{\delta}\right)}{\kappa^2},\frac{4\lambda H^2}{\kappa}\right\}$, *then with probability $1-\delta$, for all $s,a,h\in\mathcal{S}\times\mathcal{A}\times[H]$, it holds that*

$$\left|\phi(s,a)^\top\widehat{\Lambda}_h^{-1}\phi_3\right|\leq\frac{4H^2L/\kappa}{K}.$$

*Proof of Lemma D.4.* Define $\widetilde{\Lambda}_h^p=\mathbb{E}_{\mu,h}[\widetilde{\sigma}_h^{-2}(s,a)\phi(s,a)\phi(s,a)^\top]$. Then because of Assumption 2.2 and $\widetilde{\sigma}_h\leq H$, it holds that $\lambda_{\min}(\widetilde{\Lambda}_h^p)\geq\frac{\kappa}{H^2}$. Therefore, due to Lemma E.13, we have with

probability $1 - \delta$,

$$\left|\phi(s,a)^\top \widehat{\Lambda}_h^{-1} \phi_3\right| \leq \|\phi(s,a)\|_{\widehat{\Lambda}_h^{-1}} \cdot \|\phi_3\|_{\widehat{\Lambda}_h^{-1}}$$

$$\leq \frac{4}{K} \|\phi(s,a)\|_{(\widetilde{\Lambda}_h^p)^{-1}} \cdot \|\phi_3\|_{(\widetilde{\Lambda}_h^p)^{-1}}$$

$$\leq \frac{4L}{K} \|(\widetilde{\Lambda}_h^p)^{-1}\|$$

$$\leq \frac{4H^2 L/\kappa}{K}.$$

The first inequality is because of Cauchy-Schwarz inequality. The second inequality holds with probability $1 - \delta$ due to Lemma E.13 and a union bound. The third inequality holds because $\sqrt{a^\top \cdot A \cdot a} \leq \sqrt{\|a\|_2 \|A\|_2 \|a\|_2} = \|a\|_2 \sqrt{\|A\|_2}$. The last inequality arises from $\|(\widetilde{\Lambda}_h^p)^{-1}\| = \lambda_{\max}((\widetilde{\Lambda}_h^p)^{-1}) = \lambda_{\min}^{-1}(\widetilde{\Lambda}_h^p) \leq \frac{H^2}{\kappa}$. $\square$

The difference between $\widetilde{\Lambda}_h^{-1}$ and $\widehat{\Lambda}_h^{-1}$ can be bounded by the following Lemma D.5

**Lemma D.5.** *Under the high probability event in Lemma D.2, suppose $K \geq \frac{128 H^4 \log \frac{2dH}{\delta}}{\kappa^2}$, then with probability $1 - \delta$, for all $h \in [H]$, it holds that $\|\widehat{\Lambda}_h^{-1} - \widetilde{\Lambda}_h^{-1}\| \leq \frac{4H^4 E/\kappa^2}{K^2}$.*

*Proof of Lemma D.5.* First of all, we have

$$\|\widehat{\Lambda}_h^{-1} - \widetilde{\Lambda}_h^{-1}\| = \|\widehat{\Lambda}_h^{-1} \cdot (\widehat{\Lambda}_h - \widetilde{\Lambda}_h) \cdot \widetilde{\Lambda}_h^{-1}\|$$

$$\leq \|\widehat{\Lambda}_h^{-1}\| \cdot \|\widehat{\Lambda}_h - \widetilde{\Lambda}_h\| \cdot \|\widetilde{\Lambda}_h^{-1}\| \qquad (41)$$

$$\leq \lambda_{\min}^{-1}(\widehat{\Lambda}_h) \cdot \lambda_{\min}^{-1}(\widetilde{\Lambda}_h) \cdot E.$$

The first inequality is because $\|A \cdot B\| \leq \|A\| \cdot \|B\|$. The second inequality is due to Lemma D.2.

Let $\widehat{\Lambda}_h' = \frac{1}{K} \widehat{\Lambda}_h$, then because of Lemma E.12, with probability $1 - \delta$, it holds that for all $h \in [H]$,

$$\left\| \widehat{\Lambda}_h' - \mathbb{E}_{\mu,h}[\phi(s,a)\phi(s,a)^\top/\widetilde{\sigma}_h^2(s,a)] - \frac{\lambda}{K} I_d \right\| \leq \frac{4\sqrt{2}}{\sqrt{K}} \left( \log \frac{2dH}{\delta} \right)^{1/2},$$

which implies that when $K \geq \frac{128 H^4 \log \frac{2dH}{\delta}}{\kappa^2}$, it holds that (according to Weyl's Inequality)

$$\lambda_{\min}(\widehat{\Lambda}_h') \geq \lambda_{\min}(\mathbb{E}_{\mu,h}[\phi(s,a)\phi(s,a)^\top/\widetilde{\sigma}_h^2(s,a)]) + \frac{\lambda}{K} - \frac{\kappa}{2H^2} \geq \frac{\kappa}{2H^2}.$$

Under this high probability event, we have $\lambda_{\min}(\widehat{\Lambda}_h) \geq \frac{K\kappa}{2H^2}$ and therefore $\lambda_{\min}(\widetilde{\Lambda}_h) \geq \lambda_{\min}(\widehat{\Lambda}_h) \geq \frac{K\kappa}{2H^2}$. Plugging these two results into (41), we have

$$\|\widehat{\Lambda}_h^{-1} - \widetilde{\Lambda}_h^{-1}\| \leq \frac{4H^4 E/\kappa^2}{K^2}.$$

$\square$

Then we can bound term (iii) by the following Lemma D.6

**Lemma D.6.** *Suppose $K \geq \max\{\frac{128 H^4 \log \frac{2dH}{\delta}}{\kappa^2}, \frac{\sqrt{2}L}{\sqrt{d}\kappa}\}$, under the high probability events in Lemma D.2 and Lemma D.5, it holds that for all $s, a, h \in \mathcal{S} \times \mathcal{A} \times [H]$,*

$$\left| \phi(s,a)^\top (\widehat{\Lambda}_h^{-1} - \widetilde{\Lambda}_h^{-1}) \left( \sum_{\tau=1}^K \phi(s_h^\tau, a_h^\tau) \cdot \left( r_h^\tau + \widetilde{V}_{h+1}(s_{h+1}^\tau) \right) / \widetilde{\sigma}_h^2(s_h^\tau, a_h^\tau) + \phi_3 \right) \right| \leq \frac{4\sqrt{2}H^4 E\sqrt{d}/\kappa^{3/2}}{K}.$$

*Proof of Lemma D.6.* First of all, the left hand side is bounded by

$$\left\| (\widehat{\Lambda}_h^{-1} - \widetilde{\Lambda}_h^{-1}) \left( \sum_{\tau=1}^K \phi(s_h^\tau, a_h^\tau) \cdot \left( r_h^\tau + \widetilde{V}_{h+1}(s_{h+1}^\tau) \right) / \widetilde{\sigma}_h^2(s_h^\tau, a_h^\tau) \right) \right\|_2 + \frac{4H^4 EL/\kappa^2}{K^2}$$

due to Lemma D.5. Then the left hand side can be further bounded by

$$
H \sum_{\tau=1}^{K} \left\| (\widehat{\Lambda}_h^{-1} - \widetilde{\Lambda}_h^{-1}) \phi\left(s_h^\tau, a_h^\tau\right) / \widetilde{\sigma}_h(s_h^\tau, a_h^\tau) \right\|_2 + \frac{4 H^4 EL/\kappa^2}{K^2}
$$

$$
\leq H \sum_{\tau=1}^{K} \sqrt{ Tr\left( (\widehat{\Lambda}_h^{-1} - \widetilde{\Lambda}_h^{-1}) \cdot \frac{\phi\left(s_h^\tau, a_h^\tau\right) \phi\left(s_h^\tau, a_h^\tau\right)^\top}{\widetilde{\sigma}_h^2(s_h^\tau, a_h^\tau)} \cdot (\widehat{\Lambda}_h^{-1} - \widetilde{\Lambda}_h^{-1}) \right) } + \frac{4 H^4 EL/\kappa^2}{K^2}
$$

$$
\leq H \sqrt{ K \cdot Tr\left( (\widehat{\Lambda}_h^{-1} - \widetilde{\Lambda}_h^{-1}) \cdot \widehat{\Lambda}_h \cdot (\widehat{\Lambda}_h^{-1} - \widetilde{\Lambda}_h^{-1}) \right) } + \frac{4 H^4 EL/\kappa^2}{K^2}
$$

$$
\leq H \sqrt{ K d \cdot \lambda_{\max}\left( (\widehat{\Lambda}_h^{-1} - \widetilde{\Lambda}_h^{-1}) \cdot \widehat{\Lambda}_h \cdot (\widehat{\Lambda}_h^{-1} - \widetilde{\Lambda}_h^{-1}) \right) } + \frac{4 H^4 EL/\kappa^2}{K^2}
$$

$$
= H \sqrt{ K d \cdot \left\| (\widehat{\Lambda}_h^{-1} - \widetilde{\Lambda}_h^{-1}) \cdot \widehat{\Lambda}_h \cdot (\widehat{\Lambda}_h^{-1} - \widetilde{\Lambda}_h^{-1}) \right\|_2 } + \frac{4 H^4 EL/\kappa^2}{K^2}
$$

$$
\leq H \sqrt{ K d \cdot \left\| \widetilde{\Lambda}_h^{-1} \right\|_2 \cdot \left\| \widetilde{\Lambda}_h - \widehat{\Lambda}_h \right\|_2 \cdot \left\| \widehat{\Lambda}_h^{-1} - \widetilde{\Lambda}_h^{-1} \right\|_2 } + \frac{4 H^4 EL/\kappa^2}{K^2}
$$

$$
\leq \frac{2\sqrt{2} H^4 E \sqrt{d}/\kappa^{3/2}}{K} + \frac{4 H^4 EL/\kappa^2}{K^2}
$$

$$
\leq \frac{4\sqrt{2} H^4 E \sqrt{d}/\kappa^{3/2}}{K}.
$$

The first inequality is because $\|a\|_2 = \sqrt{a^\top a} = \sqrt{Tr(aa^\top)}$. The second inequality is due to Cauchy-Schwarz inequality. The third inequality is because for positive definite matrix $A$, it holds that $Tr(A) = \sum_{i=1}^d \lambda_i(A) \leq d\lambda_{\max}(A)$. The equation is because for symmetric, positive definite matrix $A$, $\|A\|_2 = \lambda_{\max}(A)$. The forth inequality is due to $\|A \cdot B\| \leq \|A\| \cdot \|B\|$. The fifth inequality is because of Lemma D.2, Lemma D.5 and the statement in the proof of Lemma D.5 that $\lambda_{\min}(\widetilde{\Lambda}_h) \geq \frac{K\kappa}{2H^2}$. The last inequality uses the assumption that $K \geq \frac{\sqrt{2}L}{\sqrt{d}\kappa}$. $\qquad\square$

Now the remaining part is term (i), we have

$$
\underbrace{ \phi(s,a)^\top w_h - \phi(s,a)^\top \widehat{\Lambda}_h^{-1} \left( \sum_{\tau=1}^{K} \phi\left(s_h^\tau, a_h^\tau\right) \cdot \left( r_h^\tau + \widetilde{V}_{h+1}\left(s_{h+1}^\tau\right) \right) / \widetilde{\sigma}_h^2(s_h^\tau, a_h^\tau) \right) }_{(i)}
$$

$$
= \underbrace{ \phi(s,a)^\top w_h - \phi(s,a)^\top \widehat{\Lambda}_h^{-1} \left( \sum_{\tau=1}^{K} \phi\left(s_h^\tau, a_h^\tau\right) \cdot \left( \mathcal{T}_h \widetilde{V}_{h+1} \right)\left(s_h^\tau, a_h^\tau\right) / \widetilde{\sigma}_h^2(s_h^\tau, a_h^\tau) \right) }_{(iv)}
$$

$$
\underbrace{ - \phi(s,a)^\top \widehat{\Lambda}_h^{-1} \left( \sum_{\tau=1}^{K} \phi\left(s_h^\tau, a_h^\tau\right) \cdot \left( r_h^\tau + \widetilde{V}_{h+1}\left(s_{h+1}^\tau\right) - \left( \mathcal{T}_h \widetilde{V}_{h+1} \right)\left(s_h^\tau, a_h^\tau\right) \right) / \widetilde{\sigma}_h^2(s_h^\tau, a_h^\tau) \right) }_{(v)}.
$$

$$
\tag{42}
$$

We are able to bound term (iv) by the following Lemma D.7.

**Lemma D.7.** *Recall $\kappa$ in Assumption 2.2. Under the high probability event in Lemma D.2, suppose $K \geq \max\left\{ \frac{512 H^4 \cdot \log\left(\frac{2Hd}{\delta}\right)}{\kappa^2}, \frac{4\lambda H^2}{\kappa} \right\}$, then with probability $1 - \delta$, for all $s, a, h \in \mathcal{S} \times \mathcal{A} \times [H]$,*

$$
\left| \phi(s,a)^\top w_h - \phi(s,a)^\top \widehat{\Lambda}_h^{-1} \left( \sum_{\tau=1}^{K} \phi\left(s_h^\tau, a_h^\tau\right) \cdot \left( \mathcal{T}_h \widetilde{V}_{h+1} \right)\left(s_h^\tau, a_h^\tau\right) / \widetilde{\sigma}_h^2(s_h^\tau, a_h^\tau) \right) \right| \leq \frac{8\lambda H^3 \sqrt{d}/\kappa}{K}.
$$

*Proof of Lemma D.7.* Recall $\mathcal{T}_h \widetilde{V}_{h+1} = \phi^\top w_h$ and apply Lemma E.13, we obtain with probability $1 - \delta$, for all $s, a, h \in \mathcal{S} \times \mathcal{A} \times [H]$,

$$\left| \phi(s,a)^\top w_h - \phi(s,a)^\top \widehat{\Lambda}_h^{-1} \left( \sum_{\tau=1}^K \phi(s_h^\tau, a_h^\tau) \cdot \left( \mathcal{T}_h \widetilde{V}_{h+1} \right) (s_h^\tau, a_h^\tau) / \widetilde{\sigma}_h^2(s_h^\tau, a_h^\tau) \right) \right|$$

$$= \left| \phi(s,a)^\top w_h - \phi(s,a)^\top \widehat{\Lambda}_h^{-1} \left( \sum_{\tau=1}^K \phi(s_h^\tau, a_h^\tau) \cdot \phi(s_h^\tau, a_h^\tau)^\top w_h / \widetilde{\sigma}_h^2(s_h^\tau, a_h^\tau) \right) \right|$$

$$= \left| \phi(s,a)^\top w_h - \phi(s,a)^\top \widehat{\Lambda}_h^{-1} \left( \widehat{\Lambda}_h - \lambda I \right) w_h \right|$$

$$= \left| \lambda \cdot \phi(s,a)^\top \widehat{\Lambda}_h^{-1} w_h \right|$$

$$\leq \lambda \| \phi(s,a) \|_{\widehat{\Lambda}_h^{-1}} \cdot \| w_h \|_{\widehat{\Lambda}_h^{-1}}$$

$$\leq \frac{4\lambda}{K} \| \phi(s,a) \|_{(\widetilde{\Lambda}_h^p)^{-1}} \cdot \| w_h \|_{(\widetilde{\Lambda}_h^p)^{-1}}$$

$$\leq \frac{4\lambda}{K} \cdot 2H\sqrt{d} \cdot \left\| (\widetilde{\Lambda}_h^p)^{-1} \right\|$$

$$\leq \frac{8\lambda H^3 \sqrt{d}/\kappa}{K},$$

where $\widetilde{\Lambda}_h^p := \mathbb{E}_{\mu,h} \left[ \widetilde{\sigma}_h(s,a)^{-2} \phi(s,a)\phi(s,a)^\top \right]$. The first inequality applies Cauchy-Schwarz inequality. The second inequality holds with probability $1 - \delta$ due to Lemma E.13 and a union bound. The third inequality uses $\sqrt{a^\top \cdot A \cdot a} \leq \sqrt{\|a\|_2 \|A\|_2 \|a\|_2} = \|a\|_2 \sqrt{\|A\|_2}$ and $\|w_h\| \leq 2H\sqrt{d}$. Finally, as $\lambda_{\min}(\widetilde{\Lambda}_h^p) \geq \frac{\kappa}{\max_{h,s,a} \widetilde{\sigma}_h(s,a)^2} \geq \frac{\kappa}{H^2}$ implies $\left\| (\widetilde{\Lambda}_h^p)^{-1} \right\| \leq \frac{H^2}{\kappa}$, the last inequality holds. □

For term (v), denote: $x_\tau = \frac{\phi(s_h^\tau, a_h^\tau)}{\widetilde{\sigma}_h(s_h^\tau, a_h^\tau)}, \quad \eta_\tau = \left( r_h^\tau + \widetilde{V}_{h+1}(s_{h+1}^\tau) - \left( \mathcal{T}_h \widetilde{V}_{h+1} \right)(s_h^\tau, a_h^\tau) \right) / \widetilde{\sigma}_h(s_h^\tau, a_h^\tau)$, then by Cauchy-Schwarz inequality, it holds that for all $h, s, a \in [H] \times \mathcal{S} \times \mathcal{A}$,

$$\left| \phi(s,a)^\top \widehat{\Lambda}_h^{-1} \left( \sum_{\tau=1}^K \phi(s_h^\tau, a_h^\tau) \cdot \left( r_h^\tau + \widetilde{V}_{h+1}(s_{h+1}^\tau) - \left( \mathcal{T}_h \widetilde{V}_{h+1} \right)(s_h^\tau, a_h^\tau) \right) / \widetilde{\sigma}_h^2(s_h^\tau, a_h^\tau) \right) \right|$$

$$\leq \sqrt{\phi(s,a)^\top \widehat{\Lambda}_h^{-1} \phi(s,a)} \cdot \left\| \sum_{\tau=1}^K x_\tau \eta_\tau \right\|_{\widehat{\Lambda}_h^{-1}}.$$

$$(43)$$

We bound $\sqrt{\phi(s,a)^\top \widehat{\Lambda}_h^{-1} \phi(s,a)}$ by $\sqrt{\phi(s,a)^\top \widetilde{\Lambda}_h^{-1} \phi(s,a)}$ using the following Lemma D.8.

**Lemma D.8.** *Suppose $K \geq \max\{ \frac{128 H^4 \log \frac{2dH}{\delta}}{\kappa^2}, \frac{\sqrt{2}L}{\sqrt{d}\kappa} \}$, under the high probability events in Lemma D.2 and Lemma D.5, it holds that for all $s, a, h \in \mathcal{S} \times \mathcal{A} \times [H]$,*

$$\sqrt{\phi(s,a)^\top \widehat{\Lambda}_h^{-1} \phi(s,a)} \leq \sqrt{\phi(s,a)^\top \widetilde{\Lambda}_h^{-1} \phi(s,a)} + \frac{2H^2 \sqrt{E}/\kappa}{K}.$$

*Proof of Lemma D.8.*

$$\sqrt{\phi(s,a)^\top \widehat{\Lambda}_h^{-1} \phi(s,a)} = \sqrt{\phi(s,a)^\top \widetilde{\Lambda}_h^{-1} \phi(s,a) + \phi(s,a)^\top (\widehat{\Lambda}_h^{-1} - \widetilde{\Lambda}_h^{-1}) \phi(s,a)}$$

$$\leq \sqrt{\phi(s,a)^\top \widetilde{\Lambda}_h^{-1} \phi(s,a) + \left\| \widehat{\Lambda}_h^{-1} - \widetilde{\Lambda}_h^{-1} \right\|_2}$$

$$\leq \sqrt{\phi(s,a)^\top \widetilde{\Lambda}_h^{-1} \phi(s,a)} + \sqrt{\left\| \widehat{\Lambda}_h^{-1} - \widetilde{\Lambda}_h^{-1} \right\|_2}$$

$$(44)$$

$$\leq \sqrt{\phi(s,a)^\top \widetilde{\Lambda}_h^{-1} \phi(s,a)} + \frac{2H^2 \sqrt{E}/\kappa}{K}.$$

The first inequality uses $|a^\top A a| \leq \|a\|_2^2 \cdot \|A\|$. The second inequality is because for $a, b \geq 0$, $\sqrt{a} + \sqrt{b} \geq \sqrt{a+b}$. The last inequality uses Lemma D.5. $\qquad\square$

**Remark D.9.** *Similarly, under the same assumption in Lemma D.8, we also have for all $s, a, h \in \mathcal{S} \times \mathcal{A} \times [H]$,*

$$\sqrt{\phi(s,a)^\top \widetilde{\Lambda}_h^{-1} \phi(s,a)} \leq \sqrt{\phi(s,a)^\top \widehat{\Lambda}_h^{-1} \phi(s,a)} + \frac{2H^2\sqrt{E}/\kappa}{K}.$$

### D.3.3 An intermediate result: bounding the variance

Before we handle $\left\|\sum_{\tau=1}^K x_\tau \eta_\tau\right\|_{\widehat{\Lambda}_h^{-1}}$, we first bound $\sup_h \left\|\widetilde{\sigma}_h^2 - \sigma_{\widetilde{V}_{h+1}}^2\right\|_\infty$ by the following Lemma D.10.

**Lemma D.10** (Private version of Lemma C.7 in [Yin et al., 2022]). *Recall the definition of $\widetilde{\sigma}_h(\cdot,\cdot)^2 = \max\{1, \widetilde{\mathrm{Var}}_h \widetilde{V}_{h+1}(\cdot,\cdot)\}$ in Algorithm 2 where $[\widetilde{\mathrm{Var}}_h \widetilde{V}_{h+1}](\cdot,\cdot) = \langle \phi(\cdot,\cdot), \widetilde{\beta}_h \rangle_{[0,(H-h+1)^2]} - [\langle \phi(\cdot,\cdot), \widetilde{\theta}_h \rangle_{[0,H-h+1]}]^2$ ($\widetilde{\beta}_h$ and $\widetilde{\theta}_h$ are defined in Algorithm 2) and $\sigma_{\widetilde{V}_{h+1}}(\cdot,\cdot)^2 := \max\{1, \mathrm{Var}_{P_h} \widetilde{V}_{h+1}(\cdot,\cdot)\}$. Suppose $K \geq \max\left\{\frac{512\log\left(\frac{2Hd}{\delta}\right)}{\kappa^2}, \frac{4\lambda}{\kappa}, \frac{128\log\frac{2dH}{\delta}}{\kappa^2}, \frac{\sqrt{2}L}{H\sqrt{d\kappa}}\right\}$ and $K \geq \max\{\frac{4L^2}{H^2d^3\kappa}, \frac{32E^2}{d^2\kappa^2}, \frac{16\lambda^2}{d^2\kappa}\}$, under the high probability event in Lemma D.2, it holds that with probability $1 - 6\delta$,*

$$\sup_h \|\widetilde{\sigma}_h^2 - \sigma_{\widetilde{V}_{h+1}}^2\|_\infty \leq 36\sqrt{\frac{H^4 d^3}{\kappa K} \log\left(\frac{(\lambda+K)2KdH^2}{\lambda\delta}\right)}.$$

*Proof of Lemma D.10.* **Step 1:** The first step is to show for all $h, s, a \in [H] \times \mathcal{S} \times \mathcal{A}$, with probability $1 - 3\delta$,

$$\left|\langle \phi(s,a), \widetilde{\beta}_h \rangle_{[0,(H-h+1)^2]} - \mathbb{P}_h(\widetilde{V}_{h+1})^2(s,a)\right| \leq 12\sqrt{\frac{H^4 d^3}{\kappa K} \log\left(\frac{(\lambda+K)2KdH^2}{\lambda\delta}\right)}.$$

**Proof of Step 1.** We can bound the left hand side by the following decomposition:

$$\left|\langle \phi(s,a), \widetilde{\beta}_h \rangle_{[0,(H-h+1)^2]} - \mathbb{P}_h(\widetilde{V}_{h+1})^2(s,a)\right| \leq \left|\langle \phi(s,a), \widetilde{\beta}_h \rangle - \mathbb{P}_h(\widetilde{V}_{h+1})^2(s,a)\right|$$

$$= \left|\phi(s,a)^\top \widetilde{\Sigma}_h^{-1}\left(\sum_{\tau=1}^K \phi(\bar{s}_h^\tau, \bar{a}_h^\tau) \cdot \widetilde{V}_{h+1}(\bar{s}_{h+1}^\tau)^2 + \phi_1\right) - \mathbb{P}_h(\widetilde{V}_{h+1})^2(s,a)\right|$$

$$\leq \underbrace{\left|\phi(s,a)^\top \bar{\Sigma}_h^{-1}\left(\sum_{\tau=1}^K \phi(\bar{s}_h^\tau, \bar{a}_h^\tau) \cdot \widetilde{V}_{h+1}(\bar{s}_{h+1}^\tau)^2\right) - \mathbb{P}_h(\widetilde{V}_{h+1})^2(s,a)\right|}_{(1)} + \underbrace{\left|\phi(s,a)^\top \bar{\Sigma}_h^{-1}\phi_1\right|}_{(2)}$$

$$+ \underbrace{\left|\phi(s,a)^\top(\widetilde{\Sigma}_h^{-1} - \bar{\Sigma}_h^{-1})\left(\sum_{\tau=1}^K \phi(\bar{s}_h^\tau, \bar{a}_h^\tau) \cdot \widetilde{V}_{h+1}(\bar{s}_{h+1}^\tau)^2 + \phi_1\right)\right|}_{(3)},$$

where $\bar{\Sigma}_h = \widetilde{\Sigma}_h - K_1 = \sum_{\tau=1}^K \phi(\bar{s}_h^\tau, \bar{a}_h^\tau)\phi(\bar{s}_h^\tau, \bar{a}_h^\tau)^\top + \lambda I$.

Similar to the proof in Lemma D.5, when $K \geq \max\{\frac{128\log\frac{2dH}{\delta}}{\kappa^2}, \frac{\sqrt{2}L}{H\sqrt{d\kappa}}\}$, it holds that with probability $1 - \delta$, for all $h \in [H]$,

$$\lambda_{\min}(\bar{\Sigma}_h) \geq \frac{K\kappa}{2}, \quad \lambda_{\min}(\widetilde{\Sigma}_h) \geq \frac{K\kappa}{2}, \quad \left\|\widetilde{\Sigma}_h^{-1} - \bar{\Sigma}_h^{-1}\right\|_2 \leq \frac{4E/\kappa^2}{K^2}.$$

(The only difference to Lemma D.5 is here $\mathbb{E}_{\mu,h}[\phi(s,a)\phi(s,a)^\top] \geq \kappa$.)

Under this high probability event, for term (2), it holds that for all $h, s, a \in [H] \times \mathcal{S} \times \mathcal{A}$,

$$\left|\phi(s,a)^\top \bar{\Sigma}_h^{-1}\phi_1\right| \leq \|\phi(s,a)\| \cdot \left\|\bar{\Sigma}_h^{-1}\right\| \cdot \|\phi_1\| \leq \lambda_{\min}^{-1}(\bar{\Sigma}_h) \cdot HL \leq \frac{2HL/\kappa}{K}. \qquad (45)$$

For term (3), similar to Lemma D.6, we have for all $h, s, a \in [H] \times \mathcal{S} \times \mathcal{A}$,

$$\left| \phi(s,a)^\top (\widetilde{\Sigma}_h^{-1} - \bar{\Sigma}_h^{-1}) \left( \sum_{\tau=1}^K \phi(\bar{s}_h^\tau, \bar{a}_h^\tau) \cdot \widetilde{V}_{h+1}(\bar{s}_{h+1}^\tau)^2 + \phi_1 \right) \right| \le \frac{4\sqrt{2}H^2 E\sqrt{d}/\kappa^{3/2}}{K}. \quad (46)$$

(The only difference to Lemma D.6 is that here $\widetilde{V}_{h+1}(s)^2 \le H^2$, $\|\phi_1\|_2 \le HL$, $\left\|\widetilde{\Sigma}_h^{-1}\right\|_2 \le \frac{2}{K\kappa}$ and $\left\|\widetilde{\Sigma}_h^{-1} - \bar{\Sigma}_h^{-1}\right\|_2 \le \frac{4E/\kappa^2}{K^2}$.)

We further decompose term (1) as below.

$$(1) = \left| \phi(s,a)^\top \bar{\Sigma}_h^{-1} \left( \sum_{\tau=1}^K \phi(\bar{s}_h^\tau, \bar{a}_h^\tau) \cdot \widetilde{V}_{h+1}(\bar{s}_{h+1}^\tau)^2 \right) - \mathbb{P}_h(\widetilde{V}_{h+1})^2(s,a) \right|$$

$$= \left| \phi(s,a)^\top \bar{\Sigma}_h^{-1} \sum_{\tau=1}^K \phi(\bar{s}_h^\tau, \bar{a}_h^\tau) \cdot \widetilde{V}_{h+1}(\bar{s}_{h+1}^\tau)^2 - \phi(s,a)^\top \bar{\Sigma}_h^{-1} \left( \sum_{\tau=1}^K \phi(\bar{s}_h^\tau, \bar{a}_h^\tau) \phi(\bar{s}_h^\tau, \bar{a}_h^\tau)^\top + \lambda I \right) \int_{\mathcal{S}} (\widetilde{V}_{h+1})^2(s') d\nu_h(s') \right|$$

$$\le \underbrace{\left| \phi(s,a)^\top \bar{\Sigma}_h^{-1} \sum_{\tau=1}^K \phi(\bar{s}_h^\tau, \bar{a}_h^\tau) \cdot \left( \widetilde{V}_{h+1}(\bar{s}_{h+1}^\tau)^2 - \mathbb{P}_h(\widetilde{V}_{h+1})^2(\bar{s}_h^\tau, \bar{a}_h^\tau) \right) \right|}_{(4)} + \lambda \underbrace{\left| \phi(s,a)^\top \bar{\Sigma}_h^{-1} \int_{\mathcal{S}} (\widetilde{V}_{h+1})^2(s') d\nu_h(s') \right|}_{(5)}.$$

$$(47)$$

For term (5), because $K \ge \max\left\{ \frac{512 \log\left(\frac{2Hd}{\delta}\right)}{\kappa^2}, \frac{4\lambda}{\kappa} \right\}$, by Lemma E.13 and a union bound, with probability $1 - \delta$, for all $h, s, a \in [H] \times \mathcal{S} \times \mathcal{A}$,

$$\lambda \left| \phi(s,a)^\top \bar{\Sigma}_h^{-1} \int_{\mathcal{S}} (\widetilde{V}_{h+1})^2(s') d\nu_h(s') \right| \le \lambda \|\phi(s,a)\|_{\bar{\Sigma}_h^{-1}} \left\| \int_{\mathcal{S}} (\widetilde{V}_{h+1})^2(s') d\nu_h(s') \right\|_{\bar{\Sigma}_h^{-1}}$$

$$\le \lambda \frac{2}{\sqrt{K}} \|\phi(s,a)\|_{(\Sigma_h^p)^{-1}} \frac{2}{\sqrt{K}} \left\| \int_{\mathcal{S}} (\widetilde{V}_{h+1})^2(s') d\nu_h(s') \right\|_{(\Sigma_h^p)^{-1}} \le 4\lambda \left\| (\Sigma_h^p)^{-1} \right\| \frac{H^2\sqrt{d}}{K} \le 4\lambda \frac{H^2\sqrt{d}}{\kappa K},$$

$$(48)$$

where $\Sigma_h^p = \mathbb{E}_{\mu,h}[\phi(s,a)\phi(s,a)^\top]$ and $\lambda_{\min}(\Sigma_h^p) \ge \kappa$.

For term (4), it can be bounded by the following inequality (because of Cauchy-Schwarz inequality).

$$(4) \le \|\phi(s,a)\|_{\bar{\Sigma}_h^{-1}} \cdot \left\| \sum_{\tau=1}^K \phi(\bar{s}_h^\tau, \bar{a}_h^\tau) \cdot \left( \widetilde{V}_{h+1}(\bar{s}_{h+1}^\tau)^2 - \mathbb{P}_h(\widetilde{V}_{h+1})^2(\bar{s}_h^\tau, \bar{a}_h^\tau) \right) \right\|_{\bar{\Sigma}_h^{-1}}. \quad (49)$$

**Bounding using covering.** Note for any fix $V_{h+1}$, we can define $x_\tau = \phi(\bar{s}_h^\tau, \bar{a}_h^\tau)$ ($\|\phi\|_2 \le 1$) and $\eta_\tau = V_{h+1}(\bar{s}_{h+1}^\tau)^2 - \mathbb{P}_h(V_{h+1})^2(\bar{s}_h^\tau, \bar{a}_h^\tau)$ is $H^2$-subgaussian, by Lemma E.9 (where $t = K$ and $L = 1$), it holds that with probability $1 - \delta$,

$$\left\| \sum_{\tau=1}^K \phi(\bar{s}_h^\tau, \bar{a}_h^\tau) \cdot \left( V_{h+1}(\bar{s}_{h+1}^\tau)^2 - \mathbb{P}_h(V_{h+1})^2(\bar{s}_h^\tau, \bar{a}_h^\tau) \right) \right\|_{\bar{\Sigma}_h^{-1}} \le \sqrt{8H^4 \cdot \frac{d}{2} \log\left( \frac{\lambda + K}{\lambda \delta} \right)}.$$

Let $\mathcal{N}_h(\epsilon)$ be the minimal $\epsilon$-cover (with respect to the supremum norm) of

$$\mathcal{V}_h := \left\{ V_h : V_h(\cdot) = \max_{a \in \mathcal{A}} \{ \min\{ \phi(s,a)^\top \theta - C_1 \sqrt{d \cdot \phi(\cdot,\cdot)^\top \widetilde{\Lambda}_h^{-1} \phi(\cdot,\cdot)} - C_2, H - h + 1 \}^+ \} \right\}.$$

That is, for any $V \in \mathcal{V}_h$, there exists a value function $V' \in \mathcal{N}_h(\epsilon)$ such that $\sup_{s \in \mathcal{S}} |V(s) - V'(s)| < \epsilon$. Now by a union bound, we obtain with probability $1 - \delta$,

$$\sup_{V_{h+1} \in \mathcal{N}_{h+1}(\epsilon)} \left\| \sum_{\tau=1}^K \phi(\bar{s}_h^\tau, \bar{a}_h^\tau) \cdot \left( V_{h+1}(\bar{s}_{h+1}^\tau)^2 - \mathbb{P}_h(V_{h+1})^2(\bar{s}_h^\tau, \bar{a}_h^\tau) \right) \right\|_{\bar{\Sigma}_h^{-1}} \le \sqrt{8H^4 \cdot \frac{d}{2} \log\left( \frac{\lambda + K}{\lambda \delta} |\mathcal{N}_{h+1}(\epsilon)| \right)}$$

which implies

$$\left\| \sum_{\tau=1}^{K} \phi(\bar{s}_h^\tau, \bar{a}_h^\tau) \cdot \left( \widetilde{V}_{h+1}(\bar{s}_{h+1}^\tau)^2 - \mathbb{P}_h(\widetilde{V}_{h+1})^2(\bar{s}_h^\tau, \bar{a}_h^\tau) \right) \right\|_{\bar{\Sigma}_h^{-1}}$$

$$\leq \sqrt{8H^4 \cdot \frac{d}{2} \log \left( \frac{\lambda + K}{\lambda \delta} |\mathcal{N}_{h+1}(\epsilon)| \right)} + 4H^2 \sqrt{\epsilon^2 K^2 / \lambda}$$

choosing $\epsilon = d\sqrt{\lambda}/K$, applying Lemma B.3 of [Jin et al., 2021][13] to the covering number $\mathcal{N}_{h+1}(\epsilon)$ w.r.t. $\mathcal{V}_{h+1}$, we can further bound above by

$$\leq \sqrt{8H^4 \cdot \frac{d^3}{2} \log \left( \frac{\lambda + K}{\lambda \delta} 2dHK \right)} + 4H^2 \sqrt{d^2} \leq 6\sqrt{H^4 \cdot d^3 \log \left( \frac{\lambda + K}{\lambda \delta} 2dHK \right)}$$

Apply a union bound for $h \in [H]$, we have with probability $1 - \delta$, for all $h \in [H]$,

$$\left\| \sum_{\tau=1}^{K} \phi(\bar{s}_h^\tau, \bar{a}_h^\tau) \cdot \left( \widetilde{V}_{h+1}(\bar{s}_{h+1}^\tau)^2 - \mathbb{P}_h(\widetilde{V}_{h+1})^2(\bar{s}_h^\tau, \bar{a}_h^\tau) \right) \right\|_{\bar{\Sigma}_h^{-1}} \leq 6\sqrt{H^4 d^3 \log \left( \frac{(\lambda + K)2KdH^2}{\lambda \delta} \right)} \tag{50}$$

and similar to term (2), it holds that for all $h, s, a \in [H] \times \mathcal{S} \times \mathcal{A}$,

$$\|\phi(s, a)\|_{\bar{\Sigma}_h^{-1}} \leq \sqrt{\|\bar{\Sigma}_h^{-1}\|} \leq \sqrt{\frac{2}{\kappa K}}. \tag{51}$$

Combining (45), (46), (47), (48), (49), (50), (51) and the assumption that $K \geq \max\{\frac{4L^2}{H^2 d^3 \kappa}, \frac{32E^2}{d^2 \kappa^2}, \frac{16\lambda^2}{d^2 \kappa}\}$, we obtain with probability $1 - 3\delta$ for all $h, s, a \in [H] \times \mathcal{S} \times \mathcal{A}$,

$$\left| \langle \phi(s, a), \widetilde{\beta}_h \rangle_{[0, (H-h+1)^2]} - \mathbb{P}_h(\widetilde{V}_{h+1})^2(s, a) \right| \leq 12\sqrt{\frac{H^4 d^3}{\kappa K} \log \left( \frac{(\lambda + K)2KdH^2}{\lambda \delta} \right)}.$$

**Step 2:** The second step is to show for all $h, s, a \in [H] \times \mathcal{S} \times \mathcal{A}$, with probability $1 - 3\delta$,

$$\left| \langle \phi(s, a), \widetilde{\theta}_h \rangle_{[0, H-h+1]} - \mathbb{P}_h(\widetilde{V}_{h+1})(s, a) \right| \leq 12\sqrt{\frac{H^2 d^3}{\kappa K} \log \left( \frac{(\lambda + K)2KdH^2}{\lambda \delta} \right)}. \tag{52}$$

The proof of Step 2 is nearly identical to Step 1 except $\widetilde{V}_h^2$ is replaced by $\widetilde{V}_h$.

**Step 3:** The last step is to prove $\sup_h \|\widetilde{\sigma}_h^2 - \sigma_{\widetilde{V}_{h+1}}^2\|_\infty \leq 36\sqrt{\frac{H^4 d^3}{\kappa K} \log \left( \frac{(\lambda + K)2KdH^2}{\lambda \delta} \right)}$ with high probability.

**Proof of Step 3.** By (52),

$$\left| \left[ \langle \phi(\cdot, \cdot), \widetilde{\theta}_h \rangle_{[0, H-h+1]} \right]^2 - \left[ \mathbb{P}_h(\widetilde{V}_{h+1})(s, a) \right]^2 \right|$$

$$= \left| \langle \phi(s, a), \widetilde{\theta}_h \rangle_{[0, H-h+1]} + \mathbb{P}_h(\widetilde{V}_{h+1})(s, a) \right| \cdot \left| \langle \phi(s, a), \widetilde{\theta}_h \rangle_{[0, H-h+1]} - \mathbb{P}_h(\widetilde{V}_{h+1})(s, a) \right|$$

$$\leq 2H \cdot \left| \langle \phi(s, a), \widetilde{\theta}_h \rangle_{[0, H-h+1]} - \mathbb{P}_h(\widetilde{V}_{h+1})(s, a) \right| \leq 24\sqrt{\frac{H^4 d^3}{\kappa K} \log \left( \frac{(\lambda + K)2KdH^2}{\lambda \delta} \right)}.$$

Combining this with Step 1, we have with probability $1 - 6\delta$, $\forall h, s, a \in [H] \times \mathcal{S} \times \mathcal{A}$,

$$\left| \widetilde{\text{Var}}_h \widetilde{V}_{h+1}(s, a) - \text{Var}_{P_h} \widetilde{V}_{h+1}(s, a) \right| \leq 36\sqrt{\frac{H^4 d^3}{\kappa K} \log \left( \frac{(\lambda + K)2KdH^2}{\lambda \delta} \right)}.$$

Finally, by the non-expansiveness of operator $\max\{1, \cdot\}$, the proof is complete. $\qquad \square$

---

[13]Note that the conclusion in [Jin et al., 2021] hold here even though we have an extra constant $C_2$.

### D.3.4 Validity of our pessimism

Recall the definition $\widehat{\Lambda}_h = \sum_{\tau=1}^K \phi(s_h^\tau, a_h^\tau) \phi(s_h^\tau, a_h^\tau)^\top / \widetilde{\sigma}_h^2(s_h^\tau, a_h^\tau) + \lambda \cdot I$ and
$\Lambda_h = \sum_{\tau=1}^K \phi(s_h^\tau, a_h^\tau) \phi(s_h^\tau, a_h^\tau)^\top / \sigma_{\widetilde{V}_{h+1}}^2(s_h^\tau, a_h^\tau) + \lambda I$. Then we have the following lemma to bound
the term $\sqrt{\phi(s,a)^\top \widehat{\Lambda}_h^{-1} \phi(s,a)}$ by $\sqrt{\phi(s,a)^\top \Lambda_h^{-1} \phi(s,a)}$.

**Lemma D.11** (Private version of lemma C.3 in [Yin et al., 2022]). *Denote the quantities* $C_1 = \max\{2\lambda, 128 \log(2dH/\delta), \frac{128 H^4 \log(2dH/\delta)}{\kappa^2}\}$ *and* $C_2 = \widetilde{O}(H^{12} d^3/\kappa^5)$. *Suppose the number of episode* $K$ *satisfies* $K > \max\{C_1, C_2\}$ *and the condition in Lemma D.10, under the high probability events in Lemma D.2 and Lemma D.10, it holds that with probability* $1 - 2\delta$, *for all* $h, s, a \in [H] \times \mathcal{S} \times \mathcal{A}$,

$$\sqrt{\phi(s,a)^\top \widehat{\Lambda}_h^{-1} \phi(s,a)} \le 2\sqrt{\phi(s,a)^\top \Lambda_h^{-1} \phi(s,a)}.$$

*Proof of Lemma D.11.* By definition $\sqrt{\phi(s,a)^\top \widehat{\Lambda}_h^{-1} \phi(s,a)} = \|\phi(s,a)\|_{\widehat{\Lambda}_h^{-1}}$. Then denote

$$\widehat{\Lambda}_h' = \frac{1}{K} \widehat{\Lambda}_h, \quad \Lambda_h' = \frac{1}{K} \Lambda_h,$$

where $\Lambda_h = \sum_{\tau=1}^K \phi(s_h^\tau, a_h^\tau) \phi(s_h^\tau, a_h^\tau)^\top / \sigma_{\widetilde{V}_{h+1}}^2(s_h^\tau, a_h^\tau) + \lambda I$. Under the assumption of $K$, by the conclusion in Lemma D.10, we have

$$
\begin{aligned}
\left\| \widehat{\Lambda}_h' - \Lambda_h' \right\| &\le \sup_{s,a} \left\| \frac{\phi(s,a)\phi(s,a)^\top}{\widetilde{\sigma}_h^2(s,a)} - \frac{\phi(s,a)\phi(s,a)^\top}{\sigma_{\widetilde{V}_{h+1}}^2(s,a)} \right\| \\
&\le \sup_{s,a} \left| \frac{\widetilde{\sigma}_h^2(s,a) - \sigma_{\widetilde{V}_{h+1}}^2(s,a)}{\widetilde{\sigma}_h^2(s,a) \cdot \sigma_{\widetilde{V}_{h+1}}^2(s,a)} \right| \cdot \|\phi(s,a)\|^2 \\
&\le \sup_{s,a} \left| \frac{\widetilde{\sigma}_h^2(s,a) - \sigma_{\widetilde{V}_{h+1}}^2(s,a)}{1} \right| \cdot 1 \\
&\le 36 \sqrt{\frac{H^4 d^3}{\kappa K} \log\left( \frac{(\lambda + K) 2K dH^2}{\lambda \delta} \right)}.
\end{aligned}
\tag{53}
$$

Next by Lemma E.12 (with $\phi$ to be $\phi/\sigma_{\widetilde{V}_{h+1}}$ and therefore $C = 1$) and a union bound, it holds with probability $1 - \delta$, for all $h \in [H]$,

$$\left\| \Lambda_h' - \left( \mathbb{E}_{\mu,h}[\phi(s,a)\phi(s,a)^\top / \sigma_{\widetilde{V}_{h+1}}^2(s,a)] + \frac{\lambda}{K} I_d \right) \right\| \le \frac{4\sqrt{2}}{\sqrt{K}} \left( \log \frac{2dH}{\delta} \right)^{1/2}.$$

Therefore by Weyl's inequality and the assumption that $K$ satisfies that $K > \max\{2\lambda, 128 \log(2dH/\delta), \frac{128 H^4 \log(2dH/\delta)}{\kappa^2}\}$, the above inequality leads to

$$
\begin{aligned}
\|\Lambda_h'\| &= \lambda_{\max}(\Lambda_h') \le \lambda_{\max}\left( \mathbb{E}_{\mu,h}[\phi(s,a)\phi(s,a)^\top / \sigma_{\widetilde{V}_{h+1}}^2(s,a)] \right) + \frac{\lambda}{K} + \frac{4\sqrt{2}}{\sqrt{K}} \left( \log \frac{2dH}{\delta} \right)^{1/2} \\
&= \left\| \mathbb{E}_{\mu,h}[\phi(s,a)\phi(s,a)^\top / \sigma_{\widetilde{V}_{h+1}}^2(s,a)] \right\|_2 + \frac{\lambda}{K} + \frac{4\sqrt{2}}{\sqrt{K}} \left( \log \frac{2dH}{\delta} \right)^{1/2} \\
&\le \|\phi(s,a)\|^2 + \frac{\lambda}{K} + \frac{4\sqrt{2}}{\sqrt{K}} \left( \log \frac{2dH}{\delta} \right)^{1/2} \le 1 + \frac{\lambda}{K} + \frac{4\sqrt{2}}{\sqrt{K}} \left( \log \frac{2dH}{\delta} \right)^{1/2} \le 2,
\end{aligned}
$$

$$
\begin{aligned}
\lambda_{\min}(\Lambda_h') &\ge \lambda_{\min}\left( \mathbb{E}_{\mu,h}[\phi(s,a)\phi(s,a)^\top / \sigma_{\widetilde{V}_{h+1}}^2(s,a)] \right) + \frac{\lambda}{K} - \frac{4\sqrt{2}}{\sqrt{K}} \left( \log \frac{2dH}{\delta} \right)^{1/2} \\
&\ge \lambda_{\min}\left( \mathbb{E}_{\mu,h}[\phi(s,a)\phi(s,a)^\top / \sigma_{\widetilde{V}_{h+1}}^2(s,a)] \right) - \frac{4\sqrt{2}}{\sqrt{K}} \left( \log \frac{2dH}{\delta} \right)^{1/2} \\
&\ge \frac{\kappa}{H^2} - \frac{4\sqrt{2}}{\sqrt{K}} \left( \log \frac{2dH}{\delta} \right)^{1/2} \ge \frac{\kappa}{2H^2}.
\end{aligned}
$$

Hence with probability $1 - \delta$, $\|\Lambda'_h\| \le 2$ and $\left\|\Lambda'^{-1}_h\right\| = \lambda^{-1}_{\min}(\Lambda'_h) \le \frac{2H^2}{\kappa}$. Similarly, one can show $\left\|\widehat{\Lambda}'^{-1}_h\right\| \le \frac{2H^2}{\kappa}$ with probability $1 - \delta$ using identical proof.

Now apply Lemma E.11 and a union bound to $\widehat{\Lambda}'_h$ and $\Lambda'_h$, we obtain with probability $1 - \delta$, for all $h, s, a \in [H] \times \mathcal{S} \times \mathcal{A}$,

$$
\begin{aligned}
\|\phi(s,a)\|_{\widehat{\Lambda}'^{-1}_h} &\le \left[1 + \sqrt{\left\|\Lambda'^{-1}_h\right\| \cdot \|\Lambda'_h\| \cdot \left\|\widehat{\Lambda}'^{-1}_h\right\| \cdot \left\|\widehat{\Lambda}'_h - \Lambda'_h\right\|}\right] \cdot \|\phi(s,a)\|_{\Lambda'^{-1}_h} \\
&\le \left[1 + \sqrt{\frac{2H^2}{\kappa} \cdot 2 \cdot \frac{2H^2}{\kappa} \cdot \left\|\widehat{\Lambda}'_h - \Lambda'_h\right\|}\right] \cdot \|\phi(s,a)\|_{\Lambda'^{-1}_h} \\
&\le \left[1 + \sqrt{\frac{288H^4}{\kappa^2}\left(\sqrt{\frac{H^4 d^3}{\kappa K} \log\left(\frac{(\lambda + K)2KdH^2}{\lambda\delta}\right)}\right)}\right] \cdot \|\phi(s,a)\|_{\Lambda'^{-1}_h} \\
&\le 2\|\phi(s,a)\|_{\Lambda'^{-1}_h}
\end{aligned}
$$

where the third inequality uses (53) and the last inequality uses $K > \widetilde{O}(H^{12}d^3/\kappa^5)$. Note the conclusion can be derived directly by the above inequality multiplying $1/\sqrt{K}$ on both sides. $\qquad \square$

In order to bound $\left\|\sum_{\tau=1}^{K} x_\tau \eta_\tau\right\|_{\widehat{\Lambda}^{-1}_h}$, we apply the following Lemma D.12.

**Lemma D.12** (Lemma C.4 in [Yin et al., 2022]). *Recall* $x_\tau = \frac{\phi(s^\tau_h, a^\tau_h)}{\widetilde{\sigma}_h(s^\tau_h, a^\tau_h)}$ *and*
$\eta_\tau = \left(r^\tau_h + \widetilde{V}_{h+1}\left(s^\tau_{h+1}\right) - \left(\mathcal{T}_h \widetilde{V}_{h+1}\right)(s^\tau_h, a^\tau_h)\right) / \widetilde{\sigma}_h(s^\tau_h, a^\tau_h)$. *Denote*

$$
\xi := \sup_{V \in [0,H],\, s' \sim P_h(s,a),\, h \in [H]} \left|\frac{r_h + V(s') - (\mathcal{T}_h V)(s,a)}{\sigma_V(s,a)}\right|.
$$

*Suppose* $K \ge \widetilde{O}(H^{12}d^3/\kappa^5)$[14], *then with probability* $1 - \delta$,

$$
\left\|\sum_{\tau=1}^{K} x_\tau \eta_\tau\right\|_{\widehat{\Lambda}^{-1}_h} \le \widetilde{O}\left(\max\left\{\sqrt{d}, \xi\right\}\right),
$$

*where* $\widetilde{O}$ *absorbs constants and Polylog terms.*

Now we are ready to prove the following key lemma, which gives a high probability bound for $\left|(\mathcal{T}_h \widetilde{V}_{h+1} - \widetilde{\mathcal{T}}_h \widetilde{V}_{h+1})(s,a)\right|$.

**Lemma D.13.** *Assume* $K > \max\{\mathcal{M}_1, \mathcal{M}_2, \mathcal{M}_3, \mathcal{M}_4\}$, *for any* $0 < \lambda < \kappa$, *suppose* $\sqrt{d} > \xi$, *where* $\xi := \sup_{V \in [0,H],\, s' \sim P_h(s,a),\, h \in [H]} \left|\frac{r_h + V(s') - (\mathcal{T}_h V)(s,a)}{\sigma_V(s,a)}\right|$. *Then with probability* $1 - \delta$, *for all* $h, s, a \in [H] \times \mathcal{S} \times \mathcal{A}$,

$$
\left|(\mathcal{T}_h \widetilde{V}_{h+1} - \widetilde{\mathcal{T}}_h \widetilde{V}_{h+1})(s,a)\right| \le \widetilde{O}\left(\sqrt{d}\sqrt{\phi(s,a)^\top \widetilde{\Lambda}^{-1}_h \phi(s,a)}\right) + \frac{D}{K},
$$

*where* $\widetilde{\Lambda}_h = \sum_{\tau=1}^{K} \phi(s^\tau_h, a^\tau_h)\phi(s^\tau_h, a^\tau_h)^\top / \widetilde{\sigma}^2_h(s^\tau_h, a^\tau_h) + \lambda I + K_2$,

$$
D = \widetilde{O}\left(\frac{H^2 L}{\kappa} + \frac{H^4 E\sqrt{d}}{\kappa^{3/2}} + H^3\sqrt{d} + \frac{H^2\sqrt{Ed}}{\kappa}\right) = \widetilde{O}\left(\frac{H^2 L}{\kappa} + \frac{H^4 E\sqrt{d}}{\kappa^{3/2}} + H^3\sqrt{d}\right)
$$

*and* $\widetilde{O}$ *absorbs constants and Polylog terms.*

---

[14]Note that here the assumption is stronger than the assumption in [Yin et al., 2022], therefore the conclusion of Lemma C.4 holds.

*Proof of Lemma D.13.* The proof is by combining (40), (42), Lemma D.4, Lemma D.6, Lemma D.7, Lemma D.8, Lemma D.12 and a union bound. $\qquad\square$

**Remark D.14.** *Under the same assumption of Lemma D.13, because of Remark D.9 and Lemma D.11, we have with probability $1 - \delta$, for all $h, s, a \in [H] \times \mathcal{S} \times \mathcal{A}$,*

$$
\begin{aligned}
\left| (\mathcal{T}_h \widetilde{V}_{h+1} - \widetilde{\mathcal{T}}_h \widetilde{V}_{h+1})(s,a) \right| &\leq \widetilde{O}\left( \sqrt{d}\sqrt{\phi(s,a)^\top \widetilde{\Lambda}_h^{-1} \phi(s,a)} \right) + \frac{D}{K} \\
&\leq \widetilde{O}\left( \sqrt{d}\sqrt{\phi(s,a)^\top \widehat{\Lambda}_h^{-1} \phi(s,a)} \right) + \frac{2D}{K} \\
&\leq \widetilde{O}\left( 2\sqrt{d}\sqrt{\phi(s,a)^\top \Lambda_h^{-1} \phi(s,a)} \right) + \frac{2D}{K}.
\end{aligned}
\tag{54}
$$

*Because $D = \widetilde{O}\left( \frac{H^2 L}{\kappa} + \frac{H^4 E \sqrt{d}}{\kappa^{3/2}} + H^3 \sqrt{d} \right)$ and $\widetilde{O}$ absorbs constant, we will write as below for simplicity:*

$$
\left| (\mathcal{T}_h \widetilde{V}_{h+1} - \widetilde{\mathcal{T}}_h \widetilde{V}_{h+1})(s,a) \right| \leq \widetilde{O}\left( \sqrt{d}\sqrt{\phi(s,a)^\top \Lambda_h^{-1} \phi(s,a)} \right) + \frac{D}{K}.
\tag{55}
$$

### D.3.5 Finalize the proof of the first part

We are ready to prove the first part of Theorem 4.1.

**Theorem D.15** (First part of Theorem 4.1)**.** *Let $K$ be the number of episodes. Suppose $\sqrt{d} > \xi$, where $\xi := \sup_{V \in [0,H], \, s' \sim P_h(s,a), \, h \in [H]} \left| \frac{r_h + V(s') - (\mathcal{T}_h V)(s,a)}{\sigma_V(s,a)} \right|$ and $K > \max\{\mathcal{M}_1, \mathcal{M}_2, \mathcal{M}_3, \mathcal{M}_4\}$. Then for any $0 < \lambda < \kappa$, with probability $1 - \delta$, for all policy $\pi$ simultaneously, the output $\widehat{\pi}$ of Algorithm 2 satisfies*

$$
v^\pi - v^{\widehat{\pi}} \leq \widetilde{O}\left( \sqrt{d} \cdot \sum_{h=1}^H \mathbb{E}_\pi \left[ \left( \phi(\cdot, \cdot)^\top \Lambda_h^{-1} \phi(\cdot, \cdot) \right)^{1/2} \right] \right) + \frac{DH}{K},
$$

*where $\Lambda_h = \sum_{\tau=1}^K \frac{\phi(s_h^\tau, a_h^\tau) \cdot \phi(s_h^\tau, a_h^\tau)^\top}{\sigma_{\widetilde{V}_{h+1}(s_h^\tau, a_h^\tau)}^2} + \lambda I_d$, $D = \widetilde{O}\left( \frac{H^2 L}{\kappa} + \frac{H^4 E \sqrt{d}}{\kappa^{3/2}} + H^3 \sqrt{d} \right)$ and $\widetilde{O}$ absorbs constants and Polylog terms.*

*Proof of Theorem D.15.* Combining Lemma D.3 and Remark D.14, we have with probability $1 - \delta$, for all policy $\pi$ simultaneously,

$$
V_1^\pi(s) - V_1^{\widehat{\pi}}(s) \leq \widetilde{O}\left( \sqrt{d} \cdot \sum_{h=1}^H \mathbb{E}_\pi \left[ \left( \phi(\cdot, \cdot)^\top \Lambda_h^{-1} \phi(\cdot, \cdot) \right)^{1/2} \Big| s_1 = s \right] \right) + \frac{DH}{K},
\tag{56}
$$

now the proof is complete by taking the initial distribution $d_1$ on both sides. $\qquad\square$

### D.3.6 Finalize the proof of the second part

To prove the second part of Theorem 4.1, we begin with a crude bound on $\sup_h \left\| V_h^\star - \widetilde{V}_h \right\|_\infty$.

**Lemma D.16** (Private version of Lemma C.8 in [Yin et al., 2022])**.** *Suppose $K \geq \max\{\mathcal{M}_1, \mathcal{M}_2, \mathcal{M}_3, \mathcal{M}_4\}$, under the high probability event in Lemma D.13, with probability at least $1 - \delta$,*

$$
\sup_h \left\| V_h^\star - \widetilde{V}_h \right\|_\infty \leq \widetilde{O}\left( \frac{H^2 \sqrt{d}}{\sqrt{\kappa K}} \right).
$$

*Proof of Lemma D.16.* **Step 1:** The first step is to show with probability at least $1 - \delta$, $\sup_h \left\| V_h^\star - V_h^{\widehat{\pi}} \right\|_\infty \leq \widetilde{O}\left( \frac{H^2 \sqrt{d}}{\sqrt{\kappa K}} \right)$.

Indeed, combine Lemma D.3 and Lemma D.13, similar to the proof of Theorem D.15, we directly have with probability $1 - \delta$, for all policy $\pi$ simultaneously, and for all $s \in \mathcal{S}, h \in [H]$,

$$V_h^\pi(s) - V_h^{\widehat{\pi}}(s) \leq \widetilde{O}\left(\sqrt{d} \cdot \sum_{t=h}^H \mathbb{E}_\pi\left[\left(\phi(\cdot,\cdot)^\top \Lambda_t^{-1}\phi(\cdot,\cdot)\right)^{1/2}\Big| s_h = s\right]\right) + \frac{DH}{K}, \qquad (57)$$

Next, since $K \geq \max\left\{\frac{512\log\left(\frac{2Hd}{\delta}\right)}{\kappa^2}, \frac{4\lambda}{\kappa}\right\}$, by Lemma E.13 and a union bound over $h \in [H]$, with probability $1 - \delta$,

$$\sup_{s,a} \|\phi(s,a)\|_{\Lambda_h^{-1}} \leq \frac{2}{\sqrt{K}}\sup_{s,a}\|\phi(s,a)\|_{(\Lambda_h^p)^{-1}} \leq \frac{2}{\sqrt{K}}\sqrt{\lambda_{\min}^{-1}(\Lambda_h^p)} \leq \frac{2H}{\sqrt{\kappa K}}, \quad \forall h \in [H],$$

where $\Lambda_h^p = \mathbb{E}_{\mu,h}[\sigma_{\widetilde{V}_{h+1}}^{-2}(s,a)\phi(s,a)\phi(s,a)^\top]$ and $\lambda_{\min}(\Lambda_h^p) \geq \frac{\kappa}{H^2}$.

Lastly, taking $\pi = \pi^\star$ in (57) to obtain

$$\begin{aligned}
0 \leq V_h^{\pi^\star}(s) - V_h^{\widehat{\pi}}(s) \leq & \widetilde{O}\left(\sqrt{d} \cdot \sum_{t=h}^H \mathbb{E}_{\pi^\star}\left[\left(\phi(\cdot,\cdot)^\top \Lambda_t^{-1}\phi(\cdot,\cdot)\right)^{1/2}\Big| s_h = s\right]\right) + \frac{DH}{K} \\
\leq & \widetilde{O}\left(\frac{H^2\sqrt{d}}{\sqrt{\kappa K}}\right) + \widetilde{O}\left(\frac{H^3 L/\kappa}{K} + \frac{H^5 E\sqrt{d}/\kappa^{3/2}}{K} + \frac{H^4\sqrt{d}}{K}\right).
\end{aligned} \qquad (58)$$

This implies by using the condition $K > \max\{\frac{H^2 L^2}{d\kappa}, \frac{H^6 E^2}{\kappa^2}, H^4\kappa\}$, we finish the proof of Step 1.

**Step 2:** The second step is to show with probability $1 - \delta$, $\sup_h \left\|\widetilde{V}_h - V_h^{\widehat{\pi}}\right\|_\infty \leq \widetilde{O}\left(\frac{H^2\sqrt{d}}{\sqrt{\kappa K}}\right)$.

Indeed, applying Lemma E.7 with $\pi = \pi' = \widehat{\pi}$, then with probability $1 - \delta$, for all $s, h$

$$\begin{aligned}
\left|\widetilde{V}_h(s) - V_h^{\widehat{\pi}}(s)\right| &= \left|\sum_{t=h}^H \mathbb{E}_{\widehat{\pi}}\left[\widehat{Q}_h(s_h, a_h) - \left(\mathcal{T}_h\widetilde{V}_{h+1}\right)(s_h, a_h)\Big| s_h = s\right]\right| \\
&\leq \sum_{t=h}^H \left\|(\widetilde{\mathcal{T}}_h\widetilde{V}_{h+1} - \mathcal{T}_h\widetilde{V}_{h+1})(s,a)\right\|_\infty + H \cdot \|\Gamma_h(s,a)\|_\infty \\
&\leq \widetilde{O}\left(H\sqrt{d}\left\|\sqrt{\phi(s,a)^\top \Lambda_h^{-1}\phi(s,a)}\right\|_\infty\right) + \widetilde{O}\left(\frac{DH}{K}\right) \\
&\leq \widetilde{O}\left(\frac{H^2\sqrt{d}}{\sqrt{\kappa K}}\right),
\end{aligned}$$

where the second inequality uses Lemma D.13, Remark D.14 and the last inequality holds due to the same reason as Step 1.

**Step 3:** The proof of the lemma is complete by combining Step 1, Step 2, triangular inequality and a union bound.

$\square$

Then we can give a high probability bound of $\sup_h\|\sigma_{\widetilde{V}_{h+1}}^2 - \sigma_h^{\star 2}\|_\infty$.

**Lemma D.17** (Private version of Lemma C.10 in [Yin et al., 2022]). *Recall* $\sigma_{\widetilde{V}_{h+1}}^2 = \max\left\{1, \mathrm{Var}_{P_h}\widetilde{V}_{h+1}\right\}$ *and* $\sigma_h^{\star 2} = \max\left\{1, \mathrm{Var}_{P_h}V_{h+1}^\star\right\}$. *Suppose* $K \geq \max\{\mathcal{M}_1, \mathcal{M}_2, \mathcal{M}_3, \mathcal{M}_4\}$, *then with probability* $1 - \delta$,

$$\sup_h\|\sigma_{\widetilde{V}_{h+1}}^2 - \sigma_h^{\star 2}\|_\infty \leq \widetilde{O}\left(\frac{H^3\sqrt{d}}{\sqrt{\kappa K}}\right).$$

*Proof of Lemma D.17.* By definition and the non-expansiveness of $\max\{1, \cdot\}$, we have

$$
\begin{aligned}
\left\| \sigma^2_{\widetilde{V}_{h+1}} - \sigma^{\star 2}_h \right\|_\infty &\leq \left\| \mathrm{Var}\widetilde{V}_{h+1} - \mathrm{Var}V^\star_{h+1} \right\|_\infty \\
&\leq \left\| \mathbb{P}_h\left(\widetilde{V}^2_{h+1} - V^{\star 2}_{h+1}\right) \right\|_\infty + \left\| (\mathbb{P}_h\widetilde{V}_{h+1})^2 - (\mathbb{P}_h V^\star_{h+1})^2 \right\|_\infty \\
&\leq \left\| \widetilde{V}^2_{h+1} - V^{\star 2}_{h+1} \right\|_\infty + \left\| (\mathbb{P}_h\widetilde{V}_{h+1} + \mathbb{P}_h V^\star_{h+1})(\mathbb{P}_h\widetilde{V}_{h+1} - \mathbb{P}_h V^\star_{h+1}) \right\|_\infty \\
&\leq 2H \left\| \widetilde{V}_{h+1} - V^\star_{h+1} \right\|_\infty + 2H \left\| \mathbb{P}_h\widetilde{V}_{h+1} - \mathbb{P}_h V^\star_{h+1} \right\|_\infty \\
&\leq \widetilde{O}\left( \frac{H^3\sqrt{d}}{\sqrt{\kappa K}} \right).
\end{aligned}
$$

The second inequality is because of the definition of variance. The last inequality comes from Lemma D.16. $\qquad\square$

We transfer $\sqrt{\phi(s,a)^\top \Lambda^{-1}_h \phi(s,a)}$ to $\sqrt{\phi(s,a)^\top \Lambda^{\star-1}_h \phi(s,a)}$ by the following Lemma D.18.

**Lemma D.18** (Private version of Lemma C.11 in [Yin et al., 2022])**.** *Suppose* $K \geq \max\{\mathcal{M}_1, \mathcal{M}_2, \mathcal{M}_3, \mathcal{M}_4\}$, *then with probability* $1 - \delta$,

$$
\sqrt{\phi(s,a)^\top \Lambda^{-1}_h \phi(s,a)} \leq 2\sqrt{\phi(s,a)^\top \Lambda^{\star-1}_h \phi(s,a)}, \quad \forall h, s, a \in [H] \times \mathcal{S} \times \mathcal{A},
$$

*Proof of Lemma D.18.* By definition $\sqrt{\phi(s,a)^\top \Lambda^{-1}_h \phi(s,a)} = \|\phi(s,a)\|_{\Lambda^{-1}_h}$. Then denote

$$
\Lambda'_h = \frac{1}{K}\Lambda_h, \quad \Lambda^{\star'}_h = \frac{1}{K}\Lambda^\star_h,
$$

where $\Lambda^\star_h = \sum_{\tau=1}^K \phi(s^\tau_h, a^\tau_h)\phi(s^\tau_h, a^\tau_h)^\top / \sigma^2_{V^\star_{h+1}}(s^\tau_h, a^\tau_h) + \lambda I$. Under the condition of $K$, by Lemma D.17, with probability $1 - \delta$, for all $h \in [H]$,

$$
\begin{aligned}
\left\| \Lambda^{\star'}_h - \Lambda'_h \right\| &\leq \sup_{s,a} \left\| \frac{\phi(s,a)\phi(s,a)^\top}{\sigma^{\star 2}_h(s,a)} - \frac{\phi(s,a)\phi(s,a)^\top}{\sigma^2_{\widetilde{V}_{h+1}}(s,a)} \right\| \\
&\leq \sup_{s,a} \left| \frac{\sigma^{\star 2}_h(s,a) - \sigma^2_{\widetilde{V}_{h+1}}(s,a)}{\sigma^{\star 2}_h(s,a) \cdot \sigma^2_{\widetilde{V}_{h+1}}(s,a)} \right| \cdot \|\phi(s,a)\|^2 \\
&\leq \sup_{s,a} \left| \frac{\sigma^{\star 2}_h(s,a) - \sigma^2_{\widetilde{V}_{h+1}}(s,a)}{1} \right| \cdot 1 \\
&\leq \widetilde{O}\left( \frac{H^3\sqrt{d}}{\sqrt{\kappa K}} \right).
\end{aligned} \tag{59}
$$

Next by Lemma E.12 (with $\phi$ to be $\phi/\sigma_{V^\star_{h+1}}$ and $C = 1$), it holds with probability $1 - \delta$,

$$
\left\| \Lambda^{\star'}_h - \left( \mathbb{E}_{\mu,h}[\phi(s,a)\phi(s,a)^\top / \sigma^2_{V^\star_{h+1}}(s,a)] + \frac{\lambda}{K}I_d \right) \right\| \leq \frac{4\sqrt{2}}{\sqrt{K}} \left( \log\frac{2dH}{\delta} \right)^{1/2}.
$$

Therefore by Weyl's inequality and the condition $K > \max\{2\lambda, 128\log\left(\frac{2dH}{\delta}\right), \frac{128H^4\log(2dH/\delta)}{\kappa^2}\}$, the above inequality implies

$$\left\|\Lambda_h^{\star'}\right\| = \lambda_{\max}(\Lambda_h^{\star'}) \leq \lambda_{\max}\left(\mathbb{E}_{\mu,h}[\phi(s,a)\phi(s,a)^\top/\sigma_{V_{h+1}^\star}^2(s,a)]\right) + \frac{\lambda}{K} + \frac{4\sqrt{2}}{\sqrt{K}}\left(\log\frac{2dH}{\delta}\right)^{1/2}$$

$$\leq \left\|\mathbb{E}_{\mu,h}[\phi(s,a)\phi(s,a)^\top/\sigma_{V_{h+1}^\star}^2(s,a)]\right\| + \frac{\lambda}{K} + \frac{4\sqrt{2}}{\sqrt{K}}\left(\log\frac{2dH}{\delta}\right)^{1/2}$$

$$\leq \|\phi(s,a)\|^2 + \frac{\lambda}{K} + \frac{4\sqrt{2}}{\sqrt{K}}\left(\log\frac{2dH}{\delta}\right)^{1/2} \leq 1 + \frac{\lambda}{K} + \frac{4\sqrt{2}}{\sqrt{K}}\left(\log\frac{2dH}{\delta}\right)^{1/2} \leq 2,$$

$$\lambda_{\min}(\Lambda_h^{\star'}) \geq \lambda_{\min}\left(\mathbb{E}_{\mu,h}[\phi(s,a)\phi(s,a)^\top/\sigma_{V_{h+1}^\star}^2(s,a)]\right) + \frac{\lambda}{K} - \frac{4\sqrt{2}}{\sqrt{K}}\left(\log\frac{2dH}{\delta}\right)^{1/2}$$

$$\geq \lambda_{\min}\left(\mathbb{E}_{\mu,h}[\phi(s,a)\phi(s,a)^\top/\sigma_{V_{h+1}^\star}^2(s,a)]\right) - \frac{4\sqrt{2}}{\sqrt{K}}\left(\log\frac{2dH}{\delta}\right)^{1/2}$$

$$\geq \frac{\kappa}{H^2} - \frac{4\sqrt{2}}{\sqrt{K}}\left(\log\frac{2dH}{\delta}\right)^{1/2} \geq \frac{\kappa}{2H^2}.$$

Hence with probability $1 - \delta$, $\left\|\Lambda_h^{\star'}\right\| \leq 2$ and $\left\|\Lambda_h^{\star'-1}\right\| = \lambda_{\min}^{-1}(\Lambda_h^{\star'}) \leq \frac{2H^2}{\kappa}$. Similarly, we can show that $\left\|\Lambda_h'^{-1}\right\| \leq \frac{2H^2}{\kappa}$ holds with probability $1 - \delta$ by using identical proof.

Now apply Lemma E.11 and a union bound to $\Lambda_h^{\star'}$ and $\Lambda_h'$, we obtain with probability $1 - \delta$, for all $h, s, a \in [H] \times \mathcal{S} \times \mathcal{A}$,

$$\|\phi(s,a)\|_{\Lambda_h'^{-1}} \leq \left[1 + \sqrt{\left\|\Lambda_h^{\star'-1}\right\| \cdot \left\|\Lambda_h^{\star'}\right\| \cdot \left\|\Lambda_h'^{-1}\right\| \cdot \left\|\Lambda_h^{\star'} - \Lambda_h'\right\|}\right] \cdot \|\phi(s,a)\|_{\Lambda_h^{\star'-1}}$$

$$\leq \left[1 + \sqrt{\frac{2H^2}{\kappa} \cdot 2 \cdot \frac{2H^2}{\kappa} \cdot \left\|\Lambda_h^{\star'} - \Lambda_h'\right\|}\right] \cdot \|\phi(s,a)\|_{\Lambda_h^{\star'-1}}$$

$$\leq \left[1 + \sqrt{\frac{H^4}{\kappa^2}\left[\widetilde{O}\left(\frac{H^3\sqrt{d}}{\sqrt{\kappa K}}\right)\right]}\right] \cdot \|\phi(s,a)\|_{\Lambda_h^{\star'-1}}$$

$$\leq 2\|\phi(s,a)\|_{\Lambda_h^{\star'-1}}$$

where the third inequality uses (59) and the last inequality uses $K \geq \widetilde{O}(H^{14}d/\kappa^5)$. The conclusion can be derived directly by the above inequality multiplying $1/\sqrt{K}$ on both sides. $\square$

Finally, the second part of Theorem 4.1 can be proven by combining Theorem D.15 (with $\pi = \pi^\star$) and Lemma D.18.

### D.4 Put everything toghther

Combining Lemma D.1, Theorem D.15, and the discussion above, the proof of Theorem 4.1 is complete.

## E Assisting technical lemmas

**Lemma E.1** (Multiplicative Chernoff bound [Chernoff et al., 1952]). *Let $X$ be a Binomial random variable with parameter $p, n$. For any $1 \geq \theta > 0$, we have that*

$$\mathbb{P}[X < (1 - \theta)pn] < e^{-\frac{\theta^2 pn}{2}}, \quad \text{and} \quad \mathbb{P}[X \geq (1 + \theta)pn] < e^{-\frac{\theta^2 pn}{3}}$$

**Lemma E.2** (Hoeffding's Inequality [Sridharan, 2002]). *Let $x_1, ..., x_n$ be independent bounded random variables such that $\mathbb{E}[x_i] = 0$ and $|x_i| \leq \xi_i$ with probability 1. Then for any $\epsilon > 0$ we have*

$$\mathbb{P}\left(\frac{1}{n}\sum_{i=1}^{n} x_i \geq \epsilon\right) \leq e^{-\frac{2n^2\epsilon^2}{\sum_{i=1}^{n}\xi_i^2}}.$$

**Lemma E.3** (Bernstein's Inequality). *Let $x_1, ..., x_n$ be independent bounded random variables such that $\mathbb{E}[x_i] = 0$ and $|x_i| \le \xi$ with probability 1. Let $\sigma^2 = \frac{1}{n} \sum_{i=1}^n \text{Var}[x_i]$, then with probability $1 - \delta$ we have*

$$\frac{1}{n} \sum_{i=1}^n x_i \le \sqrt{\frac{2\sigma^2 \cdot \log(1/\delta)}{n}} + \frac{2\xi}{3n} \log(1/\delta).$$

**Lemma E.4** (Empirical Bernstein's Inequality [Maurer and Pontil, 2009]). *Let $x_1, ..., x_n$ be i.i.d random variables such that $|x_i| \le \xi$ with probability 1. Let $\bar{x} = \frac{1}{n} \sum_{i=1}^n x_i$ and $\widehat{V}_n = \frac{1}{n} \sum_{i=1}^n (x_i - \bar{x})^2$, then with probability $1 - \delta$ we have*

$$\left| \frac{1}{n} \sum_{i=1}^n x_i - \mathbb{E}[x] \right| \le \sqrt{\frac{2\widehat{V}_n \cdot \log(2/\delta)}{n}} + \frac{7\xi}{3n} \log(2/\delta).$$

**Lemma E.5** (Lemma I.8 in [Yin and Wang, 2021b]). *Let $n \ge 2$ and $V \in \mathbb{R}^S$ be any function with $||V||_\infty \le H$, $P$ be any $S$-dimensional distribution and $\widehat{P}$ be its empirical version using $n$ samples. Then with probability $1 - \delta$,*

$$\left| \sqrt{\text{Var}_{\widehat{P}}(V)} - \sqrt{\frac{n-1}{n} \text{Var}_P(V)} \right| \le 2H \sqrt{\frac{\log(2/\delta)}{n-1}}.$$

**Lemma E.6** (Claim 2 in [Vietri et al., 2020]). *Let $y \in \mathbb{R}$ be any positive real number. Then for all $x \in \mathbb{R}$ with $x \ge 2y$, it holds that $\frac{1}{x-y} \le \frac{1}{x} + \frac{2y}{x^2}$.*

### E.1 Extended Value Difference

**Lemma E.7** (Extended Value Difference (Section B.1 in [Cai et al., 2020])). *Let $\pi = \{\pi_h\}_{h=1}^H$ and $\pi' = \{\pi'_h\}_{h=1}^H$ be two arbitrary policies and let $\{\widehat{Q}_h\}_{h=1}^H$ be any given Q-functions. Then define $\widehat{V}_h(s) := \langle \widehat{Q}_h(s, \cdot), \pi_h(\cdot \mid s) \rangle$ for all $s \in \mathcal{S}$. Then for all $s \in \mathcal{S}$,*

$$
\begin{aligned}
\widehat{V}_1(s) - V_1^{\pi'}(s) &= \sum_{h=1}^H \mathbb{E}_{\pi'} \left[ \langle \widehat{Q}_h(s_h, \cdot), \pi_h(\cdot \mid s_h) - \pi'_h(\cdot \mid s_h) \rangle \mid s_1 = s \right] \\
&+ \sum_{h=1}^H \mathbb{E}_{\pi'} \left[ \widehat{Q}_h(s_h, a_h) - \left( \mathcal{T}_h \widehat{V}_{h+1} \right)(s_h, a_h) \mid s_1 = s \right]
\end{aligned}
\tag{60}
$$

*where $(\mathcal{T}_h V)(\cdot, \cdot) := r_h(\cdot, \cdot) + (P_h V)(\cdot, \cdot)$ for any $V \in \mathbb{R}^S$.*

**Lemma E.8** (Lemma I.10 in [Yin and Wang, 2021b]). *Let $\widehat{\pi} = \{\widehat{\pi}_h\}_{h=1}^H$ and $\widehat{Q}_h(\cdot, \cdot)$ be the arbitrary policy and Q-function and also $\widehat{V}_h(s) = \langle \widehat{Q}_h(s, \cdot), \widehat{\pi}_h(\cdot | s) \rangle \; \forall s \in \mathcal{S}$, and $\xi_h(s, a) = (\mathcal{T}_h \widehat{V}_{h+1})(s, a) - \widehat{Q}_h(s, a)$ element-wisely. Then for any arbitrary $\pi$, we have*

$$
\begin{aligned}
V_1^\pi(s) - V_1^{\widehat{\pi}}(s) &= \sum_{h=1}^H \mathbb{E}_\pi [\xi_h(s_h, a_h) \mid s_1 = s] - \sum_{h=1}^H \mathbb{E}_{\widehat{\pi}} [\xi_h(s_h, a_h) \mid s_1 = s] \\
&+ \sum_{h=1}^H \mathbb{E}_\pi \left[ \langle \widehat{Q}_h(s_h, \cdot), \pi_h(\cdot | s_h) - \widehat{\pi}_h(\cdot | s_h) \rangle \mid s_1 = s \right]
\end{aligned}
$$

*where the expectation are taken over $s_h, a_h$.*

### E.2 Assisting lemmas for linear MDP setting

**Lemma E.9** (Hoeffding inequality for self-normalized martingales [Abbasi-Yadkori et al., 2011]). *Let $\{\eta_t\}_{t=1}^\infty$ be a real-valued stochastic process. Let $\{\mathcal{F}_t\}_{t=0}^\infty$ be a filtration, such that $\eta_t$ is $\mathcal{F}_t$-measurable. Assume $\eta_t$ also satisfies $\eta_t$ given $\mathcal{F}_{t-1}$ is zero-mean and $R$-subgaussian, i.e.*

$$\forall \lambda \in \mathbb{R}, \quad \mathbb{E}\left[ e^{\lambda \eta_t} \mid \mathcal{F}_{t-1} \right] \le e^{\lambda^2 R^2 / 2}.$$

Let $\{x_t\}_{t=1}^{\infty}$ be an $\mathbb{R}^d$-valued stochastic process where $x_t$ is $\mathcal{F}_{t-1}$ measurable and $\|x_t\| \leq L$. Let $\Lambda_t = \lambda I_d + \sum_{s=1}^{t} x_s x_s^{\top}$. Then for any $\delta > 0$, with probability $1 - \delta$, for all $t > 0$,

$$\left\| \sum_{s=1}^{t} x_s \eta_s \right\|_{\Lambda_t^{-1}}^2 \leq 8R^2 \cdot \frac{d}{2} \log \left( \frac{\lambda + tL}{\lambda \delta} \right).$$

**Lemma E.10** (Bernstein inequality for self-normalized martingales [Zhou et al., 2021]). *Let $\{\eta_t\}_{t=1}^{\infty}$ be a real-valued stochastic process. Let $\{\mathcal{F}_t\}_{t=0}^{\infty}$ be a filtration, such that $\eta_t$ is $\mathcal{F}_t$-measurable. Assume $\eta_t$ also satisfies*

$$|\eta_t| \leq R, \mathbb{E}\left[\eta_t \mid \mathcal{F}_{t-1}\right] = 0, \mathbb{E}\left[\eta_t^2 \mid \mathcal{F}_{t-1}\right] \leq \sigma^2.$$

*Let $\{x_t\}_{t=1}^{\infty}$ be an $\mathbb{R}^d$-valued stochastic process where $x_t$ is $\mathcal{F}_{t-1}$ measurable and $\|x_t\| \leq L$. Let $\Lambda_t = \lambda I_d + \sum_{s=1}^{t} x_s x_s^{\top}$. Then for any $\delta > 0$, with probability $1 - \delta$, for all $t > 0$,*

$$\left\| \sum_{s=1}^{t} \mathbf{x}_s \eta_s \right\|_{\mathbf{\Lambda}_t^{-1}} \leq 8\sigma \sqrt{d \log \left( 1 + \frac{tL^2}{\lambda d} \right) \cdot \log \left( \frac{4t^2}{\delta} \right)} + 4R \log \left( \frac{4t^2}{\delta} \right)$$

**Lemma E.11** (Lemma H.4 in [Yin et al., 2022]). *Let $\Lambda_1$ and $\Lambda_2 \in \mathbb{R}^{d \times d}$ be two positive semi-definite matrices. Then:*
$$\|\Lambda_1^{-1}\| \leq \|\Lambda_2^{-1}\| + \|\Lambda_1^{-1}\| \cdot \|\Lambda_2^{-1}\| \cdot \|\Lambda_1 - \Lambda_2\|$$

*and*

$$\|\phi\|_{\Lambda_1^{-1}} \leq \left[ 1 + \sqrt{\|\Lambda_2^{-1}\| \cdot \|\Lambda_2\| \cdot \|\Lambda_1^{-1}\| \cdot \|\Lambda_1 - \Lambda_2\|} \right] \cdot \|\phi\|_{\Lambda_2^{-1}}.$$

*for all $\phi \in \mathbb{R}^d$.*

**Lemma E.12** (Lemma H.4 in [Min et al., 2021]). *Let $\phi : \mathcal{S} \times \mathcal{A} \to \mathbb{R}^d$ satisfies $\|\phi(s, a)\| \leq C$ for all $s, a \in \mathcal{S} \times \mathcal{A}$. For any $K > 0, \lambda > 0$, define $\bar{G}_K = \sum_{k=1}^{K} \phi(s_k, a_k)\phi(s_k, a_k)^{\top} + \lambda I_d$ where $(s_k, a_k)$'s are i.i.d samples from some distribution $\nu$. Then with probability $1 - \delta$,*

$$\left\| \frac{\bar{G}_K}{K} - \mathbb{E}_{\nu}\left[ \frac{\bar{G}_K}{K} \right] \right\| \leq \frac{4\sqrt{2}C^2}{\sqrt{K}} \left( \log \frac{2d}{\delta} \right)^{1/2}.$$

**Lemma E.13** (Lemma H.5 in [Min et al., 2021]). *Let $\phi : \mathcal{S} \times \mathcal{A} \to \mathbb{R}^d$ be a bounded function s.t. $\|\phi\|_2 \leq C$. Define $\bar{G}_K = \sum_{k=1}^{K} \phi(s_k, a_k)\phi(s_k, a_k)^{\top} + \lambda I_d$ where $(s_k, a_k)$'s are i.i.d samples from some distribution $\nu$. Let $G = \mathbb{E}_{\nu}[\phi(s, a)\phi(s, a)^{\top}]$. Then for any $\delta \in (0, 1)$, if $K$ satisfies*

$$K \geq \max \left\{ 512C^4 \left\| \mathbf{G}^{-1} \right\|^2 \log \left( \frac{2d}{\delta} \right), 4\lambda \left\| \mathbf{G}^{-1} \right\| \right\}.$$

*Then with probability at least $1 - \delta$, it holds simultaneously for all $u \in \mathbb{R}^d$ that*

$$\|u\|_{\bar{G}_K^{-1}} \leq \frac{2}{\sqrt{K}} \|u\|_{G^{-1}}.$$

**Lemma E.14** (Lemma H.9 in [Yin et al., 2022]). *For a linear MDP, for any $0 \leq V(\cdot) \leq H$, there exists a $w_h \in \mathbb{R}^d$ s.t. $\mathcal{T}_h V = \langle \phi, w_h \rangle$ and $\|w_h\|_2 \leq 2H\sqrt{d}$ for all $h \in [H]$. Here $\mathcal{T}_h(V)(s, a) = r_h(x, a) + (P_h V)(s, a)$. Similarly, for any $\pi$, there exists $w_h^{\pi} \in \mathbb{R}^d$, such that $Q_h^{\pi} = \langle \phi, w_h^{\pi} \rangle$ with $\|w_h^{\pi}\|_2 \leq 2(H - h + 1)\sqrt{d}$.*

### E.3  Assisting lemmas for differential privacy

**Lemma E.15** (Converting zCDP to DP [Bun and Steinke, 2016]). *If M satisfies $\rho$-zCDP then M satisfies $(\rho + 2\sqrt{\rho \log(1/\delta)}, \delta)$-DP.*

**Lemma E.16** (zCDP Composition [Bun and Steinke, 2016]). *Let $M : \mathcal{U}^n \to \mathcal{Y}$ and $M' : \mathcal{U}^n \to \mathcal{Z}$ be randomized mechanisms. Suppose that M satisfies $\rho$-zCDP and $M'$ satisfies $\rho'$-zCDP. Define $M'' : \mathcal{U}^n \to \mathcal{Y} \times \mathcal{Z}$ by $M''(U) = (M(U), M'(U))$. Then $M''$ satisfies $(\rho + \rho')$-zCDP.*

**Lemma E.17** (Adaptive composition and Post processing of zCDP [Bun and Steinke, 2016])**.** *Let* $M : \mathcal{X}^n \rightarrow \mathcal{Y}$ *and* $M' : \mathcal{X}^n \times \mathcal{Y} \rightarrow \mathcal{Z}$*. Suppose $M$ satisfies $\rho$-zCDP and $M'$ satisfies $\rho'$-zCDP (as a function of its first argument). Define $M'' : \mathcal{X}^n \rightarrow \mathcal{Z}$ by $M''(x) = M'(x, M(x))$. Then $M''$ satisfies $(\rho + \rho')$-zCDP.*

**Definition E.18** ($\ell_1$ sensitivity)**.** *Define the $\ell_1$ sensitivity of a function $f : \mathbb{N}^{\mathcal{X}} \mapsto \mathbb{R}^d$ as*

$$\Delta_1(f) = \sup_{neighboring\ U, U'} \|f(U) - f(U')\|_1.$$

**Definition E.19** (Laplace Mechanism [Dwork et al., 2014])**.** *Given any function $f : \mathbb{N}^{\mathcal{X}} \mapsto \mathbb{R}^d$, the Laplace mechanism is defined as:*

$$\mathcal{M}_L(x, f, \epsilon) = f(x) + (Y_1, \cdots, Y_d),$$

*where $Y_i$ are i.i.d. random variables drawn from $\mathrm{Lap}(\Delta_1(f)/\epsilon)$.*

**Lemma E.20** (Privacy guarantee of Laplace Mechanism [Dwork et al., 2014])**.** *The Laplace mechanism preserves $(\epsilon, 0)$-differential privacy. For simplicity, we say $\epsilon$-DP.*

# F    Details for the Evaluation part

In the Evaluation part, we apply a synthetic linear MDP case that is similar to [Min et al., 2021, Yin et al., 2022] but with some modifications for our evaluation task. The linear MDP example we use consists of $|\mathcal{S}| = 2$ states and $|\mathcal{A}| = 100$ actions, while the feature dimension $d = 10$. We denote $\mathcal{S} = \{0, 1\}$ and $\mathcal{A} = \{0, 1, \ldots, 99\}$ respectively. For each action $a \in \{0, 1, \ldots, 99\}$, we obtain a vector $\mathbf{a} \in \mathbb{R}^8$ via binary encoding. More specifically, each coordinate of $\mathbf{a}$ is either $0$ or $1$.

First, we define the following indicator function $\delta(s, a) = \begin{cases} 1 & \text{if } \mathbb{1}\{s = 0\} = \mathbb{1}\{a = 0\} \\ 0 & \text{otherwise} \end{cases}$ , then

our non-stationary linear MDP example can be characterized by the following parameters.

The feature map $\phi$ is:
$$\phi(s, a) = \left(\mathbf{a}^{\top}, \delta(s, a), 1 - \delta(s, a)\right)^{\top} \in \mathbb{R}^{10}.$$
The unknown measure $\nu_h$ is:
$$\boldsymbol{\nu}_h(0) = (0, \cdots, 0, \alpha_{h,1}, \alpha_{h,2}),$$
$$\boldsymbol{\nu}_h(1) = (0, \cdots, 0, 1 - \alpha_{h,1}, 1 - \alpha_{h,2}),$$
where $\{\alpha_{h,1}, \alpha_{h,2}\}_{h \in [H]}$ is a sequence of random values sampled uniformly from $[0, 1]$.
The unknown vector $\theta_h$ is:
$$\theta_h = (r_h/8, 0, r_h/8, 1/2 - r_h/2, r_h/8, 0, r_h/8, 0, r_h/2, 1/2 - r_h/2) \in \mathbb{R}^{10},$$

where $r_h$ is also sampled uniformly from $[0, 1]$. Therefore, the transition kernel follows $P_h(s'|s, a) = \langle \phi(s, a), \boldsymbol{\nu}_h(s') \rangle$ and the expected reward function $r_h(s, a) = \langle \phi(s, a), \theta_h \rangle$.
Finally, the behavior policy is to always choose action $a = 0$ with probability $p$, and other actions uniformly with probability $(1 - p)/99$. Here we choose $p = 0.6$. The initial distribution is a uniform distribution over $\mathcal{S} = \{0, 1\}$.

