# Offline Reinforcement Learning with Differential Privacy

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

 D.3. The fifth inequality is because of Remark D.8. The last inequality is by Lemma D.7.

Below we present a crude bound of $\left| V_t^{\dagger \pi^\star}(s) - \widetilde{V}_t(s) \right|$, which can be further used to bound the main term in the main result.

**Lemma D.11** (Self-bounding, private version of Lemma D.7 in [Yin and Wang, 2021b]). *Under the high probability events in Lemma D.3, Lemma D.6 and Lemma D.7, it holds that for all $t \in [H]$ and $s \in \mathcal{S}$,*

$$
\left| V_t^{\dagger \pi^\star}(s) - \widetilde{V}_t(s) \right| \leq \frac{4\sqrt{2\iota} H^2}{\sqrt{n \cdot \bar{d}_m}} + \frac{256 S H^2 E_\rho \cdot \iota}{3 n \cdot \bar{d}_m}.
$$

*where $\bar{d}_m$ is defined in Theorem 3.4.*

*Proof of Lemma D.11.* According to (31), since $\mathrm{Var}_{\widetilde{P}_h(\cdot \mid s_h, a_h)}(\widetilde{V}_{h+1}(\cdot)) \leq H^2$, we have for all $t \in [H]$,

$$
\left| V_t^{\dagger \pi^\star}(s) - V_t^{\dagger \widehat{\pi}}(s) \right| \leq \frac{4\sqrt{2\iota} H^2}{\sqrt{n \cdot \bar{d}_m}} + \frac{256 S H^2 E_\rho \cdot \iota}{3 n \cdot \bar{d}_m}
\tag{32}
$$

Next, apply Lemma F.7 by setting $\pi = \widehat{\pi}, \pi' = \pi^\star, \widehat{Q} = \overline{Q}, \widehat{V} = \widetilde{V}$ under $M^\dagger$, then we have

$$
\begin{aligned}
V_t^{\dagger \pi^\star}(s) - \widetilde{V}_t(s) &= \sum_{h=t}^{H} \mathbb{E}_{\pi^\star}^\dagger \left[ \xi_h^\dagger(s_h, a_h) \mid s_t = s \right] + \sum_{h=t}^{H} \mathbb{E}_{\pi^\star}^\dagger \left[ \langle \overline{Q}_h(s_h, \cdot), \pi_h^\star(\cdot|s_h) - \widehat{\pi}_h(\cdot|s_h) \rangle \mid s_t = s \right] \\
&\leq \sum_{h=t}^{H} \mathbb{E}_{\pi^\star}^\dagger \left[ \xi_h^\dagger(s_h, a_h) \mid s_t = s \right] \\
&\leq \frac{4\sqrt{2\iota}H^2}{\sqrt{n \cdot \bar{d}_m}} + \frac{256 S H^2 E_\rho \cdot \iota}{3n \cdot \bar{d}_m}.
\end{aligned}
\tag{33}
$$

Also, apply Lemma F.7 by setting $\pi = \pi' = \widehat{\pi}, \widehat{Q} = \overline{Q}, \widehat{V} = \widetilde{V}$ under $M^\dagger$, then we have

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

### E.4 Put everything toghther

Combining Lemma E.1, Theorem E.15, and the discussion above, the proof of Theorem 4.1 is complete.

## F Assisting technical lemmas

**Lemma F.1** (Multiplicative Chernoff bound [Chernoff et al., 1952])**.** *Let $X$ be a Binomial random variable with parameter $p, n$. For any $1 \ge \theta > 0$, we have that*

$$\mathbb{P}[X < (1-\theta)pn] < e^{-\frac{\theta^2 pn}{2}}, \quad \text{and} \quad \mathbb{P}[X \ge (1+\theta)pn] < e^{-\frac{\theta^2 pn}{3}}$$

**Lemma F.2** (Hoeffding's Inequality [Sridharan, 2002])**.** *Let $x_1,...,x_n$ be independent bounded random variables such that $\mathbb{E}[x_i] = 0$ and $|x_i| \le \xi_i$ with probability 1. Then for any $\epsilon > 0$ we have*

$$\mathbb{P}\left(\frac{1}{n}\sum_{i=1}^{n} x_i \ge \epsilon\right) \le e^{-\frac{2n^2\epsilon^2}{\sum_{i=1}^n \xi_i^2}}.$$

**Lemma F.3** (Bernstein's Inequality). *Let $x_1, ..., x_n$ be independent bounded random variables such that $\mathbb{E}[x_i] = 0$ and $|x_i| \leq \xi$ with probability $1$. Let $\sigma^2 = \frac{1}{n}\sum_{i=1}^{n} \text{Var}[x_i]$, then with probability $1 - \delta$ we have*

$$\frac{1}{n}\sum_{i=1}^{n} x_i \leq \sqrt{\frac{2\sigma^2 \cdot \log(1/\delta)}{n}} + \frac{2\xi}{3n}\log(1/\delta).$$

**Lemma F.4** (Empirical Bernstein's Inequality [Maurer and Pontil, 2009]). *Let $x_1, ..., x_n$ be i.i.d random variables such that $|x_i| \leq \xi$ with probability $1$. Let $\bar{x} = \frac{1}{n}\sum_{i=1}^{n} x_i$ and $\widehat{V}_n = \frac{1}{n}\sum_{i=1}^{n}(x_i - \bar{x})^2$, then with probability $1 - \delta$ we have*

$$\left|\frac{1}{n}\sum_{i=1}^{n} x_i - \mathbb{E}[x]\right| \leq \sqrt{\frac{2\widehat{V}_n \cdot \log(2/\delta)}{n}} + \frac{7\xi}{3n}\log(2/\delta).$$

**Lemma F.5** (Lemma I.8 in [Yin and Wang, 2021b]). *Let $n \geq 2$ and $V \in \mathbb{R}^S$ be any function with $||V||_\infty \leq H$, $P$ be any $S$-dimensional distribution and $\widehat{P}$ be its empirical version using $n$ samples. Then with probability $1 - \delta$,*

$$\left|\sqrt{\text{Var}_{\widehat{P}}(V)} - \sqrt{\frac{n-1}{n}\text{Var}_P(V)}\right| \leq 2H\sqrt{\frac{\log(2/\delta)}{n-1}}.$$

**Lemma F.6** (Claim 2 in [Vietri et al., 2020]). *Let $y \in \mathbb{R}$ be any positive real number. Then for all $x \in \mathbb{R}$ with $x \geq 2y$, it holds that $\frac{1}{x-y} \leq \frac{1}{x} + \frac{2y}{x^2}$.*

### F.1 Extended Value Difference