# OpenReview forum: "Offline Reinforcement Learning with Differential Privacy"
_NeurIPS.cc/2023/Conference — NeurIPS 2023 poster_

### Official Review · Reviewer_ytfu · 2023-06-29

**Soundness:** 3 good
**Presentation:** 3 good
**Contribution:** 3 good
**Rating:** 7
**Confidence:** 4

**Summary:**

This paper proposes two offline RL algorithms with differential privacy guarantees. The two pessimism-based algorithms apply to both tabular and linear MDP settings. In theory, the authors prove that the proposed algorithms achieve instance-dependent sub-optimality bounds while guaranteeing differential privacy. A nice property is that the cost of privacy only appears as lower order terms, thus become negligible as the number of samples goes large.


**Strengths:**

(1) While DP algorithm has been studied in online RL settings, the study of DP in offline RL is limited. This paper provides the first provable study along this line.

(2) In spite of a theoretical paper, it is well written and easy to follow.


**Weaknesses:**


(1) The motivation of the considered problem needs more real justifications. In the introduction, the authors used a medical example to motivate the need of considering privacy in offline RL. Why could not the data owner (hospital or doctors) do offline policy evaluation or policy optimization directly on the raw data? Why do we need to generate a private policy? On the other hand, if the offline policy optimization is requested from a third-part not the data owner, it is justified that the patient's data needs to be protected. However, in this case, the third-part would not have access to the raw data. This contradicts with the setting considered in this paper as in the proposed Algorithm 1 and Algorithm 2, the input is the raw data. Therefore, it is important and helpful to provide a convincing real example to justify the problem setting and algorithm designs.

(2) In Assumption 2.2, the data distribution needs to satisfy the minimum eigenvalue condition. This assumption might be violated when the feature space is large or some features are highly correlated. It is helpful to provide some discussion on this assumption and how to remedy it when this assumption is violated.

(3) In the tabular MDP (DP-APVI) algorithm, the Gaussian noise is added to the integer counts $n_{s_h, a_h}$ and $n_{s_h, a_h, s_{h+1}}$ to obtain a private estimation of the transition kernel. Because of the design, the private counts might be negative or very small, which makes the uncertainty estimation (line 5 of Algorithm 1) to be unstable. To handle it, the authors used some truncation approach with some theoretical truncation rate $E_{\rho}$. It is unclear if it is a good choice to add the Gaussian noise to the count statistics. Can we add the Gaussian noise directly to the non-private estimation of transition kernel? Some justifications on the proposed private estimation would be helpful. It is also important to discuss the advantages and limitations of the proposed private estimation while comparing with other choices in existing RL literature.

(4) In Algorithm 2 and line 235 of Page 7, the authors require two independent offline dataset with equal length. Can authors clarify what the rigorous condition of the "two independent offline dataset"? Do you need each sample (across $K$) at each time horizon (across $H$) to be independent in each offline data? If yes, this is an unrealistic condition in offline RL problem. Moreover, I did not find such condition in the statement of Theorem 4.1. Where do you use this "independence" assumption?

(5) In the experiments (Figure 1), the simulations are for 5 replicates. Can authors include the uncertainty in Figure 1? If the uncertainty is large, it is helpful to increase the replication times.


~~~~~~~~~~~
After rebuttal: my major comments have been nicely addressed. I have increased the score to 7.

**Questions:**

See Weakness section

**Limitations:**

None.

---

> ### Author Rebuttal · Authors · 2023-08-08
>
> Thanks for your high-quality review and your support. We will reply to the weaknesses you stated.
>
> **In the introduction, the authors used a medical example to motivate the need of considering privacy in offline RL. Why could not the data owner (hospital or doctors) do offline policy evaluation or policy optimization directly on the raw data? Why do we need to generate a private policy?**
>
> Even if the owner has access to the raw data, it is risky to work directly on the raw data. If the data owner do offline RL directly on the raw data, there will be risk of privacy leakage. For instance, membership inference attack ([1]) could detect the data used in the training procedure by observing the output policy of the offline RL algorithm. In other words, the attacker could reconstruct the medical history of the patients whose data is used for training the model, which is harmful to the privacy of those patients. In contrast, training the model with DP guarantee could provably prevent such risks.
>
> [1] R. Shokri, M. Stronati, C. Song, and V. Shmatikov. Membership inference attacks against machine learning models.
>
> **On the other hand, if the offline policy optimization is requested from a third-part not the data owner, it is justified that the patient's data needs to be protected. However, in this case, the third-part would not have access to the raw data. This contradicts with the setting considered in this paper as in the proposed Algorithm 1 and Algorithm 2, the input is the raw data.**
>
> This is a very good point. Local DP, where the data needs to be privatized before being sent to the algorithm is used to characterize such situation. Under such circumstances, noise should be added to the raw data, and then the algorithm would aggregate the privatized data and learn the policy. This is an interesting but different problem from our setting. We believe that this could be a good future direction.
>
> **About Assumption 2.2: the minimum eigenvalue condition.**
>
> First, we would like to highlight that the DP guarantee does not depend on this assumption, and the assumption is only used to derive sub-optimality bounds. In addition, such coverage assumption is standard in offline RL literature since without such assumption, there will be a constant sub-optimality gap for the output policy even without constraints of DP, in the worst case ([2]). Therefore, we base on the coverage assumption and derive an asymptotic sub-optimality bound. We will add the discussions in the revision.
>
> [2] Ming Yin and Yu-Xiang Wang. Towards instance-optimal offline reinforcement learning with pessimism. NeurIPS 2021.
>
>
> **Can we add the Gaussian noise directly to the non-private estimation of transition kernel? Some justifications on the proposed private estimation would be helpful. It is also important to discuss the advantages and limitations of the proposed private estimation while comparing with other choices in existing RL literature.**
>
> Adding noise to visitation counts is a widely applied method in DP RL literature ([3],[4]). It is a possible choice to add noise to the empirical transition kernel, while there are two issues for the method. First, given the sensitivity analysis, the $\ell_1$ difference between the private and non-private transition kernel estimates will be the same order as our approach. Second, in this way, the private transition kernel will not be a valid probability distribution, which is not applicable for our Bernstein-type pessimism. In comparison, we effectively solve the second issue through solving the optimization problem.
>
> Similar approaches of private estimation also appears in the DP online RL literature. [4] directly adds Gaussian noise to the visitation counts and constructs the private transition kernel estimate. As a result, [4] could only operate on the Hoeffding-type bonus and derive sub-optimal regret bounds. Recently, [3] follows our construction of the private transition kernel estimate and operates on the Bernstein-type bonus. As a result, [3] improves the regret bound in [4] and derives nearly minimax optimal results. Therefore, our method for privately estimating the transition kernel is not improvable in general.
>
> [3] Dan Qiao and Yu-Xiang Wang. Near-Optimal Differentially Private Reinforcement Learning. AISTATS, 2023.
>
> [4] Sayak Ray Chowdhury and Xingyu Zhou. Differentially private regret minimization in episodic
> markov decision processes. arXiv preprint arXiv:2112.10599, 2021.
>
> **In Algorithm 2 and line 235 of Page 7, the authors require two independent offline dataset with equal length. Can authors clarify what the rigorous condition of the "two independent offline dataset"? Do you need each sample (across $K$) at each time horizon (across $H$) to be independent in each offline data? If yes, this is an unrealistic condition in offline RL problem. Moreover, I did not find such condition in the statement of Theorem 4.1. Where do you use this "independence" assumption?**
>
> This is a good question. Instead of requiring the data to be independent at each time horizon, we only require each trajectory (across $K$) to be independent, and this is naturally ensured by our offline RL setting where the data is sampled i.i.d according to some behavior policy. The independence of the two datasets is only for technical reason, which is to ensure that the dataset for weighted ridge regression is sampled i.i.d from the behavior policy even given the estimated variances. We will clarify this in the revision.
>
> **In the experiments (Figure 1), the simulations are for 5 replicates. Can authors include the uncertainty in Figure 1? If the uncertainty is large, it is helpful to increase the replication times.**
>
> We will add the confidence interval in the revision.
>
> Thanks again for the helpful review! We hope that our response could address your main concerns. We are open to further discussions.

---

> ### Author Response · Authors · 2023-08-17
>
> Thank you for your positive feedback and increasing the score. We will include the discussions in the final version according to your comments.

---

### Official Review · Reviewer_7uyZ · 2023-07-06

**Soundness:** 3 good
**Presentation:** 3 good
**Contribution:** 4 excellent
**Rating:** 7
**Confidence:** 3

**Summary:**

This paper addresses the offline RL with Differential Privacy constraints problem. Tabular and linear MDP are considered, while both forms of DP, traditional DP and zCDP are studied. The authors cast the DP definition into the offline RL problem as a constraint for protecting trajectories. Two algorithms, DP-APVI (resp. DP-VAPVI), are introduced, treating the tabular (resp. linear) case. Those algorithms rely on the pessimism principle, and DP is obtained by adding a Gaussian mechanism, either on the counts or during variance estimation. Authors prove zCDP compliance for both algorithms, and a comparison with non-private algorithms is provided. Finally, authors empirically evaluate the performance of DP-VAPVI under different privacy budgets.

**Strengths:**

The main strength of this paper is that it is, according to my knowledge, the first to tackle the offline RL with DP constraints, which is an important problem. This paper introduce two sound and practical algorithms, respecting Differential Privacy by design. Furthermore, we  have a discussion comparing the algorithms with their non-private counterpart and experimental validation.

**Weaknesses:**

- Globally, the paper is well-written, however, I found the section on DP-VAPVI very hard to follow, especially the algorithm
- Experiments study the performance of the algorithms under differential privacy budgets, but it feels very hard to have an interpretation of those budgets for zCDP. In the traditional DP case, one often sees very high values of \epsilon, leading to a discussable Differentially Private algorithm in practice. Would it be possible to have a discussion about \alpha?  What would be reasonable values?
- Although there may not be other paper directly studying directly DP offline RL, there is work in literature on privacy attacks in RL, such as membership inference attack: R. Shokri, M. Stronati, C. Song, and V. Shmatikov. Membership inference attacks against machine learning models) or Maziar Gomrokchi, Susan Amin, Hossein Aboutalebi, Alexander Wong, and Doina Precup. Membership Inference Attacks Against Temporally Correlated Data in Deep Reinforcement Learning. It would have been nice to have a discussion on those attacks to have a better estimation of the practical impact of DP casted like this.

**Questions:**

- See section weakness, could the authors comment on the \rho factor? And would it allow to robustness to membership inference attacks?
- You only consider an MDP with 2 states in the experiments, would the algorithm scale to bigger MDPs?

**Limitations:**

yes

---

> ### Author Rebuttal · Authors · 2023-08-08
>
> Thanks for your high-quality review and your support. We will reply to the weaknesses you stated.
>
> **Globally, the paper is well-written, however, I found the section on DP-VAPVI very hard to follow, especially the algorithm.**
>
> We apologize for this and we will improve the readability in the revision. Briefly speaking, the non-private counterpart of DP-VAPVI contains three parts: variance estimation, weighted ridge regression and adding pessimism. In the first part, the variance of value function is estimated. In the second part, the algorithm applies weighted ridge regression based on the estimated variances to derive tighter results. Finally, pessimism is added to the estimated value function. To achieve differential privacy, we add Guassian noises to sufficient statistics. We will surely improve the writing according to your suggestions.
>
> **Experiments study the performance of the algorithms under differential privacy budgets, but it feels very hard to have an interpretation of those budgets for zCDP. In the traditional DP case, one often sees very high values of $\epsilon$, leading to a discussable Differentially Private algorithm in practice. Would it be possible to have a discussion about $\rho$? What would be reasonable values?**
>
> That's a very good point and we will add the discussions. Briefly speaking, the budget for zCDP can be transferred to the budget for DP and $\rho$ is roughly $\min(\epsilon,\epsilon^2/2)$. Therefore, smaller $\rho$ provides stronger privacy protection and a small constant would be a reasonable value for the zCDP budget. For instance, [1] takes the privacy budget to be around 1. In practice, any $\rho<10$ should be considered meaningful privacy guarantees. The choice of $\rho$ is often cast as a social problem and the preferred value differ in each application. For this reason, we experimented on a range of different $\rho$ in our simulation.
>
> [1] Xu et al. Federated Learning of Gboard Language Models with Differential Privacy. arXiv:2305.18465. 2023.
>
> **Although there may not be other paper directly studying directly DP offline RL, there is work in literature on privacy attacks in RL, such as membership inference attack: R. Shokri, M. Stronati, C. Song, and V. Shmatikov. Membership inference attacks against machine learning models) or Maziar Gomrokchi, Susan Amin, Hossein Aboutalebi, Alexander Wong, and Doina Precup. Membership Inference Attacks Against Temporally Correlated Data in Deep Reinforcement Learning. It would have been nice to have a discussion on those attacks to have a better estimation of the practical impact of DP casted like this.**
>
> Thanks for pointing us to the literature regarding membership inference attack. The guarantee of DP ensures that even if all the other training data points are known, it is statistically hard to predict whether some data point appears in the training dataset. More specifically, given a fixed type-I error rate, DP guarantee provides a lower bound for the type-II error rate of the membership inference attack. Therefore, a DP algorithm provably provides robustness against membership inference attacks.
>
> **You only consider an MDP with 2 states in the experiments, would the algorithm scale to bigger MDPs?**
>
> We use the same MDP setting as previous papers regarding offline RL. The algorithm can be applied under MDPs with larger state space as well as it admits linear structure. We leave experiments on more complex benchmarks such as D4RL (which will require designing deep offline RL algorithms with DP) as a future direction.
>
> Thanks again for the helpful review! We hope that our response could address your main concerns. We are open to further discussions.

---

> > ### Comment · Reviewer_7uyZ · 2023-08-18
> >
> > My major comments have been addressed. I believe that contributions are enough for an acceptance and I therefore maintain my score.

---

> > > ### Author Response · Authors · 2023-08-18
> > >
> > > Thank you for your positive feedback. We will include the discussions in the final version according to your comments.

---

### Official Review · Reviewer_ZbTu · 2023-07-07

**Soundness:** 3 good
**Presentation:** 3 good
**Contribution:** 3 good
**Rating:** 7
**Confidence:** 5

**Summary:**

This paper focuses on reinforcement learning (RL) in an offline setting under differential privacy considerations. It studies the proposal and analysis of a new approach to learning policies in this specific setting, leveraging the Bernstein concentration inequality.

Reinforcement learning explores an agent's interaction with its environment over a series of episodes. The environment is defined as a Markov decision process (MDP) comprising states, actions available to the agent, a reward function, and transition dynamics. Through this interaction, the agent gathers feedback to learn a policy that optimizes the cumulative reward. In this paper, the agent operates under the constraints of differential privacy, implying that it interacts in an environment with sensitive data. Here, observations from each episode are deemed private, being tied to individual users.

Traditionally, two settings have been studied: the online and offline settings. In the online setting, an agent learns a policy through active data collection by exploring the environment. Conversely, the offline setting involves the agent receiving all data upfront, barring any access to the environment for training. Prior studies have explored these settings without privacy constraints, but only the online setting has been studied under differential privacy.

This paper breaks new ground by studying reinforcement learning in the offline setting under differential privacy. Like its online counterpart, the offline setting necessitates the construction of tight confidence bounds around sufficient statistics for policy approximation. However, due to privacy-induced noise, the algorithm must inflate the confidence bounds to compensate.
The primary innovation of this paper is the proposal of new confidence bounds, utilizing the Bernstein concentration inequality, a departure from the traditional Hoeffding concentration. This novel method offers improved error bounds under specific instance-dependent conditions, although the bounds are equivalent to Hoeffding's in the worst-case scenario.

The authors cleverly adapt techniques from previous research while also introducing new methodologies that could potentially extend beyond this work. A significant contribution lies in the estimation of variance from noisy statistics, thereby enabling the utilization of the Bernstein concentration.

In sum, this paper illuminates a hitherto unexplored area in the RL community, offering innovative solutions and techniques for reinforcement learning in the offline setting under differential privacy.


**Strengths:**

The primary strength of this paper lies in its pioneering examination of differential privacy reinforcement learning (DP-RL) in the offline setting, effectively bridging a notable gap in existing academic literature. By venturing into this uncharted territory, the paper opens up possibilities for practical applications in real-world scenarios where sensitive data may be involved.

The paper's successful application of differential privacy to offline RL deserves commendation, given the unique challenges associated with this endeavor. While one might assume that the principles governing DP-RL in the online setting would seamlessly translate to the offline setting, this is not the case. The paper adeptly navigates these challenges, setting a benchmark for future investigations into this area.
Another significant contribution of this work is the innovative use of Bernstein concentration in the estimation of error bounds. Prior studies primarily employed Hoeffding concentration, which, although effective, offered limited utility under specific conditions. In contrast, Bernstein concentration proves to be more flexible, offering improved error bounds, particularly under specific instance-dependent conditions. This innovative application, therefore, has the potential to enhance results not only in the offline setting but also in the online RL environment.

Furthermore, the paper presents an empirical evaluation that provides rich insights into the practical implications of their proposed techniques. This robust empirical analysis not only validates the theoretical contributions but also offers tangible results that underscore the effectiveness of the proposed methods.

In conclusion, the paper's merits extend from filling a knowledge void in DP-RL literature, effectively tackling challenges in translating DP-RL principles from online to offline settings, to innovatively employing Bernstein concentration in error bound estimation. The comprehensive empirical evaluation serves as the icing on the cake, demonstrating the practicality and effectiveness of the proposed methods. The paper's contributions are both theoretical and practical, promising to advance understanding and application of DP-RL in offline settings.


**Weaknesses:**

Many of the techniques presented in this paper are not new. The concentration bounds utilized, for instance, have been previously developed and employed in other works. However, the paper's merit lies in its successful adaptation of these existing techniques for a specific context. Therefore, this paper effectively demonstrates how to repurpose these pre-existing tools for its unique setting, thereby contributing to the literature in a meaningful way.


**Questions:**

No questions.

**Limitations:**

Yes.

---

> ### Author Rebuttal · Authors · 2023-08-08
>
> Thanks for your high-quality review and your support. We really appreciate the detailed and insightful summary. For the weakness, we agree that part of the techniques (including Gaussian mechanism and Bernstein-type pessimism) have been studied. Our main technical contribution is to privatize Bernstein-type pessimism to get tight sub-optimality bounds, as discussed in your review.
>
> Thanks again for the appreciation of our work. We are open to further discussions.

---

> ### Comment · Reviewer_ZbTu · 2023-08-19
>
> I read all reviews and rebuttals. For now I will maintain my score.

---

> > ### Author Response · Authors · 2023-08-19
> >
> > Thanks for your positive feedback. We really appreciate your support and your insightful review.

---

### Official Review · Reviewer_fvW2 · 2023-07-14

**Soundness:** 3 good
**Presentation:** 3 good
**Contribution:** 3 good
**Rating:** 5
**Confidence:** 2

**Summary:**

The paper proposes an algorithm for offline reinforcement learning with differential privacy (DP), which protects the privacy of the original information using a Gaussian mechanism based on pessimism. The motivation and ideas behind the paper are clear and meaningful. However, there are some issues with the methods and experiments.

Overall, the paper presents some interesting ideas, but additional experiments and comparisons are needed to fully evaluate the proposed method and its practical value.


**Strengths:**

1.The authors propose a method for protecting the privacy of original information in offline reinforcement learning. They achieve this goal by implementing differential privacy (DP) in their proposed method, which is a meaningful contribution to the field of privacy-preserving machine learning.
2. The authors implement their ideas in APVI and VAPVI models and provide a thorough theoretical analysis of the proposed method. The writing is clear and the theoretical analysis is extensive, providing a strong foundation for the authors' claims.
3. The authors conduct experiments on simulated datasets, which provide preliminary evidence of the method's performance.


**Weaknesses:**

The paper proposes two models, DP-APVI and DP-VAPVI, for solving the offline reinforcement learning problem with privacy guarantees. While the paper presents some interesting ideas, there are several issues:
1. The paper only includes results for DP-VAPVI, and does not provide any experimental results for DP-APVI. It would be helpful to see how DP-APVI performs in comparison to DP-VAPVI and other baseline methods.
2. The results for DP-VAPVI consistently show a performance gap compared to VAPVI, which raises questions about the competitiveness of DP-VAPVI in practice. Without additional experiments or comparisons with other methods, it is difficult to assess the practical value of DP-VAPVI.
3. The paper claims that DP-VAPVI will converge to VAPVI as the dataset size increases, but there is no experimental evidence to support this claim.
4. The paper does not discuss the impact of the privacy budget (ρ) on the privacy protection and performance of the algorithms. It would be helpful to see how different values of ρ affect the results.
5. The paper lacks ablation experiments to investigate the extent to which DP itself as an optimization plug-in for APVI and VAPVI models maintains privacy and affects performance. It would be helpful to see how different components of the DP-APVI and DP-VAPVI models contribute to the overall performance.
6. The paper only compares DP-VAPVI and VAPVI with PEVI, which is not a privacy-preserving method. It would be helpful to see how DP-VAPVI and DP-APVI compare to other privacy-preserving methods.
7. The experiments are conducted on synthetic datasets, which may not fully reflect the complexity and diversity of real-world problems. It would be helpful to see how the proposed methods perform on real-world datasets.
Methodologically, the paper proposes to add additional Gaussian mechanisms to APVI and VAPVI models to ensure privacy, but the main methods are still based on APVI and VAPVI. While the paper's definition of neighboring datasets and the use of Gaussian mechanism for differential privacy are contributions, the experimental results are not convincing enough. The authors should demonstrate the advantages of their proposed methods on a wider range of datasets and models.

I'm not an expert on differential privacy. My evaluation is based on how well I was able to comprehend the information in the paper. I don't fully comprehend how this work contributes to the overall growth of the field.

**Questions:**

Please refer to the weakness

**Limitations:**

Please refer to the weakness

---

> ### Author Rebuttal · Authors · 2023-08-08
>
> Thanks for your high-quality review and positive score. We agree that we mainly use simulations to support our theories. Since we are taking the first step towards differential privacy under offline RL, we mainly analyze our algorithms through theory while only running simulations on toy examples. We leave real-world experiments (which will require designing deep offline RL algorithms with DP) as a future direction. We will reply to the weaknesses you stated.
>
> **The paper only includes results for DP-VAPVI, and does not provide any experimental results for DP-APVI. It would be helpful to see how DP-APVI performs in comparison to DP-VAPVI and other baseline methods.**
>
> We focus on simulations under linear MDPs since this is the first simulation under linear MDP. Previous works regarding DP RL under linear MDP only focus on theories. For comparison, DP-APVI and DP-VAPVI are not directly comparable since the settings are different. In addition, this is the first work studying offline RL with DP, and there are no available baselines, that's why we do not include baseline methods.
>
> **The paper does not discuss the impact of the privacy budget ($\rho$) on the privacy protection and performance of the algorithms. It would be helpful to see how different values of $\rho$ affect the results.**
>
> As the privacy budget $\rho$ goes larger, the privacy protection will be weaker, while the performance (sub-optimality) of the output policy will be better and closer to the non-private case. The impact of the choice of $\rho$ is shown in the theorems and Figure (b) of the experiments.
>
> **The paper only compares DP-VAPVI and VAPVI with PEVI, which is not a privacy-preserving method. It would be helpful to see how DP-VAPVI and DP-APVI compare to other privacy-preserving methods.**
>
> To the best of our knowledge, this is the first work studying offline RL with DP, and there is no other privacy-preserving methods for this task. Although there exists previous works regarding off-policy evaluation (OPE) with DP, they are not comparable to our methods. Therefore, we mainly compare the DP-VAPVI algorithm to its non-private counterpart.
>
> **The experiments are conducted on synthetic datasets, which may not fully reflect the complexity and diversity of real-world problems. It would be helpful to see how the proposed methods perform on real-world datasets.**
>
> This is a very good point and we agree that real-world experiments could help validate the theories for DP RL. However, real-world problems often admit more complex structures compared to linear MDPs (this is the reason why previous works regarding DP RL all focus on synthetic datasets). Experiments on more realistic offline RL benchmark such as D4RL may require incorporating various tricks from offline Deep RL literature, and we leave those as future work.
>
> Thanks again for the helpful review! We hope that our response could address your main concerns. We would greatly appreciate it if you could consider raising the score.

---

### Official Review · Reviewer_2mbC · 2023-07-14

**Soundness:** 3 good
**Presentation:** 3 good
**Contribution:** 3 good
**Rating:** 5
**Confidence:** 2

**Summary:**

This paper focuses on the offline RL problem with differential privacy. The authors propose algorithms for offline tabular MDP and offline linear MDP with $\rho$-DP. For the first problem, the sub-optimality bound almost matches the best-existing non-private counter-part in spite of an additional term $O(\sqrt{\frac{1}{\rho}})$. For the second problem, the gap between the proposed sub-optimality bound to the best existing non-private counter part is $\mathrm{poly}(d,H,\kappa^{-1})/\sqrt{\rho}$, where $\kappa$ is the minimal coverage parameter.

**Strengths:**

1. This paper firstly provides analysis for offline RL, and proposes error bounds which  matches the non-privacy counterparts up to some lower order terms.

2. In technique, the authors  make efforts to operate on Bernstein type pessimism while keeping privacy to achieve the tighter sub-optimality bound.

**Weaknesses:**

1. The technique novelty is  somewhat limited given literature in online RL with DP.

2. The discussion about related work is insufficient. In particular, I wonder what are the best existing regret bounds for online RL with DP (either tabular or linear MDPs)? If the regret bounds are not tight, could we use the Bernstein-style bonus with DP to improve the regret bounds?

**Questions:**

1. For the non-privacy problem, does the error bound have a polynomial dependence on $\kappa$ for the linear MDP problem?



**Limitations:**

Yes

---

> ### Author Rebuttal · Authors · 2023-08-08
>
> Thanks for your high-quality review and positive score. We will reply to the weaknesses you stated.
>
> **The technique novelty is somewhat limited given literature in online RL with DP.**
>
> We politely disagree. It is true that current techniques for online RL with DP can be adapted to the offline case. However, the current works for the online case all focus on privatizing Hoeffding-type bonuses. Therefore, directly adapting current online techniques to the offline case will not provide tight sub-optimality bounds. In comparison, we operate on Bernstein-type pessimism, which is highly non-trivial. We manage to provide a confidence bound using private pessimism and lower order additional terms, prove its validity and bound this private Bernstein pessimism by its non-private counterpart in our final result. All such techniques are novel to our knowledge.
>
> **The discussion about related work is insufficient. In particular, I wonder what are the best existing regret bounds for online RL with DP (either tabular or linear MDPs)? If the regret bounds are not tight, could we use the Bernstein-style bonus with DP to improve the regret bounds?**
>
> We discuss about the algorithms for online RL with joint DP (JDP) or local DP (LDP) in Appendix B. We did not discuss about the detailed regret bounds and kindly refer you to [1] for the results under tabular MDP and [2] for the results under linear MDP.
>
> Using the Bernstein-style bonus with DP to improve the regret bounds is a very good point. Under tabular MDP, the best result ([3]) with Hoeffding-type bonus is $\sqrt{SAH^3T}+$ additional cost due to DP, which is sub-optimal in the first term. Recently, following the idea of privatizing Bernstein-type bonus, a follow-up work ([1]) of this submission has improved the first term of the regret bound to $\sqrt{SAH^2T}$ (minimax optimal) while keeping the additional cost due to DP unchanged. Under linear MDP, the best known result ([2]) is derived by privatizing Hoeffding-type bonus. It is an open problem whether the regret can be improved through Bernstein-type bonus and we believe it is a good future direction.
>
> [1] Dan Qiao and Yu-Xiang Wang. Near-Optimal Differentially Private Reinforcement Learning. AISTATS, 2023.
>
> [2] Dung Daniel Ngo, Giuseppe Vietri, Zhiwei Steven Wu. Improved Regret for Differentially Private Exploration in Linear MDP. ICML, 2022.
>
> [3] Sayak Ray Chowdhury and Xingyu Zhou. Differentially private regret minimization in episodic markov decision processes. arXiv preprint arXiv:2112.10599, 2021.
>
> **For the non-privacy problem, does the error bound have a polynomial dependence on $\kappa$ for the linear MDP problem?**
>
> For the non-privacy problem, the error bound does not have polynomial dependence on $\kappa$. The main term in our bound is identical to the non-privacy case. The polynomial dependence on $\kappa$ only happens on the lower order term and such dependence results from bounding the worst case difference between private and non-private statistics.
>
> Thanks again for the helpful review! We hope that our response could address your main concerns. We would greatly appreciate it if you could consider raising the score.

---

### Decision · Program_Chairs · 2023-09-21

**Decision:**

Accept (poster)

**Comment:**

The paper makes its main contribution by introducing offline RL algorithms with differential privacy guarantees. The paper introduces algorithms for tabular and linear MDPs. It develops a proof technique based on Bernstein concentration. The paper also studies the cost of privacy, comparing theoretical results for algorithms that come with differential privacy guarantees with those that do not. Overall, this is a well-scoped and well-executed research project.  The main shortcomings are that (1) there are some remaining concerns about novelty / related work, (2) the experimental evaluation is limited (3) the impact of privacy budget not studied sufficiently, (4) more ablations would be better and (5) experiments are on synthetic datasets only. However, these are not fatal in a theoretical paper.